# Tree-Sliced Entropy Partial Transport

**Viet-Hoang Tran**[*]
Department of Mathematics
National University of Singapore
`hoang.tranviet@u.nus.edu`

**Thanh Tran**
College of Engineering & Computer Science
VinUniversity
`21thanh.tq@vinuni.edu.vn`

**Thanh Chu**
School of Computing
National University of Singapore
`thanh.chu@u.nus.edu`

**Tam Le**[†]
Department of Advanced Data Science
Institute of Statistical Mathematics
`tam@ism.ac.jp`

**Tan M. Nguyen**[†]
Department of Mathematics
National University of Singapore
`tanmn@nus.edu.sg`

## Abstract

Optimal Transport (OT) has emerged as a fundamental tool in machine learning for comparing probability distributions in a geometrically meaningful manner. However, a key limitation of classical OT is its requirement that the source and target distributions have equal total mass, limiting its use in real-world settings involving imbalanced data, noise, outliers, or structural inconsistencies. Partial Transport (PT) addresses this limitation by allowing only a fraction of the mass to be transported, offering greater flexibility and robustness. Nonetheless, similar to OT, PT remains computationally expensive, as it typically involves solving large-scale linear programs–especially in high-dimensional spaces. To alleviate this computational burden, several emerging works have introduced the Tree-Sliced Wasserstein (TSW) distance, which projects distributions onto tree-metric spaces where OT problems admit closed-form solutions. Building on this line of research, we propose a novel framework that extends the tree-sliced approach to the PT setting, introducing the Partial Tree-Sliced Wasserstein (PartialTSW) distance. Our method is based on the key observation that, within tree-metric space, the PT problem can be equivalently reformulated as a standard balanced OT problem between suitably modified measures. This reformulation enables efficient computation while preserving the adaptability and robustness of partial transport. Our method proves effective across challenging tasks such as outlier removal and addressing class imbalance in image-to-image translation. Our code is publicly available at `https://github.com/thanhqt2002/PartialTSW`.

## 1 Introduction

Optimal Transport (OT) [87, 61] is a framework for comparing probability distributions by lifting a ground cost defined between individual points to a metric over measures. Its ability to capture the

---

[*]Corresponding author
[†]Co–last author

39th Conference on Neural Information Processing Systems (NeurIPS 2025).

geometric structure of distributions leads widespread adoption across numerous domains, including machine learning [56, 11, 23, 32], data valuation [35, 38], multimodal data analysis [59, 49], statistics [50, 54, 57, 63], and computer vision [55, 70, 76, 89]. Despite its theoretical appeal, OT exhibits two major limitations in applications. First, the computational cost of OT problems for discrete measures scales as $\mathcal{O}(n^3 \log n)$ [61]. Second, the framework imposes a strict mass equality constraint, which is often violated due to noise, outliers, or unbalanced distributions [72].

**Sliced Optimal Transport.** To mitigate the high computational cost associated with OT, the Sliced Wasserstein (SW) distance [66, 10, 67] has been introduced as an efficient approximation. SW leverages the closed-form solution of one-dimensional OT by projecting high-dimensional probability measures onto one-dimensional subspaces, computing the OT cost in each slice, and subsequently averaging these costs. This procedure reduces the computational complexity to a sequence of sort-based operations with $\mathcal{O}(n \log n)$ complexity [61], while preserving key statistical and topological properties [52, 4, 29]. The SW framework has further inspired a wide range of generalizations, including extensions based on structured projections [39, 18, 53], as well as adaptations to manifold-valued and non-Euclidean domains such as the sphere [5, 65] and hyperbolic space [8].

**Tree-Sliced Optimal Transport.** One-dimensional projections, while computationally efficient, often fail to capture the intricate topological features inherent in high-dimensional data. To address this shortcoming, a growing body of work has explored richer integration domains as alternatives to linear projections in OT. These efforts span a variety of metric settings, including Euclidean spaces [1, 60, 58], tree metrics [46, 81], graph-based structures [45, 43], spherical geometries [65, 5, 83], and hyperbolic spaces [6, 48]. A seminal contribution in this direction is the tree system proposed by [81], which serves as a structurally enriched substitute for traditional lines. By leveraging established results and closed-form OT solutions on tree metric spaces [46, 34, 33], this framework introduces the Tree-Sliced Wasserstein (TSW) distance—a refined analogue of the classical SW distance. TSW retains the low computational complexity of SW while enhancing its capacity to reflect underlying data geometry. Recent advancements and extensions of the TSW framework are explored in [80, 79, 84].

**Partial Transport and Unbalanced Optimal Transport.** In various applied settings, it is often necessary to compare positive measures with unequal total mass—for instance, in biological applications where such measures represent cell populations of varying sizes [72]. The rigid mass conservation requirement of the classical OT can be relaxed using the Unbalanced OT (UOT) framework [73, 40, 47, 15, 22, 28], which introduces penalty terms that softly enforce mass preservation rather than enforcing it strictly. A related and widely used relaxation is Partial Transport (PT) [27, 9, 2, 68, 42, 43, 44], which allows only a fraction of the mass to be transported, thereby enabling more flexible alignment between distributions. PT improves robustness to outliers and facilitates meaningful comparisons under structural or statistical mismatches. It has also shown effectiveness in robust distributional alignment and has found applications in several domains, including deep learning theory [14, 69], cellular biology [72, 17], and domain adaptation [24, 3]. Despite its advantages, PT remains computationally intensive and is susceptible to noise in high-dimensional settings [20].

To address both the computational and mass imbalance issues, several approximate and scalable variants have been introduced, including entropic OT [16, 62] and minibatch OT [25, 24]. Recent work also has extended the sliced OT framework to the unbalanced setting [7, 22, 51, 28, 9, 2, 43], resulting in Sliced UOT variants with improved scalability and robustness.

**Contributions.** Motivated by the expanding TSW framework and recent advances in PT on tree metric spaces [42], this paper introduces a tree-sliced approach for computing partial transport between unbalanced measures in Euclidean spaces. The paper is organized as follows:

- In Section 2, we review the foundations of Optimal Transport and Entropy Partial Transport on metric spaces with tree metrics, as well as the Tree-Sliced Wasserstein distance for probability measures in Euclidean spaces based on tree systems. These concepts collectively form the theoretical foundation upon which the proposed framework is developed

- In Section 3, we formally introduce the Tree-Sliced Entropy Partial Transport (PartialTSW) distance for comparing probability measures with unbalanced mass in Euclidean spaces. We establish its metric properties and provide an analysis of its computational complexity.

- In Section 4, we empirically evaluate PartialTSW on challenging tasks, such as enhancing noise robustness for generative models and addressing class imbalance in image-to-image translation. The results underscore its practical effectiveness and computational efficiency. We conclude our work in Section 5.

All supplemental materials—including theoretical foundations, formal proofs, experimental settings accompanied by additional tables and figures, and a table of notation—are provided in the Appendix.

## 2 Building Blocks of Tree-Sliced Entropy Partial Transport

This section provides the foundations of Optimal and Entropy Partial Transport, as well as the Tree-Sliced Wasserstein distance. For the remainder of the paper, we denote the dimension by $d$.

### 2.1 Optimal and Entropy Partial Transports on Metric Spaces with Tree Metrics

**Tree Metric Spaces.** Let $\mathcal{T} = (V, E)$ be a tree rooted at node $r$, with nonnegative edge lengths $\{w_e\}_{e \in E}$. We identify $\mathcal{T}$ with the set of all nodes and points along its edges. For a metric space $\Omega$ with metric d, d is called a *tree metric* [75, 46] if there exists a tree $\mathcal{T}$ such that $\Omega \subset \mathcal{T}$ and $d(x, y)$ equals the length of the unique path between $x$ and $y$ for all $x, y \in \Omega$. $\mathcal{T}$ is called a *tree metric space*. Assume $V \subset \mathbb{R}^d$ and let $d_{\mathcal{T}}$ denote the tree metric on $\mathcal{T}$; we write $[x, y]$ for the *unique path* between $x, y \in \mathcal{T}$. Let $\omega$ be the unique Borel (length) measure on $\mathcal{T}$ satisfying $\omega([x, y]) = d_{\mathcal{T}}(x, y)$ for all $x, y \in \mathcal{T}$. For any $x \in \mathcal{T}$, the *subtree rooted at $x$* is defined as $\Lambda(x) = \{y \in \mathcal{T} : x \in [r, y]\}$. Figure 1 (left) provides a visual representation of tree metric spaces and their associated concepts.

**Optimal Transport on Tree Metric Spaces.** Let $\mathcal{P}(\mathcal{T})$ be the collection of all probability measures on $\mathcal{T}$ (i.e. total mass is equal to 1). Let $\mu, \nu \in \mathcal{P}(\mathcal{T})$, and $\mathcal{P}(\mu, \nu)$ be the set of $\pi$ coupling between $\mu$ and $\nu$. The 1-Wasserstein distance (W) [87] between $\mu, \nu$ is:

$$\mathbf{W}_{p, d_{\mathcal{T}}}(\mu, \nu) = \left( \inf_{\pi \in \mathcal{P}(\mu, \nu)} \int_{\mathcal{T} \times \mathcal{T}} d_{\mathcal{T}}(x, y)^p \, d\pi(x, y) \right)^{\frac{1}{p}}. \tag{1}$$

In the case of tree metrics and $p = 1$, the distance $\mathbf{W}_{1, d_{\mathcal{T}}}(\mu, \nu)$ admits a closed-form solution [46]:

$$\mathbf{W}_{1, d_{\mathcal{T}}}(\mu, \nu) = \int_{\mathcal{T}} |\mu(\Lambda(x)) - \nu(\Lambda(x))| \, \omega(dx). \tag{2}$$

For general $p > 1$, the distance $\mathbf{W}_{p, d_{\mathcal{T}}}(\mu, \nu)$ does not admit a closed-form expression. A natural generalization of Equation (2) leads to the Sobolev Transport (ST) [45], that is

$$\mathrm{ST}_p(\mu, \nu) = \left( \int_{\mathcal{T}} |\mu(\Lambda(x)) - \nu(\Lambda(x))|^p \, \omega(dx) \right)^{\frac{1}{p}}$$

$$\neq \left( \inf_{\pi \in \mathcal{P}(\mu, \nu)} \int_{\mathcal{T} \times \mathcal{T}} d_{\mathcal{T}}(x, y)^p \, d\pi(x, y) \right)^{\frac{1}{p}} = \mathbf{W}_{p, d_{\mathcal{T}}}(\mu, \nu). \tag{3}$$

Although $\mathrm{ST}_p$ differs from $\mathbf{W}_{p, d_{\mathcal{T}}}$, it still defines a valid metric on $\mathcal{P}(\mathcal{T})$. Due to this property, and for simplicity, we focus on the case $p = 1$, as the extension to general $p$ is analogous.

**Entropy Partial Transports on Tree Metric Spaces.** Let $\mathcal{M}(\mathcal{T})$ and $\mathcal{M}(\mathcal{T} \times \mathcal{T})$ denote the space of all nonnegative Borel measures on $\mathcal{T}$ and $\mathcal{T} \times \mathcal{T}$ respectively, with finite total mass. Given $\mu, \nu \in \mathcal{M}(\mathcal{T})$, define the set of admissible partial couplings as

$$\Pi_{\leq}(\mu, \nu) = \{ \gamma \in \mathcal{M}(\mathcal{T} \times \mathcal{T}) : \gamma_1 \leq \mu, \, \gamma_2 \leq \nu \}, \tag{4}$$

where $\gamma_1$ and $\gamma_2$ represent the marginals of $\gamma$ on the first and second marginals, respectively. For any $\gamma \in \Pi_{\leq}(\mu, \nu)$, let $f_1$ and $f_2$ be the Radon–Nikodym derivatives of $\gamma_1$ with respect to $\mu$ and $\gamma_2$ with respect to $\nu$, respectively. Let $w : \mathcal{T} \to [0, \infty)$ defined by $w(x) = a_1 \, d_{\mathcal{T}}(x, x_0) + a_0$ where $x_0 \in \mathcal{T}$, $a_1 \in [0, b]$, and $a_0 \in [0, \infty)$. We have $w$ is $b$-Lipschitz continuous. We use the entropy function $F : [0, \infty) \to (0, \infty)$ given by $F(s) = |s - 1|$. Letting $\bar{m} = \min\{\mu(\mathcal{T}), \nu(\mathcal{T})\}$, and fixing $m \in [0, \bar{m}]$, the Entropy Partial Transports (EPT) problem is formulated as

$$\mathcal{W}_m(\mu, \nu) = \inf_{\substack{\gamma \in \Pi_{\leq}(\mu, \nu) \\ \gamma(\mathcal{T} \times \mathcal{T}) = m}} \left[ \mathcal{F}_1(\gamma_1 \mid \mu) + \mathcal{F}_2(\gamma_2 \mid \nu) + b \int_{\mathcal{T} \times \mathcal{T}} d_{\mathcal{T}}(x, y) \, \gamma(dx, dy) \right], \tag{5}$$

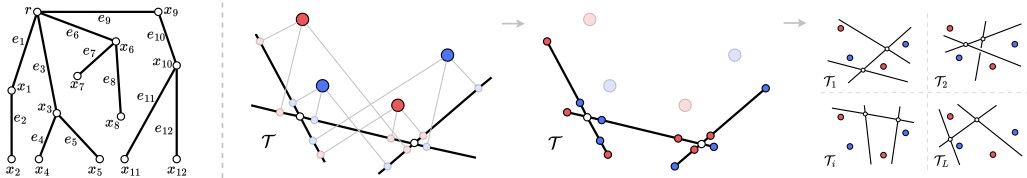

Figure 1: **An illustration of the construction of TSW.** *(Left) Tree Metric Space*. The tree is rooted at node $r$, with all other nodes denoted by $x_i$. Edges are denoted by $e_i$, each assigned a weight $w_e$. A probability distribution on the tree assigns mass to its nodes. The subtree $\Lambda(x)$ is defined as the collection of all points lying along the edges in the subtree rooted at $x$. For example, $\Lambda(r)$ includes the entire tree, $\Lambda(x_6)$ includes all points on edges $e_7$ and $e_8$, and $\Lambda(x_9)$ includes all points on edges $e_{10}$, $e_{11}$, and $e_{12}$. *(Right) An illustration of the Tree-Sliced Wasserstein computation*. Given two probability measures (depicted in red and blue), and a tree system $\mathcal{T}$, each measure is first pushed-forward onto the tree via the Radon transform, resulting in two measures supported on the tree structure. The Wasserstein distance between these tree-projected measures is then computed using Equation (2). The overall TSW distance is obtained by averaging the Wasserstein distances across a collection of such trees, typically approximated using a Monte Carlo sampling framework.

where the regularization terms are defined as the weighted relative entropies

$$\mathcal{F}_1(\gamma_1 \mid \mu) = \int_{\mathcal{T}} w(x)\, F(f_1(x))\, \mu(dx), \quad \mathcal{F}_2(\gamma_2 \mid \nu) = \int_{\mathcal{T}} w(x)\, F(f_2(x))\, \nu(dx). \quad (6)$$

To handle the mass constraint $\gamma(\mathcal{T} \times \mathcal{T}) = m$, a Lagrange multiplier $\lambda \in \mathbb{R}$ is introduced. Consider the relaxed objective

$$\mathrm{ET}_\lambda(\mu, \nu) = \inf_{\gamma \in \Pi_{\leq}(\mu, \nu)} \left[ \mathcal{F}_1(\gamma_1 \mid \mu) + \mathcal{F}_2(\gamma_2 \mid \nu) + b \int_{\mathcal{T} \times \mathcal{T}} (d_{\mathcal{T}}(x, y) - \lambda)\, \gamma(dx, dy) \right]. \quad (7)$$

According to a construction by Caffarelli and McCann [12], the problem (7) is equivalent to a balanced OT problem. Utilizing duality and regularization techniques as in [42], the objective (7) admits a closed-form solution for the $a$-regularized EPT:

For $a \in [0, b\lambda/2 + w(r)]$, it is given by:

$$\widetilde{\mathrm{ET}}_\lambda^a(\mu, \nu) = \int_{\mathcal{T}} |\mu(\Lambda(x)) - \nu(\Lambda(x))|\, \omega(dx)$$

$$- \frac{b\lambda}{2} [\mu(\mathcal{T}) + \nu(\mathcal{T})] + \left( w(r) + \frac{b\lambda}{2} - a \right) |\mu(\mathcal{T}) - \nu(\mathcal{T})|. \quad (8)$$

Define the corresponding regularized transport cost as

$$d_a(\mu, \nu) = \widetilde{\mathrm{ET}}_\lambda^a(\mu, \nu) + \frac{b\lambda}{2} [\mu(\mathcal{T}) + \nu(\mathcal{T})]. \quad (9)$$

The function $d_a$ defines a *metric* on $\mathcal{M}(\mathcal{T})$, making $(\mathcal{M}(\mathcal{T}), d_a)$ a complete metric space.

## 2.2 Tree-Sliced Wasserstein Distance on Euclidean Spaces

We adopt the setting of Tree-Sliced Wasserstein Distance on Systems of Lines as in [81, 80]. Figure 1 (right) presents the illustration relevant to the following discussion.

**Tree System.** A *line* in $\mathbb{R}^d$ is an element of $\mathbb{R}^d \times \mathbb{S}^{d-1}$, and a *system of $k$ lines* is an element of $(\mathbb{R}^d \times \mathbb{S}^{d-1})^k$. We denote a system of lines by $\mathcal{L}$, a line in $\mathcal{L}$ (also used as an index) by $l$, and the space of all such systems by $\mathbb{L}_k^d$. The *ground set* of $\mathcal{L}$ is defined as

$$\bar{\mathcal{L}} = \left\{ (x, l) \in \mathbb{R}^d \times \mathcal{L} : x = x_l + t_x \cdot \theta_l \text{ for some } t_x \in \mathbb{R} \right\},$$

where $x_l + t \cdot \theta_l$, with $t \in \mathbb{R}$, is the *parameterization of the line $l$*. For notational convenience, we index the lines as $l_1, \ldots, l_k$, where each line $l_i$ is defined by a source point $x_i \in \mathbb{R}^d$ and a direction vector $\theta_i \in \mathbb{S}^{d-1}$. A *tree system* is a system of lines endowed with an additional tree structure. To

highlight the presence of this structure, we denote the system by $\mathcal{T}$ rather than $\mathcal{L}$. The *space of tree systems*—that is, the collection of tree systems sharing a common tree structure—is denoted by $\mathbb{T}_k^d$, or simply $\mathbb{T}$. This space is equipped with a probability distribution $\sigma$, which is induced by a random sampling procedure over lines.

**Radon Transform on Tree Systems.** For $\mathcal{T} \in \mathbb{T}_k^d$, denote the *space of Lebesgue integrable functions on $\mathcal{T}$* as

$$L^1(\mathcal{T}) = \left\{ f \colon \bar{\mathcal{T}} \to \mathbb{R} : \|f\|_{\mathcal{T}} = \sum_{l \in \mathcal{T}} \int_{\mathbb{R}} |f(t_x, l)| \, dt_x < \infty \right\}.$$

Define the space $\mathcal{C}(\mathbb{R}^d \times \mathbb{T}_k^d, \Delta_{k-1})$ as the set of continuous maps from $\mathbb{R}^d \times \mathbb{T}_k^d$ to the $(k-1)$-dimensional standard simplex $\Delta_{k-1}$, named *splitting maps*. For $f \in L^1(\mathbb{R}^d)$, we define $\mathcal{R}_{\mathcal{T}}^{\alpha} f : \bar{\mathcal{T}} \to \mathbb{R}$ such that:

$$\mathcal{R}_{\mathcal{T}}^{\alpha} f(x, l) = \int_{\mathbb{R}^d} f(y) \cdot \alpha(y, \mathcal{T})_l \cdot \delta \left( t_x - \langle y - x_l, \theta_l \rangle \right) \, dy. \tag{10}$$

The function $\mathcal{R}_{\mathcal{T}}^{\alpha} f$ is in $L^1(\mathcal{T})$. The operator

$$\mathcal{R}^{\alpha} \colon L^1(\mathbb{R}^d) \longrightarrow \prod_{\mathcal{T} \in \mathbb{T}_k^d} L^1(\mathcal{T}), \qquad f \longmapsto (\mathcal{R}_{\mathcal{T}}^{\alpha} f)_{\mathcal{T} \in \mathbb{T}_k^d}, \tag{11}$$

is called the *Radon Transform on Tree Systems*. This operator is *injective*.

**Tree-Sliced Wasserstein Distance.** Tran et al. [81] proposed the *Tree-Sliced Wasserstein Distance on Systems of Lines* (TSW-SL), and later Tran et al. [80] proposed the *Distance-based Tree-Sliced Wasserstein Distance* (Db-TSW) which is the generalization of the former. Throughout this paper, we refer to both variants collectively as the Tree-Sliced Wasserstein (TSW) distance for brevity. This notion is distinct from the original TSW distance proposed in [46, 42, 34, 48], which was primarily developed for applications involving static-support measures, such as classification or topological data analysis. In contrast, the TSW-SL and Db-TSW formulations—cast as OT problems over *tree systems*—are specifically designed to handle applications with dynamic-support measures, as commonly encountered in generative modeling tasks. Given $\mu, \nu \in \mathcal{P}(\mathbb{R}^d)$ and $f_{\mu}, f_{\nu}$ are the probability density functions of $\mu, \nu$, respectively. The TSW distance between $\mu, \nu$ is defined by

$$\mathrm{TSW}(\mu, \nu) = \int_{\mathbb{T}} \mathrm{W}_{d_{\mathcal{T}}, 1} \left( \mathcal{R}_{\mathcal{T}}^{\alpha} f_{\mu}, \mathcal{R}_{\mathcal{T}}^{\alpha} f_{\nu} \right) \, d\sigma(\mathcal{T}), \tag{12}$$

TSW is a metric on $\mathcal{P}(\mathbb{R}^d)$. Leveraging the closed-form solution (2) and the Monte Carlo method, TSW in Equation (12) can be efficiently approximated by a closed-form expression.

## 3   Tree-Sliced Entropy Partial Transport

In this section, we formally introduce the Tree-Sliced Entropy Partial Transport framework and undertake a study of its theoretical properties and associated computational complexity.

### 3.1   Tree-Sliced Entropy Partial Transport

Start with a density function $f \in L^1(\mathbb{R}^d)$. The Radon Transform $\mathcal{R}^{\alpha}$ maps $f$ to a density function on a tree system while preserving its total mass, i.e.,

$$\|f\|_1 = \int_{\mathbb{R}^d} f(x) \, dx = \|\mathcal{R}_{\mathcal{T}}^{\alpha} f\|_{\mathcal{T}}, \quad \text{for all } \mathcal{T} \in \mathbb{T}. \tag{13}$$

A proof of this property is provided in Appendix D.6. For $\mu, \nu \in \mathcal{M}(\mathbb{R}^d)$, recall that $f_{\mu}$ and $f_{\nu}$ denote their respective density functions. Given a tree system $\mathcal{T} \in \mathbb{T}$ and a splitting map $\alpha \in \mathcal{C}(\mathbb{R}^d \times \mathbb{T}_k^d, \Delta_{k-1})$, the Radon Transform $\mathcal{R}^{\alpha}$, as defined in Equation (10), maps $f_{\mu}$ and $f_{\nu}$ to $\mathcal{R}_{\mathcal{T}}^{\alpha} f_{\mu}$ and $\mathcal{R}_{\mathcal{T}}^{\alpha} f_{\nu}$, respectively—both of which are density functions on $\mathcal{T}$. Denote the respective measures by $\mu_{\mathcal{T}}, \nu_{\mathcal{T}} \in \mathcal{M}(\mathcal{T})$. We then compute the regularized transport cost $d_a(\mu_{\mathcal{T}}, \nu_{\mathcal{T}})$ as in Equation (10). The proposed discrepancy is defined as the expectation of this quantity over the space of tree systems $\mathbb{T}$ with respect to the sampling distribution $\sigma$.

**Definition 3.1** (Tree-Sliced Entropy Partial Transport). The *Tree-Sliced Entropy Partial Transport*, denoted as PartialTSW, between $\mu$ and $\nu$ in $\mathcal{M}(\mathbb{R}^d)$ is defined by

$$\mathrm{PartialTSW}(\mu, \nu) := \int_{\mathbb{T}} d_a(\mu_{\mathcal{T}}, \nu_{\mathcal{T}}) d\sigma(\mathcal{T}). \tag{14}$$

**Remark 3.2.** It is worth noting that the value of Db-TSW is determined by a range of modeling choices, including the tree system space $\mathbb{T}$, the sampling distribution $\sigma$, the splitting map $\alpha \in \mathcal{C}(\mathbb{R}^d \times \mathbb{T}_k^d, \Delta_{k-1})$, and the regularization parameters $b$, $\lambda$, $w(\cdot)$, and $a$ involved in the definition of $d_a(\cdot, \cdot)$. These dependencies are excluded from the notation for simplicity and readability.

## 3.2 Properties of PartialTSW

Consider the space $\mathbb{R}^d$ equipped with the Euclidean norm $\|\cdot\|_2$. For any vector $v \in \mathbb{R}^d$, the translation by $v$ is the map $\mathbb{R}^d \to \mathbb{R}^d$ defined by $x \mapsto x + v$. The *translation group* $\mathrm{T}(d)$ consists of all such translations and is isomorphic to the additive group $\mathbb{R}^d$. The *orthogonal group* $\mathrm{O}(d)$ is the group consists of all $d \times d$ orthogonal matrices. The *Euclidean group* $\mathrm{E}(d)$ comprises all transformations of $\mathbb{R}^d$ that preserve pairwise Euclidean distances. Formally, $\mathrm{E}(d)$ is the semidirect product of $\mathrm{T}(d)$ and $\mathrm{O}(d)$. Each element $g \in \mathrm{E}(d)$ can be represented as a pair $g = (Q, v)$, where $Q \in \mathrm{O}(d)$ and $v \in \mathbb{R}^d$, and acts on $\mathbb{R}^d$ via $y \mapsto gy = Qy + v$. The canonical group action of $\mathrm{E}(d)$ on $\mathbb{R}^d$ naturally extends to the space of tree systems $\mathbb{T}_k^d$ through the rule

$$g\mathcal{T} = \{gl_i = (Qx_i + a, Q\theta_i)\}_{i=1}^k \in \mathbb{T}_k^d,$$

which preserves the underlying tree structure by construction. A splitting map $\alpha \in \mathcal{C}(\mathbb{R}^d \times \mathbb{T}_k^d, \Delta_{k-1})$ is said to be $\mathrm{E}(d)$-invariant if

$$\alpha(gx, g\mathcal{T}) = \alpha(x, \mathcal{T}), \quad \text{for all } x \in \mathbb{R}^d \text{ and } \mathcal{T} \in \mathbb{T}_k^d. \tag{15}$$

In the context of optimal transport theory, where a cost function defined on the ground space is lifted to a distance between measures, it is often desirable—particularly for measures on $\mathbb{R}^d$—that the resulting metric be equivariant under the action of the Euclidean group. Notably, both the 2-Wasserstein distance and the Sliced $p$-Wasserstein distance are known to exhibit $\mathrm{E}(d)$-invariance. Remarkably, in the case of PartialTSW, this invariance not only ensures that the discrepancy is $\mathrm{E}(d)$-invariant, but also guarantees that it is a valid metric on the space of measures $\mathcal{M}(\mathbb{R}^d)$.

**Theorem 3.3.** PartialTSW *is an* $\mathrm{E}(d)$*-invariant metric on* $\mathcal{M}(\mathbb{R}^d)$.

The proof for Theorem 3.3 is presented in Appendix §D.7.

**Remark 3.4.** As in [80], for the experiments in Section 4, we choose the splitting map $\alpha$ such that

$$\alpha(x, \mathcal{T}) = \mathrm{softmax}\left(\left\{\inf_{t \in \mathbb{R}} \|x - (x_i + t\theta_i)\|_2\right\}_{i=1}^k\right), \quad \text{for all } x \in \mathbb{R}^d \text{ and } \mathcal{T} \in \mathbb{T}, \tag{16}$$

which is $\mathrm{E}(d)$-invariant.

## 3.3 Computation of Tree-Sliced Entropy Partial Transport

Similar to UOT and PT, PartialTSW compares $\mu, \nu \in \mathcal{M}(\mathbb{R}^d)$ while offering a mechanism to softly enforce the fraction of mass to be transported. This is achieved by adjusting the total masses of their respective projections onto a tree system $\mathcal{T}$, denoted $\mu(\mathcal{T})$ and $\nu(\mathcal{T})$ (as in Equation (8) and (9)). Given that the Radon Transform $\mathcal{R}_{\mathcal{T}}^\alpha$ is mass-preserving, modifying the masses $\mu(\mathcal{T})$ and $\nu(\mathcal{T})$ directly controls the degree of partiality in the transport between the original measures $\mu$ and $\nu$. In practice, $\mu(\mathcal{T})$ is often normalized to a unit mass, with $\nu(\mathcal{T})$ then serving as a tunable hyperparameter to control this partiality.

A key application of PartialTSW is computing gradients of PartialTSW$(\mu, \nu)$ with respect to samples from $\mu$ and $\nu$, crucial for generative modeling and gradient flows. Here, samples typically have constant mass (e.g., $1/n$ where $n$ is the number of supports), rendering gradients with respect to total input masses $\mu(\mathbb{R}^d)$ and $\nu(\mathbb{R}^d)$ less meaningful. This simplifies hyperparameter selection: parameters $a$, $b$ and $\lambda$ in Equation (9) are not necessary. The mass $\nu(\mathcal{T})$ emerges as the only parameter for controlling the degree of partiality.

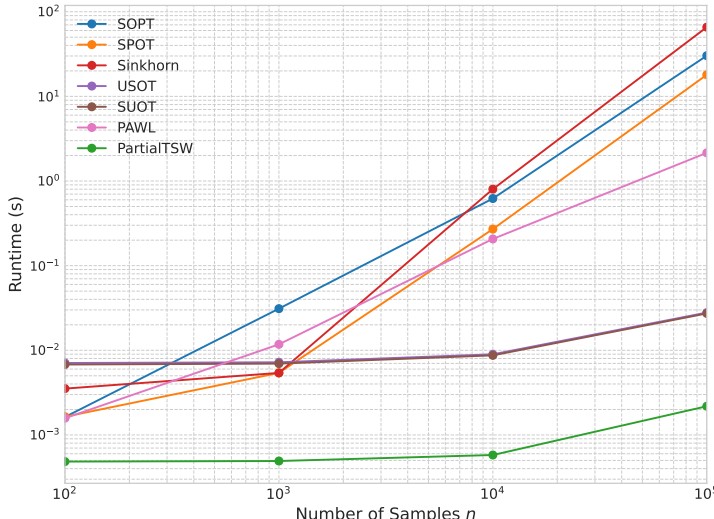

Figure 2: Runtime comparison for PartialTSW and PT/UOT solvers over $n$.

Computationally, the intractable integral in Equation (50) is approximated using Monte Carlo method:

$$\widehat{\text{PartialTSW}}(\mu, \nu) = \frac{1}{L} \sum_{i=1}^{L} d_a(\mu_{\mathcal{T}_i}, \nu_{\mathcal{T}_i}), \tag{17}$$

where $\{\mathcal{T}_i\}_{i=1}^{L}$ are tree systems independently sampled from $\sigma$ over $\mathbb{T}$. The theoretical complexity is $\mathcal{O}(Lkn\log n + Lkdn)$ (with $n$ samples, $k$ lines per tree, $d$ data dimension), identical to its balanced counterpart, TSW [80]. PartialTSW adds negligible computational overhead to TSW, mainly from adjusting the masses $\nu(\mathcal{T})$. Detailed algorithms and complexity analysis are provided in Appendices §E.1 and §E.2.

The log-linear scaling of PartialTSW with respect to the number of samples $n$ makes it significantly faster in practice than existing UOT and PT methods. Figure 2 illustrates this performance gap: for $n = 10^5$ samples, alternative approaches, such as the translation-invariant Sinkhorn [74] and PAWL [13], are approximately three orders of magnitude slower than PartialTSW. While USOT [7] and SUOT [7] exhibit similar scaling behavior due to their GPU-friendly implementations, PartialTSW maintains a speed advantage. Further details on computational efficiency are available in Appendix §E.6.

Since PartialTSW is approximated via Monte Carlo (MC) estimation, a crucial aspect is the stability and sample complexity of its estimator. Our distance is computed over $L$ sampled trees, and its approximation error is theoretically expected to decrease at a standard $\mathcal{O}(L^{-1/2})$ rate. We provide a detailed empirical analysis in Appendix §E.4 that verifies this convergence.

## 4 Experimental Results

This section details the empirical evaluation of PartialTSW against other methods in tasks requiring noise robustness for point cloud alignment, outlier rejection in generative modeling, and effective handling of class imbalance in image to image translation.

### 4.1 Noisy Point Cloud Gradient Flow

In this experiment, we aim to evaluate the robustness of PartialTSW compared to other optimal transport (OT) variants such as SW [10] and Db-TSW [80]. The source dataset $X$ consists of 10,000 points arranged in the shape of a dragon. The target dataset $Y$, also with 10,000 points, follows the shape of a bunny and is perturbed with 7% noise. Our objective is to determine whether PartialTSW can align with the target shape while ignoring noisy outliers. The clean data is taken from [2].

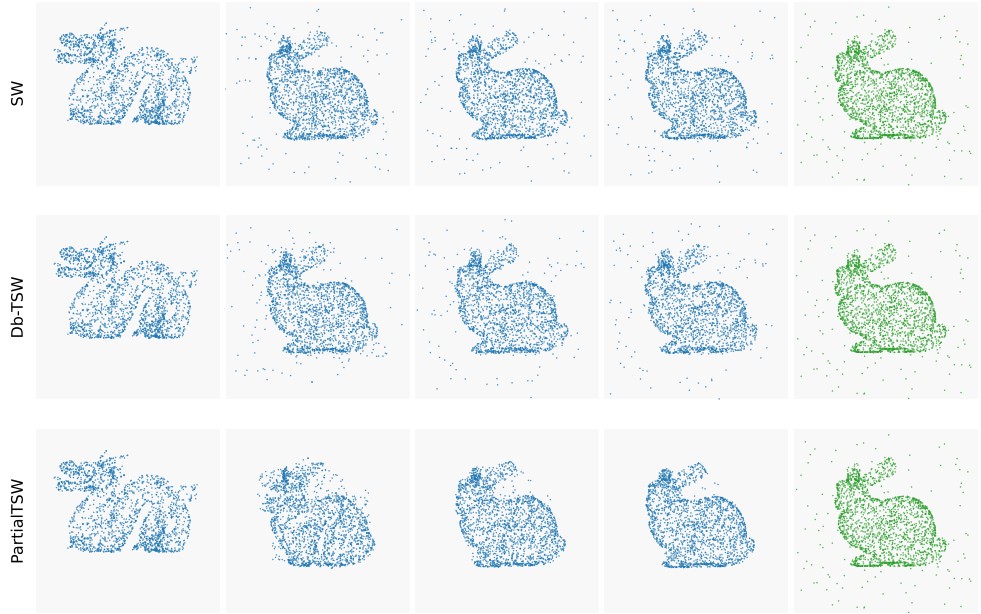

Figure 3: Visualization of point cloud gradient flows for SW, Db-TSW, and PartialTSW at steps 100, 200, and 300. The leftmost column is the source point cloud, and the rightmost column is the noise-perturbed target point cloud.

We apply gradient descent to the source points in order to minimize the distance $D(X, Y)$, where $D$ is either PartialTSW, Db-TSW, or SW. The results are illustrated in Figure 3. It is clear that OT methods such as SW and Db-TSW are affected by noisy data. In contrast, PartialTSW demonstrates greater robustness, producing smoother interpolations. We refer to Appendix §E.7 for further details.

## 4.2 Robust Generative Model

To further demonstrate its outlier robustness, PartialTSW is evaluated in a generative modeling experiment. First, an Autoencoder pre-trained on MNIST digits provides 2D latent representations for digit 0 (the target class) and digit 1 (the outlier class), scaled to approximately reside within $[-1, 1]^2$. Let $\mathcal{X}_0$ and $\mathcal{X}_1$ denote the true latent distributions for digits 0 and 1, respectively. The generator is subsequently trained using an observed dataset, $X_{\text{obs}}$, which is a mixture composed of 90% samples drawn from $\mathcal{X}_0$ and 10% samples (outliers) drawn from $\mathcal{X}_1$. This contaminated input data is illustrated in Figure 4a. A generator $G : \mathcal{N}(0, I_2) \rightarrow [-1, 1]^2$ is then trained by minimizing $D(G(Z), X_{\text{obs}})$, where $Z$ is a batch of noise samples and $D$ is the (Partial) Optimal Transport distance. The objective is for $G$ to learn to capture the target distribution $\mathcal{X}_0$ from the contaminated $X_{\text{obs}}$, effectively ignoring the outliers from $\mathcal{X}_1$. Further experimental details are available in Appendix §E.8.

Figure 4 and Table 1 summarize method performances. Standard OT methods (e.g., SW [10], Db-TSW [80]; Figure 4b-c) and several UOT/PT approaches (e.g., SOPT [2], SPOT [9], SUOT [7], USOT [7]; Figure 4d-e, g-h) struggled with the 10% outliers. Despite careful hyperparameter tuning (details in Appendix §E.8.2), these methods often produced mixed 0s and 1s or noisy outputs.

Conversely, PartialTSW demonstrates excellent robustness, achieving a 0.00% outlier rate (Table 1) by ignoring MNIST 1 outliers and generating only high-quality 0 digits (Figure 4j). This robust performance is complemented by strong sample diversity, stemming from its notably well-distributed latent space. Sinkhorn [74] and PAWL [13] also achieve 0.00% outlier rejection (Figure 4f,i); however, they exhibit less sample diversity, as suggested by their concentrated latent clusters.

PartialTSW also provides this robust and diverse generation with high computational efficiency. Its 55s runtime matches Db-TSW and is significantly faster than Sinkhorn (358s) and PAWL (88s). While SW is the fastest overall, it lacks the necessary outlier robustness.

Table 1: Quantitative comparison for robust generative modeling on MNIST, using target digit 0 and 10% 1 outliers. PartialTSW achieves perfect outlier rejection alongside a competitive runtime.

| Metric | OT | | Unbalanced OT / Partial OT | | | | | | Ours |
|---|---|---|---|---|---|---|---|---|---|
| | SW [10] | Db-TSW [80] | SOPT [2] | SPOT [9] | Sinkhorn [74] | SUOT [7] | USOT [7] | PAWL [13] | PartialTSW |
| Outliers (%) ↓ | 15.06 | 16.44 | 13.28 | 16.20 | **0.00** | 41.00 | 17.08 | **0.00** | **0.00** |
| Runtime (s) ↓ | **37** | 55 | 278 | 278 | 358 | 275 | 306 | 88 | 55 |

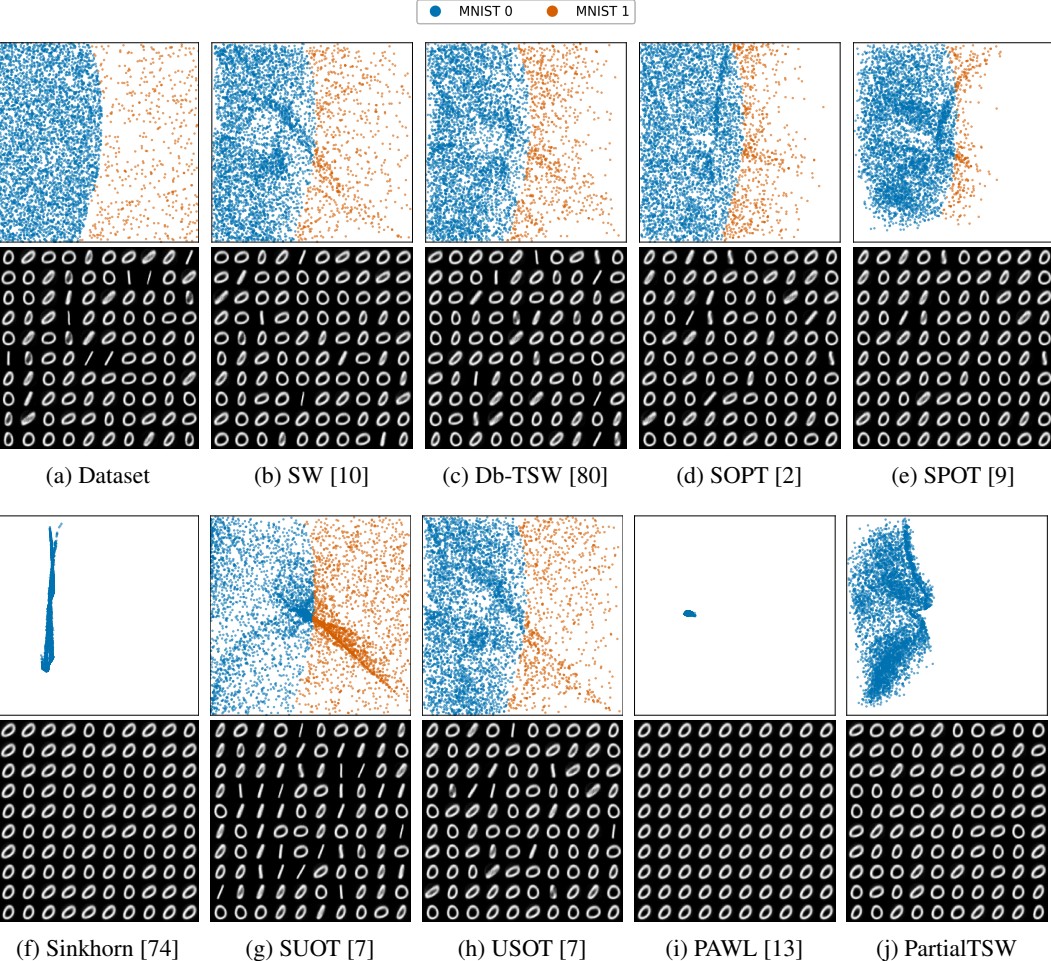

Figure 4: Qualitative comparison of outlier robustness in generative modeling. Methods learn to generate MNIST 0 (blue) from a dataset with 10% 1 outliers (orange). Subplots display generated latent distributions (top) and images (bottom). PartialTSW successfully ignores outliers and accurately learns the true latent distribution of the digit 0, leading to the diverse generated images.

Thus, PartialTSW uniquely combines strong outlier rejection, diverse sample generation, and high computational efficiency, making it highly effective for generative modeling in contaminated settings.

## 4.3 Imbalance Image to Image Translation

Another key attribute of PartialTSW is its ability to handle class imbalance. We demonstrate this capability in an image-to-image translation task, converting "Young" to "Adult" faces using the FFHQ dataset [37]. This dataset presents a significant imbalance, with 38K "Young" images and 10.5K "Adult" images. Our experimental setup follows the protocol from recent unbalanced optimal transport studies [22, 28]. Specifically, a pre-trained ALAE autoencoder [64] yields 512-dimensional

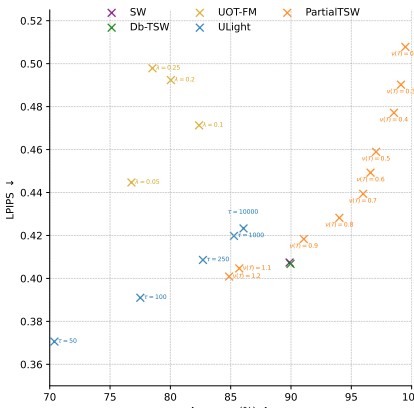

Figure 5: Visualizing the Accuracy-LPIPS trade-off in Image-to-Image translation.

Table 2: Quantitative Young-to-Adult translation results. PartialTSW (Ours) achieves a good balance of Accuracy ↑ and perceptual similarity (LPIPS ↓).

| Method | | Accuracy (%) ↑ | LPIPS ↓ |
|---|---|---|---|
| SW [10] | | $89.88 \pm 0.01$ | $0.4074 \pm 0.0002$ |
| Db-TSW [80] | | $89.96 \pm 0.01$ | $0.4068 \pm 0.0002$ |
| UOT-FM [22] | $\lambda = 0.05$ | $76.16 \pm 0.08$ | $0.4979 \pm 0.0001$ |
| | $\lambda = 0.1$ | $82.38 \pm 0.05$ | $0.4713 \pm 0.0001$ |
| | $\lambda = 0.2$ | $80.44 \pm 0.03$ | $0.4920 \pm 0.0003$ |
| ULightOT [28] | $\tau = 50$ | $70.39 \pm 0.09$ | $0.3706 \pm 0.0003$ |
| | $\tau = 250$ | $82.70 \pm 0.05$ | $0.4086 \pm 0.0002$ |
| | $\tau = 1000$ | $85.27 \pm 0.06$ | $0.4198 \pm 0.0001$ |
| | $\tau = 10000$ | $86.06 \pm 0.04$ | $0.4233 \pm 0.0001$ |
| PartialTSW | $\nu(\mathcal{T}) = 1.1$ | $85.71 \pm 0.03$ | $0.4047 \pm 0.0002$ |
| | $\nu(\mathcal{T}) = 0.9$ | $91.06 \pm 0.02$ | $0.4183 \pm 0.0002$ |
| | $\nu(\mathcal{T}) = 0.5$ | $97.05 \pm 0.03$ | $0.4590 \pm 0.0001$ |
| | $\nu(\mathcal{T}) = 0.3$ | $\mathbf{99.11 \pm 0.03}$ | $0.4902 \pm 0.0003$ |

latent image representations, where the translation is performed. All methods are evaluated on two criteria: (1) translation accuracy (whether images reconstructed from $M(X)$ classify as "Adult") and (2) perceptual similarity (LPIPS [90]) between original and translated reconstructions.

The mapping network $M$ is trained to translate latent samples $X$ from the "Young" domain to align with $Y$ from the "Adult" domain by minimizing a distance $D(M(X), Y)$, where $D$ represents our PartialTSW, SW [10], or Db-TSW [80]. We also compare against other recent procedures for this task: UOT-FM [22] and ULightOT [28]. These methods, along with PartialTSW, feature parameters to control their degree of regularization when handling data discrepancies: PartialTSW controls the transported mass via its parameter $\nu(\mathcal{T})$; UOT-FM [22] uses its parameter $\lambda$ to influence the regularization of marginal constraints; and ULightOT [28]'s parameter $\tau$ governs the extent of mass conservation. The adjustment of these parameters creates an Accuracy-LPIPS trade-off (see Figure 5).

Table 2 demonstrates the favorable balance of PartialTSW. Specifically, PartialTSW (with $\nu(\mathcal{T}) = 0.9$) achieves a higher accuracy of $91.06\%$, versus UOT-FM's (with $\lambda = 0.1$) $82.38\%$ and ULightOT's (with $\tau = 10000$) $86.06\%$. Concurrently, it achieves a lower LPIPS of $0.4183$ (indicating better perceptual similarity), compared to UOT-FM's $0.4713$ and ULightOT's $0.4233$. Furthermore, PartialTSW with $\nu(\mathcal{T}) = 0.3$ attains the highest translation accuracy of $\mathbf{99.11\%}$, significantly surpassing the accuracies of standard OT methods like SW ($89.88\%$) and Db-TSW ($89.96\%$).

These findings underscore PartialTSW's capability to handle significant class imbalances in image translation, offering a solution that effectively balances high target domain alignment with the preservation of perceptual similarity. Further experimental details are available in Appendix §E.9.

## 5    Conclusion

In this paper, we introduce Tree-Sliced Entropy Partial Transport (PartialTSW), a novel distance developed by integrating Entropy-Regularized Partial Transport for unbalanced measures on tree metric spaces with the Tree-Sliced Wasserstein (TSW) framework on tree systems. We investigate the theoretical properties of the proposed distance and establish that it constitutes a valid metric on the space of measures in Euclidean spaces. PartialTSW maintains the computational complexity of the balanced TSW distance, despite being tailored to handle unbalanced measures. Crucially, it is demonstrably faster than existing Unbalanced and Partial Optimal Transport approaches. Furthermore, comprehensive experiments demonstrate its effectiveness in addressing noise and imbalance in real-world data scenarios. A notable limitation of PartialTSW—and of existing TSW variants—is that it defines a distance without yielding explicit transport maps. An important direction for future work is therefore constructing optimal or partial transport plans within the tree-sliced setting.

## Acknowledgments and Disclosure of Funding

We thank the area chairs and anonymous reviewers for their comments. TL gratefully acknowledges the support of JSPS KAKENHI Grant number 23K11243, and Mitsui Knowledge Industry Co., Ltd. grant. TT acknowledges support from the Application Driven Mathematics Program funded and organized by the Vingroup Innovation Fund and VinBigData.

This research / project is supported by the National Research Foundation Singapore under the AI Singapore Programme (AISG Award No: AISG2-TC-2023-012-SGIL). This research / project is supported by the Ministry of Education, Singapore, under the Academic Research Fund Tier 1 (FY2023) (A-8002040-00-00, A-8002039-00-00). This research / project is also supported by the NUS Presidential Young Professorship Award (A-0009807-01-00) and the NUS Artificial Intelligence Institute–Seed Funding (A-8003062-00-00).

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

## Table of Notation

| | |
|---|---|
| $\mathbb{R}^d$ | $d$-dimensional Euclidean space |
| $\|\cdot\|_2$ | Euclidean norm |
| $\langle\cdot,\cdot\rangle$ | standard dot product |
| $\mathbb{S}^{d-1}$ | $(d-1)$-dimensional hypersphere |
| $\theta$ | unit vector |
| $\sqcup$ | disjoint union |
| $L^1(X)$ | space of Lebesgue integrable functions on $X$ |
| $\mathcal{P}(X)$ | space of probability measures on $X$ |
| $\mathcal{M}(X)$ | space of measures on $X$ |
| $\mu, \nu$ | measures |
| $\delta(\cdot)$ | 1-dimensional Dirac delta function |
| $\mathcal{U}(\mathbb{S}^{d-1})$ | uniform distribution on $\mathbb{S}^{d-1}$ |
| $\sharp$ | pushforward (measure) |
| $\mathcal{C}(X,Y)$ | space of continuous maps from $X$ to $Y$ |
| $\mathrm{d}(\cdot,\cdot)$ | metric in metric space |
| $d_\mathcal{T}(\cdot,\cdot)$ | tree metric |
| $\mathrm{T}(d)$ | translation group of order $d$ |
| $\mathrm{O}(d)$ | orthogonal group of order $d$ |
| $\mathrm{E}(d)$ | Euclidean group of order $d$ |
| $g$ | element of group |
| $\mathrm{W}_p$ | $p$-Wasserstein distance |
| $\mathrm{SW}_p$ | Sliced $p$-Wasserstein distance |
| $\Lambda$ | (rooted) subtree |
| $e$ | edge in graph |
| $w_e$ | weight of edge in graph |
| $l$ | line, index of line |
| $\mathcal{L}$ | system of lines, tree system |
| $\bar{\mathcal{L}}$ | ground set of system of lines, tree system |
| $\mathbb{L}_k^d$ | space of systems of $k$ lines in $\mathbb{R}^d$ |
| $\mathcal{T}$ | tree structure in system of lines |
| $L$ | number of tree systems |
| $k$ | number of lines in a system of lines or a tree system |
| $\mathcal{R}^\alpha$ | Radon Transform on Systems of Lines |
| $\Delta_{k-1}$ | $(k-1)$-dimensional standard simplex |
| $\alpha$ | splitting map |
| $\xi$ | tuning parameter in splitting maps |
| $\mathbb{T}$ | space of tree systems |
| $\sigma$ | distribution on space of tree systems |

# Appendix of "Tree-Sliced Entropy Partial Transport"

**Table of Contents**

# A   Background on Optimal Transport on Metric Spaces with Tree Metrics

Let $\mathcal{T} = (V, E)$ be a tree rooted at a node $r$, where each edge $e \in E$ is assigned a nonnegative length $w_e$. Here, $V$ denotes the set of nodes and $E$ the set of edges. For notational convenience, we use $\mathcal{T}$ to also refer to the union of all nodes and the continuous points along the edges. We now recall the formal definition of a tree metric:

**Definition A.1** (Tree metric [75, Section 7, p.145–182]). A metric $d : \Omega \times \Omega \to [0, \infty)$ is said to be a *tree metric* on a set $\Omega$ if there exists a tree $\mathcal{T}$ such that $\Omega \subset \mathcal{T}$ and, for all $x, y \in \Omega$, the distance $d(x, y)$ equals the length of the unique path in $\mathcal{T}$ connecting $x$ and $y$.

Suppose $V$ is a subset of a vector space, and let $d_{\mathcal{T}}(\cdot, \cdot)$ denote the tree metric defined on $\mathcal{T}$. We denote by $[x, y]$ the unique shortest path in $\mathcal{T}$ between any two points $x$ and $y$. Let $\omega$ be the unique Borel (length) measure on $\mathcal{T}$ satisfying $\omega([x, y]) = d_{\mathcal{T}}(x, y)$ for all $x, y \in \mathcal{T}$. For any $x \in \mathcal{T}$, we define the subtree rooted at $x$ by

$$\Lambda(x) := \{y \in \mathcal{T} : x \in [r, y]\}. \tag{18}$$

Let $\mathcal{P}(\mathcal{T})$ denote the set of all probability measures on $\mathcal{T}$, i.e., Borel measures with total mass equal to one. The following result provides a closed-form expression for the 1-Wasserstein distance on the tree metric space $\mathcal{T}$.

**Theorem A.2** (Optimal Transport on Tree Metric Spaces [46, Section 3, Proposition 1]). *For any $\mu, \nu \in \mathcal{P}(\mathcal{T})$, the 1-Wasserstein distance with respect to the tree metric $d_{\mathcal{T}}$ is given by*

$$\mathrm{W}_{1, d_{\mathcal{T}}}(\mu, \nu) = \int_{\mathcal{T}} |\mu(\Lambda(x)) - \nu(\Lambda(x))| \, \omega(dx). \tag{19}$$

# B   Background on Entropy Partial Transport on Metric Spaces with Tree Metrics

In this section, we revisit the Entropy Partial Transport (EPT) formulation introduced in [42] for completeness. All theoretical proofs are outlined in Appendix D.

We denote by $\mathcal{M}(\mathcal{T})$ the collection of all nonnegative Borel measures on $\mathcal{T}$ with finite total mass. Let $C(\mathcal{T})$ denote the space of continuous functions defined on $\mathcal{T}$, and let $L^{\infty}(\mathcal{T})$ denote the space of Borel measurable functions on $\mathcal{T}$ that are essentially bounded with respect to the measure $\omega$. The space $L^{\infty}(\mathcal{T})$ forms a Banach space when equipped with the norm

$$\|f\|_{L^{\infty}(\mathcal{T})} := \inf \left\{ \bar{a} \in \mathbb{R} : |f(x)| \le \bar{a} \text{ for } \omega\text{-almost every } x \in \mathcal{T} \right\}. \tag{20}$$

Let $\mathcal{M}(\mathcal{T} \times \mathcal{T})$ denote the space of all nonnegative Borel measures on $\mathcal{T} \times \mathcal{T}$ with finite total mass. Given $\mu, \nu \in \mathcal{M}(\mathcal{T})$, define the set of admissible partial couplings as

$$\Pi_{\le}(\mu, \nu) := \{\gamma \in \mathcal{M}(\mathcal{T} \times \mathcal{T}) : \gamma_1 \le \mu, \ \gamma_2 \le \nu\}, \tag{21}$$

where $\gamma_1$ and $\gamma_2$ represent the marginals of $\gamma$ on the first and second coordinates, respectively.

For any $\gamma \in \Pi_{\le}(\mu, \nu)$, let $f_1$ and $f_2$ be the Radon–Nikodym derivatives of $\gamma_1$ with respect to $\mu$ and $\gamma_2$ with respect to $\nu$, respectively. That is, $\gamma_1 = f_1 \mu$ and $\gamma_2 = f_2 \nu$, with the constraints $0 \le f_1 \le 1$ $\mu$-a.e. and $0 \le f_2 \le 1$ $\nu$-a.e.

Let $w : \mathcal{T} \to [0, \infty)$ be a $b$-Lipschitz continuous and nonnegative weight function, defined by

$$w(x) = a_1 \, d_{\mathcal{T}}(x, x_0) + a_0, \tag{22}$$

where $x_0 \in \mathcal{T}$, $a_1 \in [0, b]$, and $a_0 \in [0, \infty)$. Here, $d_{\mathcal{T}}(\cdot, \cdot)$ denotes the tree metric over $\mathcal{T}$. We use the entropy function $F : [0, \infty) \to (0, \infty)$ given by

$$F(s) = |s - 1|.$$

Letting $\bar{m} := \min\{\mu(\mathcal{T}), \nu(\mathcal{T})\}$, and fixing $m \in [0, \bar{m}]$, the EPT problem is formulated as

$$\mathcal{W}_m(\mu, \nu) := \inf_{\substack{\gamma \in \Pi_{\le}(\mu, \nu) \\ \gamma(\mathcal{T} \times \mathcal{T}) = m}} \left[ \mathcal{F}_1(\gamma_1 \mid \mu) + \mathcal{F}_2(\gamma_2 \mid \nu) + b \int_{\mathcal{T} \times \mathcal{T}} d_{\mathcal{T}}(x, y) \, \gamma(dx, dy) \right], \tag{23}$$

where the regularization terms are defined as the weighted relative entropies

$$\mathcal{F}_1(\gamma_1 \mid \mu) := \int_{\mathcal{T}} w(x)\, F(f_1(x))\, \mu(dx), \quad \mathcal{F}_2(\gamma_2 \mid \nu) := \int_{\mathcal{T}} w(x)\, F(f_2(x))\, \nu(dx). \tag{24}$$

To handle the mass constraint $\gamma(\mathcal{T} \times \mathcal{T}) = m$, we introduce a Lagrange multiplier $\lambda \in \mathbb{R}$ and instead consider the relaxed objective

$$\mathrm{ET}_\lambda(\mu, \nu) := \inf_{\gamma \in \Pi_\le(\mu,\nu)} \left[ \mathcal{F}_1(\gamma_1 \mid \mu) + \mathcal{F}_2(\gamma_2 \mid \nu) + b \int_{\mathcal{T} \times \mathcal{T}} (d_{\mathcal{T}}(x, y) - \lambda)\, \gamma(dx, dy) \right]. \tag{25}$$

We now expand the entropic terms and define

$$\mathcal{C}_\lambda(\gamma) := \int_{\mathcal{T}} w(x)\, \mu(dx) + \int_{\mathcal{T}} w(x)\, \nu(dx) - \int_{\mathcal{T}} w(x)\, \gamma_1(dx) - \int_{\mathcal{T}} w(x)\, \gamma_2(dx)$$
$$+ b \int_{\mathcal{T} \times \mathcal{T}} (d_{\mathcal{T}}(x, y) - \lambda)\, \gamma(dx, dy), \tag{26}$$

so that Equation (25) is equivalent to

$$\mathrm{ET}_\lambda(\mu, \nu) = \inf_{\gamma \in \Pi_\le(\mu,\nu)} \mathcal{C}_\lambda(\gamma). \tag{27}$$

As established in [42, Theorem 3.1, part i)], the solutions to Equation (23) and Equation (27) are related via the identity

$$\mathcal{W}_m(\mu, \nu) = \mathrm{ET}_\lambda(\mu, \nu) + \lambda b\, m. \tag{28}$$

Inspired by the construction proposed by Caffarelli and McCann [12], we recast the entropy-regularized partial transport problem in Equation (27) as a classical optimal transport (OT) problem between balanced measures. To achieve this, we augment the original domain $\mathcal{T}$ by introducing an auxiliary point $\hat{s} \notin \mathcal{T}$, and define the extended space $\hat{\mathcal{T}} := \mathcal{T} \cup \{\hat{s}\}$.

We then lift the unbalanced measures $\mu, \nu \in \mathcal{M}(\mathcal{T})$ to balanced counterparts supported on $\hat{\mathcal{T}}$:

$$\hat{\mu} := \mu + \nu(\mathcal{T})\, \delta_{\hat{s}}, \quad \hat{\nu} := \nu + \mu(\mathcal{T})\, \delta_{\hat{s}}, \tag{29}$$

where $\delta_{\hat{s}}$ denotes the Dirac measure at point $\hat{s}$. Next, we define a cost function $\hat{c} : \hat{\mathcal{T}} \times \hat{\mathcal{T}} \to \mathbb{R}$ that extends the original transport cost:

$$\hat{c}(x, y) := \begin{cases} b\,[d_{\mathcal{T}}(x, y) - \lambda] & \text{if } x, y \in \mathcal{T}, \\ w(x) & \text{if } x \in \mathcal{T} \text{ and } y = \hat{s}, \\ w(y) & \text{if } y \in \mathcal{T} \text{ and } x = \hat{s}, \\ 0 & \text{if } x = y = \hat{s}. \end{cases} \tag{30}$$

Using this extended cost, we formulate the balanced OT problem over $\hat{\mu}$ and $\hat{\nu}$:

$$\mathrm{KT}(\hat{\mu}, \hat{\nu}) := \inf_{\hat{\gamma} \in \Gamma(\hat{\mu},\hat{\nu})} \int_{\hat{\mathcal{T}} \times \hat{\mathcal{T}}} \hat{c}(x, y)\, \hat{\gamma}(dx, dy), \tag{31}$$

where the set of admissible transport plans $\Gamma(\hat{\mu}, \hat{\nu})$ is given by:

$$\Gamma(\hat{\mu}, \hat{\nu}) := \left\{ \hat{\gamma} \in \mathcal{M}(\hat{\mathcal{T}} \times \hat{\mathcal{T}}) : \hat{\gamma}(U \times \hat{\mathcal{T}}) = \hat{\mu}(U),\ \hat{\gamma}(\hat{\mathcal{T}} \times U) = \hat{\nu}(U),\ \forall \text{ Borel set } U \subset \hat{\mathcal{T}} \right\}. \tag{32}$$

The connection between the entropy-regularized partial transport formulation $\mathrm{ET}_\lambda$ in Equation (27) and the balanced optimal transport problem $\mathrm{KT}$ in Equation (31) is established by the following result.

**Proposition B.1** (Equivalence of $\mathrm{ET}_\lambda$ and KT). *Let $\mu, \nu \in \mathcal{M}(\mathcal{T})$. Then the two formulations coincide:*

$$\mathrm{ET}_\lambda(\mu, \nu) = \mathrm{KT}(\hat{\mu}, \hat{\nu}). \tag{33}$$

*Furthermore, the optimal plans $\gamma$ for the partial transport problem and $\hat{\gamma}$ for the balanced transport problem are related by:*

$$\hat{\gamma} = \gamma + (1 - f_1)\mu \otimes \delta_{\hat{s}} + \delta_{\hat{s}} \otimes (1 - f_2)\nu + \gamma(\mathcal{T} \times \mathcal{T})\, \delta_{(\hat{s}, \hat{s})}, \tag{34}$$

*where $f_1$ and $f_2$ are the Radon–Nikodym derivatives of the marginals of $\gamma$ with respect to $\mu$ and $\nu$, respectively.*

The detailed proof is provided in Appendix §D.1. Note that KT corresponds to a classical optimal transport problem defined between two balanced measures over the extended space $\hat{\mathcal{T}}$ and governed by the cost function $\hat{c}$. This allows us to invoke standard OT duality theory, such as [12, Corollary 2.6], to obtain a variational dual formulation for $\mathrm{ET}_\lambda$, as described below.

**Theorem B.2** (Dual Representation of $\mathrm{ET}_\lambda$). *The dual problem associated with the entropy-regularized partial transport functional* $\mathrm{ET}_\lambda(\mu, \nu)$ *is given by:*

$$\mathrm{ET}_\lambda(\mu, \nu) = \sup \left\{ \int_{\mathcal{T}} f \, (d\mu - d\nu) : f \in \mathbb{L} \right\} - \frac{b\lambda}{2} \left[ \mu(\mathcal{T}) + \nu(\mathcal{T}) \right], \tag{35}$$

*where the admissible function class* $\mathbb{L}$ *is defined as*

$$\mathbb{L} := \left\{ f \in C(\mathcal{T}) : -w - \frac{b\lambda}{2} \le f \le w + \frac{b\lambda}{2}, \quad |f(x) - f(y)| \le b \, d_{\mathcal{T}}(x, y) \text{ for all } x, y \in \mathcal{T} \right\}.$$

The proof of Theorem B.2 is deferred to Appendix §D.2. To obtain a tractable approximation of the dual problem, we introduce a regularization based on a restricted class of test functions. Let $r$ denote the root of the tree $\mathcal{T}$, and let $\omega$ be the associated length measure on $\mathcal{T}$. For a fixed parameter $a \in [0, \frac{b\lambda}{2} + w(r)]$, define the function class $\mathbb{L}_a$ to consist of all functions $f : \mathcal{T} \to \mathbb{R}$ of the form:

$$f(x) = s + \int_{[r, x]} g(y) \, \omega(dy), \tag{36}$$

where $s$ is a constant satisfying

$$s \in \left[ -w(r) - \frac{b\lambda}{2} + a, \ w(r) + \frac{b\lambda}{2} - a \right], \tag{37}$$

and $g \in L^\infty(\mathcal{T})$ is a bounded function with $\|g\|_{L^\infty(\mathcal{T})} \le b$.

The $a$-regularized entropy partial transport is then defined as:

$$\widetilde{\mathrm{ET}}_\lambda^a(\mu, \nu) := \sup_{f \in \mathbb{L}_a} \left\{ \int_{\mathcal{T}} f \, (d\mu - d\nu) \right\} - \frac{b\lambda}{2} \left[ \mu(\mathcal{T}) + \nu(\mathcal{T}) \right]. \tag{38}$$

This regularized formulation admits an explicit closed-form expression:

**Proposition B.3** (Closed-Form Solution for $\widetilde{\mathrm{ET}}_\lambda^a$). *For* $\mu, \nu \in \mathcal{M}(\mathcal{T})$, *we have:*

$$\widetilde{\mathrm{ET}}_\lambda^a(\mu, \nu) = \int_{\mathcal{T}} |\mu(\Lambda(x)) - \nu(\Lambda(x))| \, \omega(dx)$$

$$- \frac{b\lambda}{2} \left[ \mu(\mathcal{T}) + \nu(\mathcal{T}) \right] + \left( w(r) + \frac{b\lambda}{2} - a \right) |\mu(\mathcal{T}) - \nu(\mathcal{T})|. \tag{39}$$

The proof is provided in Appendix §D.3. We now compare the original entropy transport value $\mathrm{ET}_\lambda$ with its regularized approximation $\widetilde{\mathrm{ET}}_\lambda^a$:

**Proposition B.4** (Comparison Bounds between $\mathrm{ET}_\lambda$ and $\widetilde{\mathrm{ET}}_\lambda^a$). *The following inequalities hold:*

$$\mathrm{ET}_\lambda(\mu, \nu) \le \widetilde{\mathrm{ET}}_\lambda^0(\mu, \nu), \tag{40}$$

*and if the condition*

$$[4L_{\mathcal{T}} - \lambda]b \le 2w(r), \quad \text{where } L_{\mathcal{T}} := \max_{x \in \mathcal{T}} \omega([r, x]), \tag{41}$$

*is satisfied, then*

$$\widetilde{\mathrm{ET}}_\lambda^a(\mu, \nu) \le \mathrm{ET}_\lambda(\mu, \nu) \tag{42}$$

*for all $a$ such that* $2bL_{\mathcal{T}} \le a \le \frac{b\lambda}{2} + w(r)$.

The proof appears in Appendix §D.4. For $0 \le a < \frac{b\lambda}{2} + w(r)$, recall that the regularized transport cost is defined as:

$$d_a(\mu, \nu) := \widetilde{\mathrm{ET}}_\lambda^a(\mu, \nu) + \frac{b\lambda}{2} \left[ \mu(\mathcal{T}) + \nu(\mathcal{T}) \right]. \tag{43}$$

This cost function defines a genuine metric, as shown below:

**Proposition B.5** (Metric Structure of $d_a$). $(\mathcal{M}(\mathcal{T}), d_a)$ *is a complete metric space.*

The proof is presented in Appendix §D.5.

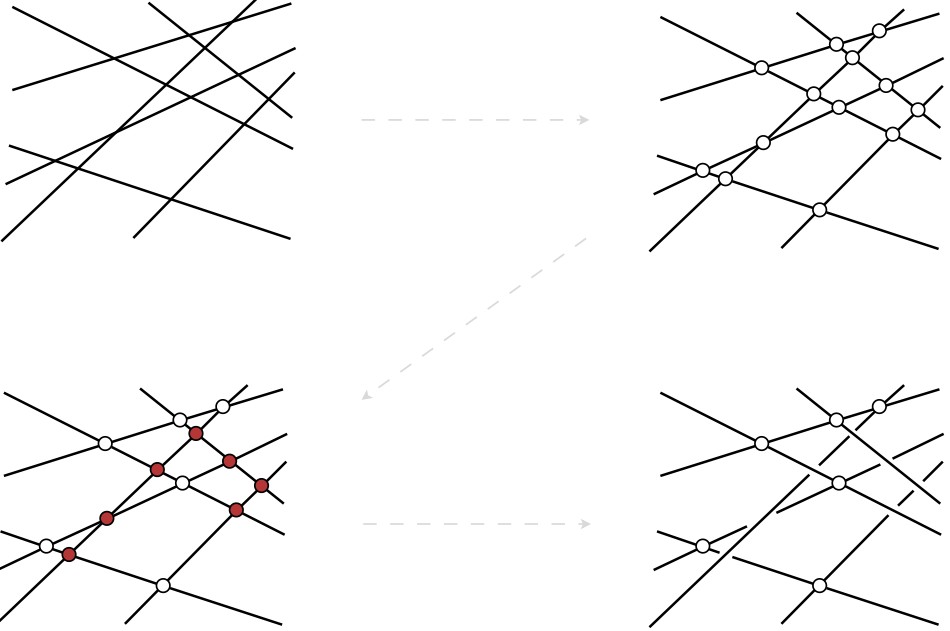

Figure 6: An illustration of the tree system construction is presented in the two-dimensional space $\mathbb{R}^2$, though the method readily generalizes to higher dimensions. The process starts with a collection of infinite lines arranged without any inherent structure. All pairwise intersections among these lines are identified (some of which may not be visible in the figure due to the lines' unbounded nature). A subset of intersections is selected and marked in red to indicate those to be discarded. The remaining intersections are retained to enforce a tree structure on the system—ensuring that any two points lying on the lines are connected by a unique path that passes only through the preserved intersections. These remaining intersections act as the essential nodes that define the tree topology. Once the red (discarded) intersections are removed, the resulting configuration forms the desired tree system.

## C  Background on Tree-Sliced Wasserstein Distance on Euclidean Spaces

This section reviews foundational concepts underlying the Tree-Sliced Wasserstein distance defined over Tree Systems. To ensure completeness, we revisit key definitions and theoretical formulations; detailed proofs and additional exposition are available in [81, 80].

### C.1  Tree System

**Line.**  A *line* in the Euclidean space $\mathbb{R}^d$ is specified by a tuple $(x, \theta) \in \mathbb{R}^d \times \mathbb{S}^{d-1}$ and is expressed parametrically as $x + t \cdot \theta$ for $t \in \mathbb{R}$. Throughout, we use $l = (x_l, \theta_l) \in \mathbb{R}^d \times \mathbb{S}^{d-1}$ to denote a line, where $x_l$ denotes the *source* point and $\theta_l$ the *direction* vector.

**System of lines.**  Given an integer $k \geq 1$, a *system of $k$ lines in $\mathbb{R}^d$* refers to a collection of $k$ such lines. The notation $(\mathbb{R}^d \times \mathbb{S}^{d-1})^k$ is abbreviated as $\mathbb{L}_k^d$, representing the *space of systems of $k$ lines in $\mathbb{R}^d$*. An element in this space, commonly denoted by $\mathcal{L}$, corresponds to a specific system of lines.

**Tree System.**  A system $\mathcal{L}$ is said to be *connected* if the union of all points lying on the constituent lines forms a connected subset of $\mathbb{R}^d$. By selectively *removing* certain intersection points between the lines, one can enforce a tree structure on $\mathcal{L}$—yielding a *tree system*—in which any two points are connected by a unique path. An illustration of this construction is provided in Figure 6.

**Remark C.1.**  The term *tree system* is used because there is a unique path between any two points, analogous to the definition of trees in graph theory.

Beginning with the remaining intersections, we employ the concepts of disjoint union and quotient topology [31] to construct a tree system by coherently gluing together multiple copies of $\mathbb{R}$. This

topological framework induces a natural metric, under which the resulting space satisfies the properties of a tree metric space.

**Sampling Procedure for Chain-Structured Tree Systems.** The space of tree systems is inherently rich and diverse, primarily due to the wide range of possible underlying tree topologies. [81] presents a general framework that accommodates arbitrary tree structures, while placing particular emphasis on a subclass of chain-like trees. The following describes the sampling procedure for generating tree systems with this chain-based architecture:

*Step* 1. Draw an initial point $x_1 \sim \mu_1$ and a direction $\theta_1 \sim \nu_1$, where $\mu_1$ is a probability measure on $\mathbb{R}^d$ and $\nu_1$ is a measure on the unit sphere $\mathbb{S}^{d-1}$.

*Step* $i$. For each subsequent node, sample $t_i \sim \mu_i$ and $\theta_i \sim \nu_i$, then compute $x_i = x_{i-1} + t_i \cdot \theta_{i-1}$. Here, $\mu_i$ is a distribution over $\mathbb{R}$ and $\nu_i$ over $\mathbb{S}^{d-1}$.

All distributions $\mu_i$ and $\nu_i$ are assumed to be mutually independent. Specifically, we consider the following choices: The initial position distribution $\mu_1$ is supported on a bounded subset of $\mathbb{R}^d$, such as the uniform distribution over the cube $[-1, 1]^d$, i.e., $\mathcal{U}([-1, 1]^d)$; For $i > 1$, each $\mu_i$ is defined on a bounded interval of $\mathbb{R}$—for example, $\mathcal{U}([-1, 1])$; Finally, each direction $\theta_i$ is drawn from a distribution over the unit sphere, e.g., the uniform distribution $\mathcal{U}(\mathbb{S}^{d-1})$. An example of such a tree system is illustrated in Figure 6.

**Remark C.2.** This generative process induces a probability measure $\sigma$ on the space $\mathbb{T}$ of all chain-structured tree systems produced via this construction.

## C.2 A Variant of Radon Transform for Systems of Lines

Let $L^1(\mathbb{R}^d)$ denote the space of Lebesgue integrable functions on $\mathbb{R}^d$, equipped with the standard $L^1$ norm $\| \cdot \|_1$. Consider a configuration of $k$ lines $\mathcal{L} \in \mathbb{L}_k^d$. A real-valued function $f$ defined on the domain $\bar{\mathcal{L}}$ consists of all points of $\mathcal{L}$, is said to be *integrable over the line system* if the following condition holds:

$$\|f\|_{\mathcal{L}} := \sum_{l \in \mathcal{L}} \int_{\mathbb{R}} |f(t_x, l)| \, dt_x < \infty. \tag{44}$$

The set of such functions is denoted by $L^1(\mathcal{L})$, representing the space of Lebesgue integrable functions over the line system $\mathcal{L}$. Recall the standard $(k-1)$-simplex:

$$\Delta_{k-1} = \left\{ (a_l)_{l \in \mathcal{L}} \in \mathbb{R}^k \mid a_l \geq 0, \sum_{l \in \mathcal{L}} a_l = 1 \right\}. \tag{45}$$

Define the space $\mathcal{C}(\mathbb{R}^d \times \mathbb{L}_k^d, \Delta_{k-1})$ as the set of continuous maps from $\mathbb{R}^d \times \mathbb{L}_k^d$ to $\Delta_{k-1}$, referred to as *splitting maps*. Given a line system $\mathcal{L} \in \mathbb{L}_k^d$ and a splitting map $\alpha \in \mathcal{C}(\mathbb{R}^d \times \mathbb{L}_k^d, \Delta_{k-1})$, we define a linear operator that projects a function $f \in L^1(\mathbb{R}^d)$ to the line system $\mathcal{L}$ as follows:

$$\mathcal{R}_{\mathcal{L}}^{\alpha} f \colon \bar{\mathcal{L}} \longrightarrow \mathbb{R} \tag{46}$$

$$(x, l) \longmapsto \int_{\mathbb{R}^d} f(y) \cdot \alpha(y, \mathcal{L})_l \cdot \delta \left( t_x - \langle y - x_l, \theta_l \rangle \right) \, dy, \tag{47}$$

where $\delta$ denotes the Dirac delta function in one dimension, and $(x_l, \theta_l)$ encodes the location and direction of line $l$. It can be shown that $\mathcal{R}_{\mathcal{L}}^{\alpha} f$ belongs to $L^1(\mathcal{L})$ for any $f \in L^1(\mathbb{R}^d)$, and furthermore satisfies the inequality

$$\|\mathcal{R}_{\mathcal{L}}^{\alpha} f\|_{\mathcal{L}} \leq \|f\|_1.$$

Hence, the operator $\mathcal{R}_{\mathcal{L}}^{\alpha} \colon L^1(\mathbb{R}^d) \to L^1(\mathcal{L})$ is well-defined. These properties are proven in [80]. Extending this to all line systems, we define the *Radon transform on Systems of Lines* as follows. For a fixed splitting map $\alpha \in \mathcal{C}(\mathbb{R}^d \times \mathbb{L}_k^d, \Delta_{k-1})$, define:

$$\mathcal{R}^{\alpha} \colon L^1(\mathbb{R}^d) \longrightarrow \prod_{\mathcal{L} \in \mathbb{L}_k^d} L^1(\mathcal{L}) \tag{48}$$

$$f \longmapsto (\mathcal{R}_{\mathcal{L}}^{\alpha} f)_{\mathcal{L} \in \mathbb{L}_k^d}. \tag{49}$$

If the splitting map $\alpha$ is invariant under the Euclidean group $\mathrm{E}(d)$—the group of all isometries of $\mathbb{R}^d$—then the operator $\mathcal{R}^{\alpha}$ is injective.

## C.3 Tree-Sliced Wasserstein Distance for Probability Measures on Euclidean Spaces

Let $\mu, \nu \in \mathcal{P}(\mathbb{R}^d)$ be probability measures. For a tree-structured system of lines $\mathcal{L} \in \mathbb{T}$ and an $\mathrm{E}(d)$-invariant splitting map $\alpha \in \mathcal{C}(\mathbb{R}^d \times \mathbb{L}_k^d, \Delta_{k-1})$, the transform $\mathcal{R}_{\mathcal{L}}^{\alpha}$ pushes forward $\mu$ and $\nu$ to corresponding measures $\mathcal{R}_{\mathcal{L}}^{\alpha}\mu$ and $\mathcal{R}_{\mathcal{L}}^{\alpha}\nu$ on $\mathcal{L}$. Since each $\mathcal{L} \in \mathbb{T}$ is equipped with a tree metric $d_{\mathcal{L}}$, the 1-Wasserstein distance $\mathrm{W}_{d_{\mathcal{L}},1}$ between the transformed measures can be computed. This leads to the following definition of the *Distance-based Tree-Sliced Wasserstein* (Db-TSW) [80] distance:

$$\mathrm{Db\text{-}TSW}(\mu, \nu) := \int_{\mathbb{T}} \mathrm{W}_{d_{\mathcal{L}},1} \left( \mathcal{R}_{\mathcal{L}}^{\alpha}\mu, \mathcal{R}_{\mathcal{L}}^{\alpha}\nu \right) \, d\sigma(\mathcal{L}), \tag{50}$$

where $\sigma$ is a probability measure over the space of tree systems $\mathbb{T}$. It is important to note that the value of Db-TSW depends on the choice of the tree system space $\mathbb{T}$, the sampling distribution $\sigma$, and the specific $\mathrm{E}(d)$-invariant splitting map $\alpha$, although these dependencies are omitted from the notation for brevity. The resulting Db-TSW defines an $\mathrm{E}(d)$-invariant metric on $\mathcal{P}(\mathbb{R}^d)$.

**Remark C.3.** As established in [80], if the tree systems consist solely of a single line, the Db-TSW distance reduces exactly to the classical Sliced Wasserstein (SW) distance on $\mathbb{R}^d$.

**Constructing $\mathrm{E}(d)$-Invariant Splitting Maps.** The Euclidean group $\mathrm{E}(d)$ consists of all transformations of $\mathbb{R}^d$ that preserve pairwise Euclidean distances. As such, this invariance extends not only to distances between points but also to the shortest distance from a point to a line. Given a point $x \in \mathbb{R}^d$ and a system of lines $\mathcal{L} \in \mathbb{L}_k^d$, define the distance from $x$ to a line $l \in \mathcal{L}$ by:

$$d(x, \mathcal{L})_l = \inf_{y \in l} \|x - y\|_2. \tag{51}$$

This quantity is preserved under the action of $\mathrm{E}(d)$, meaning that any function constructed solely from the collection $\{d(x, \mathcal{L})_l\}_{l \in \mathcal{L}}$ will inherit $\mathrm{E}(d)$-invariance.

Based on this observation, invariant splitting maps is constructed by applying a post-processing function $\beta \colon \mathbb{R}^k \to \Delta_{k-1}$ to the vector of distances. The resulting splitting map,

$$\alpha(x, \mathcal{L})_l = \beta \left( \{d(x, \mathcal{L})_l\}_{l \in \mathcal{L}} \right), \tag{52}$$

is guaranteed to be $\mathrm{E}(d)$-invariant for any choice of continuous $\beta$. Empirically, effective performance in applications is achieved when $\beta$ is taken to be the softmax function with a tunable scaling parameter $\xi > 0$. This yields the practical definition:

$$\alpha(x, \mathcal{L})_l = \mathrm{softmax} \left( \{\xi \cdot d(x, \mathcal{L})_l\}_{l \in \mathcal{L}} \right), \tag{53}$$

which distributes weights across lines in $\mathcal{L}$ according to their proximity to $x$, while respecting the geometric symmetries of the Euclidean space.

**Remark C.4.** The concepts of equivariance and invariance are widely employed in machine learning to ensure model robustness under transformations that preserve the semantic or structural properties of the input. Such principles are foundational in the design of architectures that respect inherent symmetries within data. Applications of equivariant models span various domains, including equivariant graph neural networks [71, 19, 82], equivariant metanetworks [86, 78, 88, 36], parameter symmetry [30, 26, 85], and optimization [91], among others.

# D  Theoretical Proofs

To ensure completeness, we provide full derivations of the result, closely following the methodology of [42].

## D.1  Proof for Proposition B.1

To ensure completeness, we provide full derivations of the result, closely following the methodology of [42].

*Proof.* We begin by proving the inequality $\mathrm{KT}(\hat{\mu}, \hat{\nu}) \leq \mathrm{ET}_{\lambda}(\mu, \nu)$. Let $\gamma \in \Pi_{\leq}(\mu, \nu)$ be any admissible partial transport plan, and define $\hat{\gamma}$ according to the expression in Equation (34). By

construction, $\hat{\gamma} \in \Gamma(\hat{\mu}, \hat{\nu})$. Then, evaluating the cost of $\hat{\gamma}$ under the extended transport objective yields:

$$
\begin{aligned}
\mathrm{KT}(\hat{\mu}, \hat{\nu}) &\leq \int_{\hat{\mathcal{T}} \times \hat{\mathcal{T}}} \hat{c}(x, y)\, \hat{\gamma}(dx, dy) \\
&= b \int_{\mathcal{T} \times \mathcal{T}} [d_{\mathcal{T}}(x, y) - \lambda]\, \gamma(dx, dy) \\
&\qquad + \int_{\mathcal{T}} w(x)\, [1 - f_1(x)]\, \mu(dx) + \int_{\mathcal{T}} w(x)\, [1 - f_2(x)]\, \nu(dx). \quad (54)
\end{aligned}
$$

Taking the infimum over all $\gamma \in \Pi_{\leq}(\mu, \nu)$ on the right-hand side implies:

$$
\mathrm{KT}(\hat{\mu}, \hat{\nu}) \leq \mathrm{ET}_{\lambda}(\mu, \nu). \tag{55}
$$

We now establish the reverse inequality, i.e., $\mathrm{KT}(\hat{\mu}, \hat{\nu}) \geq \mathrm{ET}_{\lambda}(\mu, \nu)$. Let $\hat{\gamma} \in \Gamma(\hat{\mu}, \hat{\nu})$ be any feasible coupling in the balanced OT problem, and let $\gamma$ be its restriction to $\mathcal{T} \times \mathcal{T}$. Then, by [42, Lemma 3.2], we have $\gamma \in \Pi_{\leq}(\mu, \nu)$ and the decomposition in Equation (34) holds. We now compute the total cost of $\hat{\gamma}$ under $\hat{c}$:

$$
\begin{aligned}
\int_{\hat{\mathcal{T}} \times \hat{\mathcal{T}}} \hat{c}(x, y)\, \hat{\gamma}(dx, dy) &= b \int_{\mathcal{T} \times \mathcal{T}} [d_{\mathcal{T}}(x, y) - \lambda]\, \gamma(dx, dy) \\
&\qquad + \int_{\mathcal{T}} w(x)\, [1 - f_1(x)]\, \mu(dx) + \int_{\mathcal{T}} w(x)\, [1 - f_2(x)]\, \nu(dx) \\
&\geq \mathrm{ET}_{\lambda}(\mu, \nu). \quad (56)
\end{aligned}
$$

Taking the infimum over all admissible $\hat{\gamma} \in \Gamma(\hat{\mu}, \hat{\nu})$ yields the desired inequality:

$$
\mathrm{KT}(\hat{\mu}, \hat{\nu}) \geq \mathrm{ET}_{\lambda}(\mu, \nu). \tag{57}
$$

Combining both bounds, we conclude the equivalence:

$$
\mathrm{KT}(\hat{\mu}, \hat{\nu}) = \mathrm{ET}_{\lambda}(\mu, \nu). \tag{58}
$$

The correspondence between optimal couplings $\gamma$ and $\hat{\gamma}$ follows directly from the construction and identities established above. $\qquad\square$

## D.2 Proof for Theorem B.2

To ensure completeness, we provide full derivations of the result, closely following the methodology of [42].

*Proof.* We begin by establishing the intermediate result:

$$
\mathrm{ET}_{\lambda}(\mu, \nu) = \sup_{(u, v) \in \mathcal{K}} \left[ \int_{\mathcal{T}} u(x)\, \mu(dx) + \int_{\mathcal{T}} v(x)\, \nu(dx) \right], \tag{59}
$$

where the admissible set $\mathcal{K}$ is defined as

$$
\mathcal{K} := \Big\{ (u, v) \in L^1(\mu) \times L^1(\nu) \ \Big|\ u(x) \leq w(x), \quad \forall x \in \mathcal{T},
$$
$$
- b\lambda + \inf_{x \in \mathcal{T}} [b\, d_{\mathcal{T}}(x, y) - w(x)] \leq v(y) \leq w(y), \quad \forall y \in \mathcal{T},
$$
$$
u(x) + v(y) \leq b[d_{\mathcal{T}}(x, y) - \lambda], \quad \forall x, y \in \mathcal{T} \Big\}.
$$

This identity follows from the dual representation of $\mathrm{KT}(\hat{\mu}, \hat{\nu})$ via Proposition B.1 and [12, Corollary 2.6], which yields:

$$
\begin{aligned}
\mathrm{ET}_{\lambda}(\mu, \nu) &= \sup_{\substack{\hat{u} \in L^1(\hat{\mu}),\ \hat{v} \in L^1(\hat{\nu}) \\ \hat{u}(x) + \hat{v}(y) \leq \hat{c}(x, y)}} \left[ \int_{\hat{\mathcal{T}}} \hat{u}(x)\, \hat{\mu}(dx) + \int_{\hat{\mathcal{T}}} \hat{v}(y)\, \hat{\nu}(dy) \right] \\
&=: I. \quad (60)
\end{aligned}
$$

We aim to show that this supremum $I$ coincides with

$$J := \sup_{(u,v) \in \mathcal{K}} \left[ \int_{\mathcal{T}} u(x) \, \mu(dx) + \int_{\mathcal{T}} v(x) \, \nu(dx) \right]. \tag{61}$$

To show $I \geq J$, let $(u,v) \in \mathcal{K}$. Extend these functions to $\hat{\mathcal{T}}$ by setting:

$$\hat{u}(x) := \begin{cases} u(x) & \text{if } x \in \mathcal{T}, \\ 0 & \text{if } x = \hat{s}, \end{cases} \quad \hat{v}(x) := \begin{cases} v(x) & \text{if } x \in \mathcal{T}, \\ 0 & \text{if } x = \hat{s}. \end{cases}$$

Since $(u,v) \in \mathcal{K}$, it follows directly from the definition of $\hat{c}$ that $\hat{u}(x) + \hat{v}(y) \leq \hat{c}(x,y)$ for all $x, y \in \hat{\mathcal{T}}$. Consequently:

$$\begin{aligned} I &\geq \int_{\hat{\mathcal{T}}} \hat{u}(x) \, \hat{\mu}(dx) + \int_{\hat{\mathcal{T}}} \hat{v}(x) \, \hat{\nu}(dx) \\ &= \int_{\mathcal{T}} u(x) \, \mu(dx) + \int_{\mathcal{T}} v(x) \, \nu(dx), \end{aligned} \tag{62}$$

which implies $I \geq J$.

To prove the reverse inequality $I \leq J$, let $(\hat{u}, \hat{v})$ be a maximizer for $I$. Without loss of generality, we can normalize $\hat{u}(\hat{s}) = 0$ by observing that replacing $(\hat{u}, \hat{v})$ with $(\hat{u} - \hat{u}(\hat{s}), \hat{v} + \hat{u}(\hat{s}))$ preserves admissibility and the objective value. Moreover, define:

$$v(y) := \inf_{x \in \hat{\mathcal{T}}} [\hat{c}(x,y) - \hat{u}(x)] \quad \forall y \in \hat{\mathcal{T}}. \tag{63}$$

Then $\hat{v}(y) \leq v(y)$, and $(\hat{u}, v)$ remains admissible and achieves the same supremum, so we may further assume $\hat{v}(y) = \inf_x [\hat{c}(x,y) - \hat{u}(x)]$ and $\hat{u}(\hat{s}) = 0$. In particular,

$$\hat{v}(\hat{s}) = \inf_{x \in \hat{\mathcal{T}}} [\hat{c}(x, \hat{s}) - \hat{u}(x)]. \tag{64}$$

To proceed, we define $w(\hat{s}) := 0$ and consider two cases based on the structure of $\hat{u}$ and $\hat{v}$.

*Case 1.* Suppose that

$$\inf_{x \in \hat{\mathcal{T}}} [w(x) - \hat{u}(x)] \geq 0. \tag{65}$$

In this case, observe that $\hat{u}(\hat{s}) = 0$ by assumption. Since

$$\hat{c}(\hat{s}, \hat{s}) - \hat{u}(\hat{s}) = 0 \quad \text{and} \quad \inf_{x \in \mathcal{T}} [\hat{c}(x, \hat{s}) - \hat{u}(x)] = \inf_{x \in \mathcal{T}} [w(x) - \hat{u}(x)] \geq 0,$$

we conclude that

$$\hat{v}(\hat{s}) = \inf_{x \in \hat{\mathcal{T}}} [\hat{c}(x, \hat{s}) - \hat{u}(x)] = 0. \tag{66}$$

Next, for all $y \in \hat{\mathcal{T}}$, we bound $\hat{v}(y)$ from above:

$$\hat{v}(y) = \inf_{x \in \hat{\mathcal{T}}} [\hat{c}(x,y) - \hat{u}(x)] \leq \hat{c}(\hat{s}, y) - \hat{u}(\hat{s}) = w(y), \tag{67}$$

where we have used $\hat{u}(\hat{s}) = 0$.

To lower-bound $\hat{v}(y)$ for $y \in \mathcal{T}$, note that $\hat{u}(x) \leq w(x)$ for all $x \in \mathcal{T}$, and $w(\hat{s}) = 0$. Therefore,

$$\begin{aligned} \hat{v}(y) &= \inf_{x \in \hat{\mathcal{T}}} [\hat{c}(x,y) - \hat{u}(x)] \\ &\geq \inf_{x \in \hat{\mathcal{T}}} [\hat{c}(x,y) - w(x)] \\ &= \inf_{x \in \mathcal{T}} [b(d_{\mathcal{T}}(x,y) - \lambda) - w(x)] \\ &= -b\lambda + \inf_{x \in \mathcal{T}} [b \, d_{\mathcal{T}}(x,y) - w(x)]. \end{aligned} \tag{68}$$

Combining both bounds, we find that $\hat{v}(y)$ satisfies all constraints in the definition of $\mathcal{K}$, and $\hat{u} \leq w$ holds by assumption. Hence, $(\hat{u}, \hat{v}) \in \mathcal{K}$. We now compute the dual objective:

$$
\begin{aligned}
I &= \int_{\hat{\mathcal{T}}} \hat{u}(x)\,\hat{\mu}(dx) + \int_{\hat{\mathcal{T}}} \hat{v}(x)\,\hat{\nu}(dx) \\
&= \int_{\mathcal{T}} \hat{u}(x)\,\mu(dx) + \int_{\mathcal{T}} \hat{v}(x)\,\nu(dx) + \hat{v}(\hat{s})\,\mu(\mathcal{T}) \\
&= \int_{\mathcal{T}} \hat{u}(x)\,\mu(dx) + \int_{\mathcal{T}} \hat{v}(x)\,\nu(dx) \tag{69} \\
&\leq J. \tag{70}
\end{aligned}
$$

Thus, under this case, the supremum $I$ is bounded above by $J$, completing the proof for Case 1.

*Case 2.* Suppose now that

$$
\inf_{x \in \hat{\mathcal{T}}} \left[ w(x) - \hat{u}(x) \right] < 0. \tag{71}
$$

As in Case 1, we deduce that

$$
\hat{v}(\hat{s}) = \inf_{x \in \mathcal{T}} \left[ w(x) - \hat{u}(x) \right] < 0, \tag{72}
$$

and the dual objective becomes

$$
I = \int_{\mathcal{T}} \hat{u}(x)\,\mu(dx) + \int_{\mathcal{T}} \hat{v}(x)\,\nu(dx) + \mu(\mathcal{T}) \cdot \inf_{\mathcal{T}} \left[ w - \hat{u} \right]. \tag{73}
$$

Define a truncated version of $\hat{u}$ by setting:

$$
\tilde{u}(x) := \min\{\hat{u}(x), w(x)\}. \tag{74}
$$

This ensures that $\tilde{u}(x) \leq w(x)$ and, since $\hat{u}(\hat{s}) = 0$, we also have $\tilde{u}(\hat{s}) = 0$. Furthermore, for all $x, y \in \hat{\mathcal{T}}$,

$$
\tilde{u}(x) + \hat{v}(y) \leq \hat{c}(x, y), \tag{75}
$$

due to the pointwise minimum structure of $\tilde{u}$ and the feasibility of $(\hat{u}, \hat{v})$.

Since $\inf_{x \in \mathcal{T}} \left[ w(x) - \hat{u}(x) \right] < 0$, there exists $x_0 \in \mathcal{T}$ such that $\hat{u}(x_0) > w(x_0)$. Thus, at $x_0$, we have $\tilde{u}(x_0) = w(x_0)$ and therefore

$$
\inf_{\mathcal{T}} \left[ w - \tilde{u} \right] \leq 0. \tag{76}
$$

On the other hand, since $\tilde{u}(x) \leq w(x)$ everywhere, it follows that

$$
\inf_{\mathcal{T}} \left[ w - \tilde{u} \right] = 0. \tag{77}
$$

We now rewrite the first two terms in Equation (73) as:

$$
\begin{aligned}
\int_{\mathcal{T}} \hat{u}(x)\,\mu(dx) + \mu(\mathcal{T}) \cdot \inf_{\mathcal{T}}[w - \hat{u}] = \int_{\mathcal{T}} \tilde{u}(x)\,\mu(dx) + \int_{\{x : \hat{u}(x) > w(x)\}} &[\hat{u}(x) - w(x)]\,\mu(dx) \\
&+ \mu(\mathcal{T}) \cdot \inf_{\mathcal{T}}[w - \hat{u}] \\
\leq \int_{\mathcal{T}} \tilde{u}(x)\,\mu(dx). &\tag{78}
\end{aligned}
$$

Substituting this into Equation (73), we obtain the upper bound:

$$
I \leq \int_{\mathcal{T}} \tilde{u}(x)\,\mu(dx) + \int_{\mathcal{T}} \hat{v}(x)\,\nu(dx). \tag{79}
$$

We now define a new function $\tilde{v} : \mathcal{T} \to \mathbb{R}$ by

$$
\tilde{v}(y) := \inf_{x \in \hat{\mathcal{T}}} \left[ \hat{c}(x, y) - \tilde{u}(x) \right]. \tag{80}
$$

By construction, $\tilde{v}(y) \geq \hat{v}(y)$ and for all $y \in \mathcal{T}$,

$$\tilde{v}(y) \leq \hat{c}(\hat{s}, y) - \tilde{u}(\hat{s}) = w(y). \tag{81}$$

Furthermore, using $\tilde{u}(x) \leq w(x)$ and the form of $\hat{c}$, we obtain a lower bound:

$$\begin{aligned}
\tilde{v}(y) &= \inf_{x \in \hat{\mathcal{T}}} [\hat{c}(x, y) - \tilde{u}(x)] \\
&\geq \inf_{x \in \mathcal{T}} [b(d_{\mathcal{T}}(x, y) - \lambda) - w(x)] \\
&= -b\lambda + \inf_{x \in \mathcal{T}} [b\, d_{\mathcal{T}}(x, y) - w(x)].
\end{aligned} \tag{82}$$

Combining these, we find that $(\tilde{u}, \tilde{v}) \in \mathcal{K}$. Hence,

$$I \leq \int_{\mathcal{T}} \tilde{u}(x)\, \mu(dx) + \int_{\mathcal{T}} \tilde{v}(x)\, \nu(dx) \leq J. \tag{83}$$

This completes the analysis for Case 2 and thus confirms the desired equality:

$$\mathrm{ET}_\lambda(\mu, \nu) = \sup_{(u,v) \in \mathcal{K}} \left[ \int_{\mathcal{T}} u(x)\, \mu(dx) + \int_{\mathcal{T}} v(x)\, \nu(dx) \right], \tag{84}$$

where

$$\mathcal{K} := \left\{ (u, v) : u \leq w, \quad -b\lambda + \inf_{x \in \mathcal{T}} [b\, d_{\mathcal{T}}(x, y) - w(x)] \leq v(y) \leq w(y), \right.$$
$$\left. u(x) + v(y) \leq b(d_{\mathcal{T}}(x, y) - \lambda) \right\}. \tag{85}$$

We are now ready to complete the proof of the theorem. Since the weight function $w$ is $b$-Lipschitz, it satisfies the following inequality for all $x \in \mathcal{T}$:

$$-w(x) \leq \inf_{y \in \mathcal{T}} [b\, d_{\mathcal{T}}(x, y) - w(y)]. \tag{86}$$

Let $(u, v) \in \mathcal{K}$ be arbitrary. Define the following sequence of dual potentials via infimal convolutions:

$$v^*(x) := \inf_{y \in \mathcal{T}} \{b[d_{\mathcal{T}}(x, y) - \lambda] - v(y)\} = -b\lambda + \inf_{y \in \mathcal{T}} [b\, d_{\mathcal{T}}(x, y) - v(y)] \geq u(x), \tag{87}$$

$$v^{**}(y) := \inf_{x \in \mathcal{T}} \{b[d_{\mathcal{T}}(x, y) - \lambda] - v^*(x)\} = -b\lambda + \inf_{x \in \mathcal{T}} [b\, d_{\mathcal{T}}(x, y) - v^*(x)] \geq v(y). \tag{88}$$

Now, observe that the lower and upper bounds for $v$ imply that

$$-b\lambda + \inf_{x \in \mathcal{T}} [b\, d_{\mathcal{T}}(x, y) - w(x)] \leq v(y) \leq w(y).$$

Using this together with Equation (86), we can derive pointwise bounds on $v^*$ for any $x \in \mathcal{T}$:

$$v^*(x) \leq -b\lambda - v(x) \leq -\inf_{y \in \mathcal{T}} [b\, d_{\mathcal{T}}(x, y) - w(y)] \leq w(x), \tag{89}$$

$$v^*(x) \geq -b\lambda + \inf_{y \in \mathcal{T}} [b\, d_{\mathcal{T}}(x, y) - w(y)] \geq -b\lambda - w(x). \tag{90}$$

We now show that $v^*$ is $b$-Lipschitz. Let $x_1, x_2 \in \mathcal{T}$ and fix an arbitrary $\varepsilon > 0$. By the definition of infimum, there exists $y_1 \in \mathcal{T}$ such that

$$b\, d_{\mathcal{T}}(x_1, y_1) - v(y_1) < v^*(x_1) + b\lambda + \varepsilon.$$

Then,

$$\begin{aligned}
v^*(x_2) - v^*(x_1) &\leq b\, d_{\mathcal{T}}(x_2, y_1) - v(y_1) - [b\, d_{\mathcal{T}}(x_1, y_1) - v(y_1)] + \varepsilon \\
&= b\, [d_{\mathcal{T}}(x_2, y_1) - d_{\mathcal{T}}(x_1, y_1)] + \varepsilon \leq b\, d_{\mathcal{T}}(x_1, x_2) + \varepsilon.
\end{aligned} \tag{91}$$

Since this holds for all $\varepsilon > 0$, we conclude that

$$v^*(x_2) - v^*(x_1) \leq b\, d_{\mathcal{T}}(x_1, x_2). \tag{92}$$

By symmetry, the reverse inequality also holds, so

$$|v^*(x_1) - v^*(x_2)| \le b \, d_{\mathcal{T}}(x_1, x_2), \tag{93}$$

which confirms that $v^*$ is $b$-Lipschitz.

Thus, $v^*$ belongs to the following class of functions:

$$\mathbb{L}' := \{f \in C(\mathcal{T}) : -b\lambda - w(x) \le f(x) \le w(x), \quad |f(x) - f(y)| \le b \, d_{\mathcal{T}}(x, y)\}. \tag{94}$$

This concludes the key regularity properties needed for the dual formulation.

We now establish the identity $v^{**} = -b\lambda - v^*$. To begin, note from the definition that:

$$v^{**}(y) = \inf_{x \in \mathcal{T}} [b(d_{\mathcal{T}}(x, y) - \lambda) - v^*(x)] \le -b\lambda - v^*(y). \tag{95}$$

On the other hand, since $v^*$ is $b$-Lipschitz, we have for all $x \in \mathcal{T}$:

$$-v^*(y) \le b \, d_{\mathcal{T}}(x, y) - v^*(x),$$

which implies

$$-b\lambda - v^*(y) \le \inf_{x \in \mathcal{T}} [b(d_{\mathcal{T}}(x, y) - \lambda) - v^*(x)] = v^{**}(y). \tag{96}$$

Combining both bounds, we conclude that

$$v^{**}(y) = -b\lambda - v^*(y). \tag{97}$$

Using this identity, we now bound the dual objective for any $(u, v) \in \mathcal{K}$:

$$\int_{\mathcal{T}} u(x) \, \mu(dx) + \int_{\mathcal{T}} v(x) \, \nu(dx) \le \int_{\mathcal{T}} v^*(x) \, \mu(dx) + \int_{\mathcal{T}} v^{**}(x) \, \nu(dx)$$

$$= \int_{\mathcal{T}} v^*(x) \, \mu(dx) - \int_{\mathcal{T}} v^*(x) \, \nu(dx) - b\lambda \, \nu(\mathcal{T})$$

$$= -b\lambda \, \nu(\mathcal{T}) + \int_{\mathcal{T}} v^*(x) \, (d\mu - d\nu). \tag{98}$$

Since $v^* \in \mathbb{L}'$ as shown earlier, we conclude:

$$\int_{\mathcal{T}} u(x) \, \mu(dx) + \int_{\mathcal{T}} v(x) \, \nu(dx) \le -b\lambda \, \nu(\mathcal{T}) + \sup_{f \in \mathbb{L}'} \int_{\mathcal{T}} f(x) \, (d\mu - d\nu). \tag{99}$$

Using the variational characterization of $\mathrm{ET}_\lambda(\mu, \nu)$ (proved earlier), we deduce the upper bound:

$$\mathrm{ET}_\lambda(\mu, \nu) \le -b\lambda \, \nu(\mathcal{T}) + \sup_{f \in \mathbb{L}'} \int_{\mathcal{T}} f(x) \, (d\mu - d\nu). \tag{100}$$

To prove the reverse inequality, let $f \in \mathbb{L}'$ and define:

$$u := f, \qquad v := -b\lambda - f.$$

Then:

$$u(x) \le w(x), \quad v(x) \le -b\lambda - (-b\lambda - w(x)) = w(x),$$

and

$$v(x) = -b\lambda - f(x) \ge -b\lambda - w(x)$$
$$\ge -b\lambda + \inf_{y \in \mathcal{T}} [b \, d_{\mathcal{T}}(x, y) - w(y)]. \tag{101}$$

Moreover, the $b$-Lipschitz property of $f$ yields:

$$u(x) + v(y) = f(x) - f(y) - b\lambda \le b \, (d_{\mathcal{T}}(x, y) - \lambda), \tag{102}$$

which confirms that $(u, v) \in \mathcal{K}$. Applying the variational formula for $\mathrm{ET}_\lambda$, we obtain:

$$-b\lambda \, \nu(\mathcal{T}) + \int_{\mathcal{T}} f(x) \, (d\mu - d\nu) = \int_{\mathcal{T}} u(x) \, \mu(dx) + \int_{\mathcal{T}} v(x) \, \nu(dx) \le \mathrm{ET}_\lambda(\mu, \nu). \tag{103}$$

Since this holds for all $f \in \mathbb{L}'$, we deduce:

$$-b\lambda \nu(\mathcal{T}) + \sup_{f \in \mathbb{L}'} \int_{\mathcal{T}} f(x) \, (d\mu - d\nu) \leq \text{ET}_\lambda(\mu, \nu). \tag{104}$$

Putting both directions together, we conclude:

$$\text{ET}_\lambda(\mu, \nu) = -b\lambda \nu(\mathcal{T}) + \sup_{f \in \mathbb{L}'} \int_{\mathcal{T}} f(x) \, (d\mu - d\nu). \tag{105}$$

To recover the symmetric form in Theorem B.2, let $f = \tilde{f} - \frac{b\lambda}{2}$. Then, $f \in \mathbb{L}'$ if and only if $\tilde{f} \in \mathbb{L}$. Furthermore:

$$\int_{\mathcal{T}} f(x) \, (d\mu - d\nu) = \int_{\mathcal{T}} \left( \tilde{f}(x) - \frac{b\lambda}{2} \right) (d\mu - d\nu) = \int_{\mathcal{T}} \tilde{f}(x) \, (d\mu - d\nu) - \frac{b\lambda}{2} \left[ \mu(\mathcal{T}) - \nu(\mathcal{T}) \right]. \tag{106}$$

Substituting into Equation (105), we obtain the final expression:

$$\text{ET}_\lambda(\mu, \nu) = \sup_{f \in \mathbb{L}} \int_{\mathcal{T}} f(x) \, (d\mu - d\nu) - \frac{b\lambda}{2} \left[ \mu(\mathcal{T}) + \nu(\mathcal{T}) \right], \tag{107}$$

which completes the proof. $\square$

### D.3 Proof for Proposition B.3

To ensure completeness, we provide full derivations of the result, closely following the methodology of [42].

*Proof.* We begin by expanding the definition of the regularized entropy transport:

$$\widetilde{\text{ET}}_\lambda^a(\mu, \nu) = -\frac{b\lambda}{2} \left[ \mu(\mathcal{T}) + \nu(\mathcal{T}) \right]$$
$$+ \sup \left\{ s \cdot \left[ \mu(\mathcal{T}) - \nu(\mathcal{T}) \right] : s \in \left[ -\frac{b\lambda}{2} - w(r) + a, \; w(r) + \frac{b\lambda}{2} - a \right] \right\}$$
$$+ \sup \left\{ \int_{\mathcal{T}} \left( \int_{[r,x]} g(y) \, \omega(dy) \right) (\mu - \nu)(dx) : \|g\|_{L^\infty(\mathcal{T})} \leq b \right\}. \tag{108}$$

We now evaluate each supremum separately:

- The first supremum corresponds to maximizing a linear function over a symmetric interval. Therefore, it evaluates to

$$\left[ w(r) + \frac{b\lambda}{2} - a \right] \cdot |\mu(\mathcal{T}) - \nu(\mathcal{T})|. \tag{109}$$

- The second supremum is equivalent to the dual representation of a Lipschitz-type transport energy over tree-structured domains. As established in [21, pp. 575–576], we have:

$$\sup \left\{ \int_{\mathcal{T}} \left( \int_{[r,x]} g(y) \, \omega(dy) \right) (\mu - \nu)(dx) : \|g\|_{L^\infty(\mathcal{T})} \leq b \right\} = \int_{\mathcal{T}} |\mu(\Lambda(x)) - \nu(\Lambda(x))| \, \omega(dx). \tag{110}$$

Combining both components, we obtain the closed-form expression:

$$\widetilde{\text{ET}}_\lambda^a(\mu, \nu) = \int_{\mathcal{T}} |\mu(\Lambda(x)) - \nu(\Lambda(x))| \, \omega(dx) - \frac{b\lambda}{2} \left[ \mu(\mathcal{T}) + \nu(\mathcal{T}) \right]$$
$$+ \left[ w(r) + \frac{b\lambda}{2} - a \right] \cdot |\mu(\mathcal{T}) - \nu(\mathcal{T})|. \tag{111}$$

This concludes the proof. $\square$

## D.4 Proof for Proposition B.4

To ensure completeness, we provide full derivations of the result, closely following the methodology of [42].

*Proof.* We begin with the upper bound $\mathrm{ET}_\lambda(\mu, \nu) \le \widetilde{\mathrm{ET}}_\lambda^0(\mu, \nu)$. This follows directly from the inclusion $\mathbb{L} \subset \mathbb{L}_0$ and the dual representation of $\mathrm{ET}_\lambda$ established in Theorem B.2.

Next, consider $a$ satisfying

$$2bL(\mathcal{T}) \le a \le \frac{b\lambda}{2} + w(r). \tag{112}$$

We will show that under this condition, the inclusion $\mathbb{L}_a \subset \mathbb{L}$ holds. Then, by Theorem B.2, it follows that

$$\widetilde{\mathrm{ET}}_\lambda^a(\mu, \nu) \le \mathrm{ET}_\lambda(\mu, \nu). \tag{113}$$

To prove $\mathbb{L}_a \subset \mathbb{L}$, we need to show that any function $f \in \mathbb{L}_a$ satisfies

$$-w(x) - \frac{b\lambda}{2} \le f(x) \le w(x) + \frac{b\lambda}{2}, \quad \forall x \in \mathcal{T}. \tag{114}$$

Let $f \in \mathbb{L}_a$. Then by definition,

$$f(x) = s + \int_{[r,x]} g(y)\,\omega(dy), \tag{115}$$

where $s \in \left[-w(r) - \frac{b\lambda}{2} + a,\ w(r) + \frac{b\lambda}{2} - a\right]$ and $\|g\|_{L^\infty(\mathcal{T})} \le b$. Using this, we bound $f(x)$ from above:

$$\begin{aligned}
f(x) &\le s + \|g\|_{L^\infty(\mathcal{T})} \cdot \omega([r,x]) \\
&\le w(r) + \frac{b\lambda}{2} - a + bL(\mathcal{T}) \\
&\le w(x) + \frac{b\lambda}{2} - a + 2bL(\mathcal{T}) \\
&\le w(x) + \frac{b\lambda}{2}.
\end{aligned} \tag{116}$$

For the lower bound, we have:

$$\begin{aligned}
f(x) &\ge s - \|g\|_{L^\infty(\mathcal{T})} \cdot \omega([r,x]) \\
&\ge -w(r) - \frac{b\lambda}{2} + a - bL(\mathcal{T}) \\
&\ge -w(x) - \frac{b\lambda}{2} + a - 2bL(\mathcal{T}) \\
&\ge -w(x) - \frac{b\lambda}{2}.
\end{aligned} \tag{117}$$

Hence, $f$ satisfies the defining constraints of $\mathbb{L}$ and we conclude that $f \in \mathbb{L}$. Therefore, $\mathbb{L}_a \subset \mathbb{L}$ for all $a \ge 2bL(\mathcal{T})$.

It follows from Theorem B.2 and the definition of $\widetilde{\mathrm{ET}}_\lambda^a$ that

$$\widetilde{\mathrm{ET}}_\lambda^a(\mu, \nu) \le \mathrm{ET}_\lambda(\mu, \nu). \tag{118}$$

This concludes the proof. $\qquad\square$

## D.5 Proof for Proposition B.5

To ensure completeness, we provide full derivations of the result, closely following the methodology of [42].

*Proof.* We first observe that the metric $d_a$ depends solely on the value of the weight function at the root $r$ of the tree $\mathcal{T}$. This follows directly from the definition of $\mathbb{L}_a$, where only $w(r)$ appears explicitly.

By construction, we have the variational characterization:

$$d_a(\mu, \nu) = \sup \left\{ \int_{\mathcal{T}} f \,(d\mu - d\nu) : f \in \mathbb{L}_a \right\}. \tag{119}$$

Let us now verify the metric properties:

**(Non-negativity)** Clearly, $d_a(\mu, \nu) \geq 0$ from the supremum structure. Moreover, $d_a(\mu, \mu) = 0$ for all $\mu$ by linearity of the integral. Now suppose that $d_a(\mu, \nu) = 0$. Using the closed-form expression for $\widetilde{\mathrm{ET}}_\lambda^a$ in Proposition B.3, this implies:

$$\left[ w(r) + \frac{b\lambda}{2} - a \right] \cdot |\mu(\mathcal{T}) - \nu(\mathcal{T})| + \int_{\mathcal{T}} |\mu(\Lambda(x)) - \nu(\Lambda(x))| \, \omega(dx) = 0. \tag{120}$$

Since the first term has a strictly positive coefficient by assumption ($a < w(r) + \frac{b\lambda}{2}$), we must have $\mu(\mathcal{T}) = \nu(\mathcal{T})$ and

$$\mu(\Lambda(x)) = \nu(\Lambda(x)) \quad \text{for all } x \in \mathcal{T}. \tag{121}$$

By [42, Lemma A.2], this implies that $\mu = \nu$, establishing identity of indiscernibles.

**(Symmetry)** Note that if $f \in \mathbb{L}_a$, then $-f \in \mathbb{L}_a$ by the symmetric definition of the function class. Therefore, from Equation (119), we obtain

$$d_a(\mu, \nu) = d_a(\nu, \mu). \tag{122}$$

**(Triangle Inequality)** The triangle inequality holds immediately from the supremum definition over a convex, symmetric function class:

$$d_a(\mu, \sigma) + d_a(\sigma, \nu) \geq \int_{\mathcal{T}} f \,(d\mu - d\sigma) + \int_{\mathcal{T}} f \,(d\sigma - d\nu) = \int_{\mathcal{T}} f \,(d\mu - d\nu), \tag{123}$$

for all $f \in \mathbb{L}_a$, and taking the supremum yields the inequality.

Hence, $d_a$ satisfies all properties of a metric on $\mathcal{M}(\mathcal{T})$, and the proof is complete. $\square$

### D.6 Proof for Equation (13)

*Proof.* We recall Equation (13). Let $f \in L^1(\mathbb{R}^d)$ be a non-negative density function. The Radon Transform $\mathcal{R}^\alpha$ maps $f$ to a density defined on a tree system $\mathcal{T}$, while preserving the total mass:

$$\|f\|_1 = \int_{\mathbb{R}^d} f(x) \, dx = \|\mathcal{R}_{\mathcal{T}}^\alpha f\|_{\mathcal{T}}, \quad \text{for all} \quad \mathcal{T} \in \mathbb{T}. \tag{124}$$

To establish this property, we first observe that the non-negativity of $\alpha$ ensures that the transform preserves non-negativity: if $f \geq 0$, then $\mathcal{R}_{\mathcal{T}}^\alpha f \geq 0$, implying that the transformed function is a valid density. The preservation of total mass then follows directly from the definition of $\mathcal{R}^\alpha$, which integrates over linearly parameterized subsets aligned with the structure of $\mathcal{T}$.

$$
\begin{aligned}
\|\mathcal{R}_{\mathcal{T}}^\alpha f\|_{\mathcal{T}} &= \sum_{l \in \mathcal{T}} \int_{\mathbb{R}} |\mathcal{R}_{\mathcal{T}}^\alpha f(t_x, l)| \, dt_x \\
&= \sum_{l \in \mathcal{L}} \int_{\mathbb{R}} \left| \int_{\mathbb{R}^d} f(y) \cdot \alpha(y, \mathcal{L})_l \cdot \delta\left(t_x - \langle y - x_l, \theta_l \rangle\right) \, dy \right| \, dt_x \\
&= \sum_{l \in \mathcal{L}} \int_{\mathbb{R}} \left( \int_{\mathbb{R}^d} f(y) \cdot \alpha(y, \mathcal{L})_l \cdot \delta\left(t_x - \langle y - x_l, \theta_l \rangle\right) \, dy \right) \, dt_x \\
&= \sum_{l \in \mathcal{L}} \int_{\mathbb{R}^d} \left( \int_{\mathbb{R}} f(y) \cdot \alpha(y, \mathcal{L})_l \cdot \delta\left(t_x - \langle y - x_l, \theta_l \rangle\right) \, dt_x \right) \, dy
\end{aligned}
$$

$$
\begin{aligned}
&= \sum_{l \in \mathcal{L}} \int_{\mathbb{R}^d} f(y) \cdot \alpha(y, \mathcal{L})_l \cdot \left( \int_{\mathbb{R}} \delta \left( t_x - \langle y - x_l, \theta_l \rangle \right) \, dt_x \right) \, dy \\
&= \sum_{l \in \mathcal{L}} \int_{\mathbb{R}^d} f(y) \cdot \alpha(y, \mathcal{L})_l \, dy \\
&= \int_{\mathbb{R}^d} f(y) \cdot \sum_{l \in \mathcal{L}} \alpha(y, \mathcal{L})_l \, dy \\
&= \int_{\mathbb{R}^d} f(y) \, dy \\
&= \|f\|_1.
\end{aligned}
\tag{125}
$$

The proof is completed. $\qquad\square$

### D.7  Proof for Theorem 3.3

*Proof.* We consider the expression

$$
\text{PartialTSW}(\mu, \nu) = \int_{\mathbb{T}} d_a(\mu_{\mathcal{T}}, \nu_{\mathcal{T}}) \, d\sigma(\mathcal{T}),
\tag{126}
$$

and show that it defines a metric on $\mathcal{M}(\mathbb{R}^d)$. Since the splitting map $\alpha$ is $\text{E}(d)$-invariant, the Radon Transform $\mathcal{R}^\alpha$ is injective; that is, for any $f \in L^1(\mathbb{R}^d)$, if $\mathcal{R}_{\mathcal{T}}^\alpha f = 0$ for all $\mathcal{T} \in \mathbb{T}$, then $f = 0$ (see [80]). We now verify the three properties required for PartialTSW to be a metric on $\mathcal{M}(\mathbb{R}^d)$.

**Positive definiteness.** For $\mu, \nu \in \mathcal{P}(\mathbb{R}^n)$, it is clear that $\text{PartialTSW}(\mu, \mu) = 0$ and $\text{PartialTSW}(\mu, \nu) \geq 0$. Moreover, if $\text{PartialTSW}(\mu, \nu) = 0$, then $d_a(\mu_{\mathcal{T}}, \nu_{\mathcal{T}}) = 0$ for all $\mathcal{T} \in \mathbb{T}$. Since $d_a$ is a metric on $\mathcal{M}(\mathcal{T})$, it follows that $\mu_{\mathcal{T}} = \nu_{\mathcal{T}}$ for all $\mathcal{T}$. Hence, $\mathcal{R}_{\mathcal{T}}^\alpha f_\mu = \mathcal{R}_{\mathcal{T}}^\alpha f_\nu$ for all $\mathcal{T} \in \mathbb{T}$. By the injectivity of $\mathcal{R}^\alpha$, we conclude that $f_\mu = f_\nu$, and thus $\mu = \nu$.

**Symmetry.** For any $\mu, \nu \in \mathcal{M}(\mathbb{R}^n)$, we have:

$$
\begin{aligned}
\text{PartialTSW}(\mu, \nu) &= \int_{\mathbb{T}} d_a(\mu_{\mathcal{T}}, \nu_{\mathcal{T}}) \, d\sigma(\mathcal{T}) \\
&= \int_{\mathbb{T}} d_a(\nu_{\mathcal{T}}, \mu_{\mathcal{T}}) \, d\sigma(\mathcal{T}) \tag{127} \\
&= \text{PartialTSW}(\nu, \mu). \tag{128}
\end{aligned}
$$

Therefore, $\text{PartialTSW}(\mu, \nu) = \text{PartialTSW}(\nu, \mu)$.

**Triangle inequality.** For $\mu_1, \mu_2, \mu_3 \in \mathcal{M}(\mathbb{R}^n)$, we compute:

$$
\begin{aligned}
&\text{PartialTSW}(\mu_1, \mu_2) + \text{PartialTSW}(\mu_2, \mu_3) \\
&= \int_{\mathbb{T}} d_a(\mu_{1,\mathcal{T}}, \mu_{2,\mathcal{T}}) \, d\sigma(\mathcal{T}) + \int_{\mathbb{T}} d_a(\mu_{2,\mathcal{T}}, \mu_{3,\mathcal{T}}) \, d\sigma(\mathcal{T}) \\
&= \int_{\mathbb{T}} \left( d_a(\mu_{1,\mathcal{T}}, \mu_{2,\mathcal{T}}) + d_a(\mu_{2,\mathcal{T}}, \mu_{3,\mathcal{T}}) \right) \, d\sigma(\mathcal{T}) \\
&\geq \int_{\mathbb{T}} d_a(\mu_{1,\mathcal{T}}, \mu_{3,\mathcal{T}}) \, d\sigma(\mathcal{T}) \\
&= \text{PartialTSW}(\mu_1, \mu_3), \tag{129}
\end{aligned}
$$

where the inequality follows from the triangle inequality satisfied by $d_a$ on each tree $\mathcal{T}$.

In conclusion, PartialTSW satisfies all properties of a metric on the space $\mathcal{M}(\mathbb{R}^d)$.

We aim to show that PartialTSW is $\text{E}(d)$-invariant, meaning that for any $g \in \text{E}(d)$, the following holds:

$$
\text{PartialTSW}(\mu, \nu) = \text{PartialTSW}(g \sharp \mu, g \sharp \nu),
\tag{130}
$$

where $g\sharp\mu$ and $g\sharp\nu$ denote the pushforwards of $\mu$ and $\nu$, respectively, under the Euclidean transformation $g\colon \mathbb{R}^d \to \mathbb{R}^d$.

Let $\mathcal{T} \in \mathbb{T}$ be a tree system given by $\mathcal{T} = \{l_i = (x_i, \theta_i)\}_{i=1}^k$. Then, under the action of $g = (Q, a)$, we have

$$g\mathcal{T} = \{gl_i = (Qx_i + a, Q\theta_i)\}_{i=1}^k. \tag{131}$$

We also note that $g\sharp f_\mu = f_{g\sharp\mu}$ and $g\sharp f_\nu = f_{g\sharp\nu}$. Since $|\det(Q)| = 1$, we compute:

$$
\begin{aligned}
\mathcal{R}_{g\mathcal{L}}^\alpha(g\sharp f_\mu)(gx, gl) &= \int_{\mathbb{R}^d} (g\sharp f_\mu)(y) \cdot \alpha(y, g\mathcal{L})_l \cdot \delta\left(t_{gx} - \langle y - x_{gl}, \theta_{gl}\rangle\right) \, dy \\
&= \int_{\mathbb{R}^d} f_\mu(g^{-1}y) \cdot \alpha(y, g\mathcal{L})_l \cdot \delta\left(t_x - \langle y - x_{gl}, \theta_{gl}\rangle\right) \, dy \\
&= \int_{\mathbb{R}^d} f_\mu(g^{-1}gy) \cdot \alpha(gy, g\mathcal{L})_l \cdot \delta\left(t_x - \langle gy - x_{gl}, \theta_{gl}\rangle\right) \, d(gy) \\
&= \int_{\mathbb{R}^d} f_\mu(y) \cdot \alpha(y, \mathcal{L})_l \cdot \delta\left(t_x - \langle gy - x_{gl}, \theta_{gl}\rangle\right) \, dy \\
&= \int_{\mathbb{R}^d} f_\mu(y) \cdot \alpha(y, \mathcal{L})_l \cdot \delta\left(t_x - \langle Qy + a - Qx_l - a, Q\theta_l\rangle\right) \, dy \\
&= \int_{\mathbb{R}^d} f_\mu(y) \cdot \alpha(y, \mathcal{L})_l \cdot \delta\left(t_x - \langle Q(y - x_l), Q\theta_l\rangle\right) \, dy \\
&= \int_{\mathbb{R}^d} f_\mu(y) \cdot \alpha(y, \mathcal{L})_l \cdot \delta\left(t_x - \langle y - x_l, \theta_l\rangle\right) \, dy \\
&= \mathcal{R}_{\mathcal{L}}^\alpha f_\mu(x, l). \tag{132}
\end{aligned}
$$

A similar computation gives:

$$\mathcal{R}_{g\mathcal{L}}^\alpha(g\sharp f_\nu)(gx, gl) = \mathcal{R}_{\mathcal{L}}^\alpha f_\nu(x, l). \tag{133}$$

Moreover, since $g$ acts isometrically on tree systems, the induced measures satisfy:

$$d_a(\mu_\mathcal{T}, \nu_\mathcal{T}) = d_a((g\sharp\mu)_{g\mathcal{T}}, (g\sharp\nu)_{g\mathcal{T}}). \tag{134}$$

Thus, we compute:

$$
\begin{aligned}
\text{PartialTSW}(g\sharp\mu, g\sharp\nu) &= \int_{\mathbb{T}} d_a((g\sharp\mu)_\mathcal{T}, (g\sharp\nu)_\mathcal{T}) \, d\sigma(\mathcal{T}) \\
&= \int_{\mathbb{T}} d_a((g\sharp\mu)_{g\mathcal{T}}, (g\sharp\nu)_{g\mathcal{T}}) \, d\sigma(g\mathcal{T}) \\
&= \int_{\mathbb{T}} d_a(\mu_\mathcal{T}, \nu_\mathcal{T}) \, d\sigma(g\mathcal{T}) \\
&= \int_{\mathbb{T}} d_a(\mu_\mathcal{T}, \nu_\mathcal{T}) \, d\sigma(\mathcal{T}) \\
&= \text{PartialTSW}(\mu, \nu). \tag{135}
\end{aligned}
$$

We conclude that PartialTSW is $\mathrm{E}(d)$-invariant. $\qquad\square$

**Remark D.1.** We omit almost-sure conditions in the above proof, as they are straightforward to verify and would otherwise obscure the main argument.

# E    Experimental Details

## E.1    Algorithm for Partial Tree-Sliced Wasserstein Distance

The computation of the Partial Tree-Sliced Wasserstein (PartialTSW) distance is outlined in Algorithm 1. This procedure estimates the distance by averaging costs derived from multiple tree-based projections of the input measures.

**Algorithm 1** Partial Tree-Sliced Wasserstein distance.

---

**Input:** Measures $\mu$ and $\nu$ in $\mathcal{M}(\mathbb{R}^d)$, number of tree systems $L$, number of lines in tree system $k$, space of tree systems $\mathbb{T}$, splitting maps $\alpha$, parameters $a, b, \lambda$, total mass $\mu(\mathcal{T}), \nu(\mathcal{T})$.
Scale total mass of $\mu$ and $\nu$ such that $\mu(\mathbb{R}^d) = \mu(\mathcal{T})$, $\nu(\mathbb{R}^d) = \nu(\mathcal{T})$.
**for** $i = 1$ to $L$ **do**
    Sampling $x \in \mathbb{R}^d$ and $\theta_1, \ldots, \theta_k \overset{i.i.d}{\sim} \mathcal{U}(\mathbb{S}^{d_\theta - 1})$.
    Contruct tree system $\mathcal{L}_i = \{(x, \theta_1), \ldots, (x, \theta_k)\}$.
    Projecting $\mu$ and $\nu$ onto $\mathcal{T}_i$ to get $\mathcal{R}^\alpha_{\mathcal{L}_i} \mu$ and $\mathcal{R}^\alpha_{\mathcal{L}_i} \nu$.
    Compute $\widehat{\text{PartialTSW}}(\mu, \nu) = (1/L) \cdot d_a(\mathcal{R}^\alpha_{\mathcal{L}_i} \mu, \mathcal{R}^\alpha_{\mathcal{L}_i} \nu)$.
**end for**
**Return:** $\widehat{\text{PartialTSW}}(\mu, \nu)$.

---

### E.2 Computational and Memory Complexity Analysis

This section details the computational and memory demands of our proposed PartialTSW distance. We consider input measures $\mu$ and $\nu$ represented by $N$ samples in a $d$-dimensional space, with $L$ tree constructions and $k$ lines per tree.

Table 3 outlines the complexity of key operations. The dominant factors are the distance-based weight splitting ($\mathcal{O}(LkNd)$) for projecting samples and the sorting of these projected 1D coordinates ($\mathcal{O}(LkN \log N)$). Consequently, the total computational complexity is $\mathcal{O}(LkNd + LkN \log N)$. The primary memory consumers are the storage of split weights, tree/line parameters, and the original data, leading to an overall memory requirement of $\mathcal{O}(LkN + Lkd + Nd)$.

Table 3: Detailed complexity analysis for PartialTSW. ($N$ = number of samples, $d$ = dimension, $L$ = number of trees, $k$ = lines per tree).

| Operation Category | Specific Steps Involved | Computational Cost | Memory Cost |
|---|---|---|---|
| Initial Mass Scaling | Adjusting sample weights for $\mu$ and $\nu$ to meet target total masses. | $O(N)$ | $O(N)$ |
| Distance-Based Weight Splitting | Calculation of distances from $N$ points to $Lk$ lines, and subsequent softmax for weight distribution. | $\mathcal{O}(LkNd)$ | $\mathcal{O}(LkN + Lkd + Nd)$ |
| Sorting Projected Data | Sorting the $N$ projected coordinates along each of the $Lk$ lines. | $\mathcal{O}(LkN \log N)$ | $\mathcal{O}(LkN)$ |
| **Overall Total** | | $\mathcal{O}(LkNd + LkN \log N)$ | $\mathcal{O}(LkN + Lkd + Nd)$ |

**GPU Memory Optimization for Distance-Based Splitting.** The practical GPU memory footprint for the distance-based splitting step can be significantly lower than a naive theoretical estimate. As highlighted by [80], this operation involves (1) computing $d$-dimensional distance vectors from points to lines, (2) calculating their norms, and (3) applying a softmax function across lines within each tree to obtain split weights. While a direct implementation might suggest $\mathcal{O}(LkNd)$ memory for storing all intermediate distance vectors, modern deep learning frameworks like PyTorch, when using compilation tools (e.g., 'torch.compile'), can perform kernel fusion. This optimization merges these sequential computations into fewer GPU kernels, potentially allowing large intermediate tensors (like the full $LkN \times d$ distance vectors) to reside in faster, smaller shared memory or be recomputed on-the-fly, rather than occupying global GPU memory. Consequently, the persistent global memory primarily stores the essential data: line parameters ($\mathcal{O}(Lkd)$), sample coordinates ($\mathcal{O}(Nd)$), and the resulting split weights ($\mathcal{O}(LkN)$), aligning with the $\mathcal{O}(LkN + Lkd + Nd)$ overall memory profile.

### E.3 Empirical Runtime and Memory Performance of PartialTSW

We present an empirical evaluation of the runtime and memory usage of PartialTSW. The experiments were conducted on a single NVIDIA H100 GPU. We fixed the number of tree iterations $L = 10$ and lines per tree $k = 4$. The analysis varies the number of samples $N \in \{100, 1K, 5K, 10K, 500K\}$ and the data dimension $d \in \{50, 100, 500, 1000\}$.

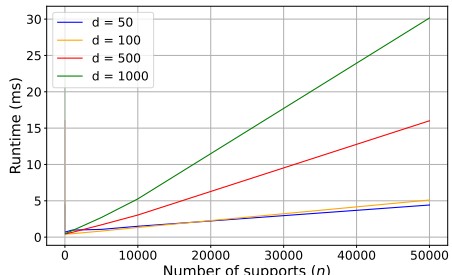 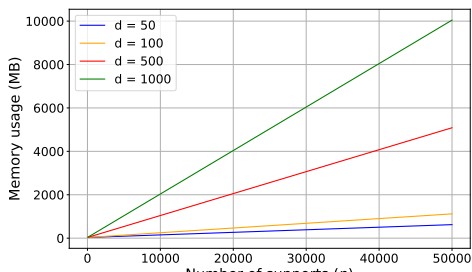

Figure 7: Empirical runtime (left) and peak memory usage (right) for PartialTSW, varying the number of samples ($N$) and data dimension ($d$). ($L = 10, k = 4$).

**Runtime Scalability.** The empirical results, depicted in Figure 7 (left), illustrate how the runtime of PartialTSW scales with the number of samples $N$ and the data dimension $d$. The runtime exhibits a near-linear increase with $N$. For instance, processing $N = 50,000$ samples takes approximately five times longer than $N = 10,000$ samples (when $d, L, k$ are fixed), which is consistent with the $\mathcal{O}(Nd + N \log N)$ dependency on $N$ from our theoretical analysis (Section E.2). Regarding dimensionality, the runtime also demonstrates a linear dependency on $d$. For example, increasing $d$ from 10000 to 50000 (a 5x increase) results in a correspondingly proportional increase in runtime for a fixed $N$. This aligns with the $\mathcal{O}(d)$ factor in the $LkNd$ term of the complexity. These empirical observations support the theoretical computational complexity.

**Memory Scalability.** Figure 7 (right) showcases the memory consumption characteristics of PartialTSW. The peak memory usage scales linearly with both the number of samples $N$ and the dimension $d$. This behavior is predictable and directly corresponds to our theoretical memory complexity of $\mathcal{O}(LkN + Lkd + Nd)$, indicating efficient memory utilization that grows manageably with data size and dimensionality.

### E.4 Sample Complexity and Estimator Stability

We first clarify the roles of the two key parameters in our method.

- Lines per Tree ($k$): This is a *structural parameter* that defines the ground-truth distance PartialTSW$_k(\mu, \nu)$. As discussed in the TSW literature, using $k > 1$ enhances the method's capacity to capture complex topological and structural features of the data.

- Number of Trees ($L$): This is the *Monte Carlo (MC) estimation parameter*. Our method approximates the ground-truth distance by averaging over $L$ independently sampled random trees.

As discussed in the computational complexity analysis in Section 3.3, the total computational cost is proportional to the total number of 1D projections, which is $N = Lk$. For a fixed computational budget $N$, this creates a natural trade-off: increasing $k$ improves topological expressiveness, but requires decreasing $L$, which in turn affects the stability of the MC estimate.

In our experiments, we tune the structural parameter $k$ for empirical performance and then set $L = N/k$ to ensure the total number of projections $N$ remains fixed, allowing for a fair comparison against baselines under the same computational budget.

This approach raises a valid concern about the stability of the estimator, as the MC approximation error for a fixed $k$ decreases at a rate of $\mathcal{O}(L^{-1/2})$. We therefore provide an empirical analysis of the estimator's convergence. We measure stability using the Coefficient of Variation (CoV = $\sigma/\mu$), a normalized metric where $\sigma$ is the standard deviation and $\mu$ is the mean of the distance estimate over multiple runs.

Table 4 shows the CoV as a function of both the MC parameter $L$ and the structural parameter $k$. The results empirically verify the expected convergence. For any fixed $k$, the estimator's stability improves (i.e., CoV decreases) as the number of MC samples $L$ increases, aligning with the theoretical $\mathcal{O}(L^{-1/2})$ rate. For instance, at $k = 5$, increasing $L$ from 10 to 1000 reduces the CoV by over $14\times$.

Table 4: Estimator stability analysis. The table shows the Coefficient of Variation (CoV = $\sigma/\mu$) for the PartialTSW distance as a function of the number of tree slices ($L$) and the number of lines per tree ($k$).

| Number of Trees ($L$) | $k = 5$ | $k = 10$ | $k = 100$ |
|---|---|---|---|
| 10 | 0.1098 | 0.1016 | 0.0456 |
| 50 | 0.0526 | 0.0252 | 0.0183 |
| 100 | 0.0263 | 0.0239 | 0.0091 |
| 500 | 0.0105 | 0.0133 | 0.0070 |
| 1000 | 0.0076 | 0.0070 | 0.0041 |
| 10000 | 0.0029 | 0.0018 | 0.0023 |
| 20000 | 0.0016 | 0.0024 | 0.0010 |

This analysis confirms that in our large-scale experiments, where we use a high number of projections (e.g., $L \geq 1000$), the resulting distance estimate is stable and reliable.

## E.5 Discussion on Hyperparameters of Evaluated Methods

This section briefly outlines the key hyper-parameters for each evaluated Unbalanced Optimal Transport (UOT) and Partial Optimal Transport (POT) method and their respective roles.

**SPOT [9].** The hyperparameter $k$ specifies the number of points to be transported, thereby defining the partial nature of the matching between distributions.

**SOPT [2].** The regularization parameter $\lambda$ controls the "partialness" of the transport by influencing the total amount of mass that is optimally transported between distributions.

**Sinkhorn [74].** The hyperparameter $reg$ is the entropic regularization coefficient that smooths the optimal transport plan. The hyperparameter $reg_m$ is the marginal regularization coefficient that penalizes deviations from the prescribed marginal constraints, thus allowing for mass variation.

**SUOT and USOT [7].** The hyper-parameters $\rho_1$ and $\rho_2$ are regularization parameters. They respectively control the cost of deviating from the source and target marginals in the sliced domain, enabling unbalanced transport by permitting mass creation or destruction.

**PAWL [13].** The hyperparameter $k$ the number of points to be transported, effectively determining the extent of partiality in this unbalanced optimal transport formulation.

**UOT-FM [22].** The hyperparameter $\lambda$ influences the regularization of marginal constraints, thereby controlling the degree to which the masses of the coupled distributions must be preserved during transport.

**ULightOT [28].** The hyperparameter $\tau$ governs the extent of mass conservation, adjusting how strictly the total mass of the transported distribution must adhere to the original or target masses.

**Partial-TSW (Ours).** The mass parameter $\nu(\mathcal{T})$ specifies the proportion of the target distribution's mass to be matched by the transport plan. The source distribution's mass proportion, $\mu(\mathcal{T})$, is typically fixed at 1, so adjusting $\nu(\mathcal{T})$ controls the partiality of the matching against the target.

## E.6 Comparing Computational Efficiency

To ensure consistent and fair results, two warm-up runs were performed for each method and each sample size $n$ before conducting 10 timed repetitions. The average runtime and peak memory usage (for GPU methods) were then recorded. Unless otherwise specified (as in the discussion on varying $d$ below), these experiments were conducted with data of dimension $d = 2$ and for sample sizes $n$ ranging from $10^2$ to $10^5$.

Since hyperparameter choices can significantly affect algorithmic runtime, the specific settings used for each method in this runtime comparison are detailed below. For a general description of these hyperparameters and their roles, please refer to Appendix §E.5.

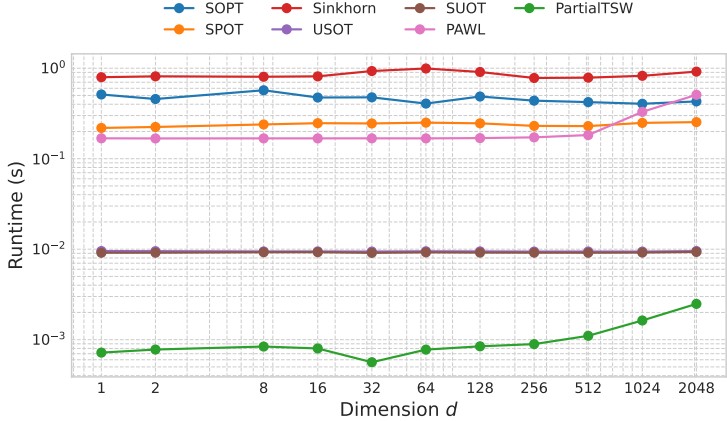

Figure 8: Runtime comparison for PartialTSW and POT/UOT solvers over data dimension $d$.

Common settings for the compared sliced-based methods (SOPT, SPOT, USOT, SUOT, PAWL) included $L = 10$ projections. For PartialTSW (Ours), we used `num_trees` $= 5$ and `num_lines` $= 2$. This configuration for PartialTSW, where the product of `num_trees` $\times$ `num_lines` $= 10$, offers a comparable number of one-dimensional sorting operations to the $L = 10$ setting in other sliced methods, aiming for a fair comparison. Specific hyperparameters for each method were then set as follows:

- **SOPT**: Regularization parameter $\lambda = 1.0$.

- **SPOT**: The number of transported points $k$ was set to $n$ (the input sample size for each distribution), implying a full matching was performed. (Number of projections $L = 10$, as stated above).

- **Sinkhorn**: Entropic regularization $reg = 0.1$, marginal KL regularization $reg_m = 1.0$, maximum number of Sinkhorn iterations 'numItermax' $= 100$, and stopping threshold 'stopThr' $= 10^{-5}$.

- **USOT and SUOT**: Regularization parameters $\rho_1 = 0.01$ and $\rho_2 = 1.0$.

- **PAWL**: The number of transported points $k$ was set to $n$ (implying a full matching).

- **PartialTSW (Ours)**: The target mass proportion $\nu(\mathcal{T})$ was set to $1.0$ (with the source mass proportion $\mu(\mathcal{T})$ typically assumed to be $1.0$). This choice was made because $\nu(\mathcal{T})$ does not affect the computational runtime of the PartialTSW implementation used in this benchmark.

Furthermore, we present a runtime comparison for varying data dimensions $d$ in Figure 8. The results indicate that the runtime is not significantly affected when $d$ increases.

The runtime comparisons for all methods were conducted with an Intel Xeon Platinum 8580 CPU and an NVIDIA H100 GPU.

### E.7 Noisy Point Cloud Gradient Flow

We used clean point cloud data obtained from [2] for the dragon and bunny shapes. Each clean dataset contains 10k data points. We randomly select and add 7% noise points to the target point cloud (bunny). Inspired by [2], the noise is sampled from the region $[-0.6M, 0.6M]^3$ where $M = \max_{i \in \{1,n\}}(\|x_i\|)$, where $x_i$ is the point in the target. In total, the target point cloud consists of 10k clean points and an additional 700 noise points. We use $L = 10$ projections for SW, and $L = 5$ trees, $k = 2$ lines for TSW and PartialTSW. All methods are trained using Adam optimizer with a learning rate of $10^{-3}$ over 300 epochs. The results are shown in Figure 3.

All experiments were conducted with an Intel Xeon Platinum 8580 CPU and an NVIDIA H100 GPU.

### E.8 Robust Generative Model

#### E.8.1 Implementation detail

**Pre-training an Autoencoder (AE).** An Autoencoder (AE) is pre-trained to provide 2D latent representations $z \in \mathbb{R}^2$ for MNIST digits. We employ a Wasserstein Autoencoder with MMD regularization (WAE-MMD) [77] architecture. The AE is trained for 50 epochs using the Adam optimizer with a learning rate of $3 \times 10^{-5}$ and a batch size of 256. The WAE-MMD loss uses a $\lambda$ hyperparameter of 500.0 to balance reconstruction and MMD regularization terms. For the MMD term, we match the aggregated posterior $q(z)$ to a uniform prior distribution $p(z) \sim \mathcal{U}[-1, 1]^2$. This encourages the learned latent space to reside approximately within $[-1, 1]^2$. The training data for the AE consists of MNIST digits 0 and 1, balanced and augmented as described below. The latent dimension is set to $d = 2$.

The Autoencoder, $AE : [0, 1]^{1 \times 28 \times 28} \rightarrow [0, 1]^{1 \times 28 \times 28}$, architecture is as follows:

- **Encoder:**
    - Input: $1 \times 28 \times 28$ (MNIST image)
    - $\texttt{Conv2d}(\text{in\_channels} = 1, \text{out\_channels} = 32, \text{kernel\_size} = 4, \text{stride} = 2, \text{padding} = 1) \rightarrow \texttt{ReLU}$  $(32 \times 14 \times 14)$
    - $\texttt{Conv2d}(32, 64, \text{kernel\_size} = 4, \text{stride} = 2, \text{padding} = 1) \rightarrow \texttt{ReLU}$  $(64 \times 7 \times 7)$
    - Flatten: $64 \times 7 \times 7 = 3136$ features
    - $\texttt{Linear}(\text{in\_features} = 3136, \text{out\_features} = 512) \rightarrow \texttt{ReLU}$
    - $\texttt{Linear}(512, \text{latent\_dim} = 2)$ (for mean $\mu$)
    - $\texttt{Linear}(512, \text{latent\_dim} = 2)$ (for log-variance $\log \sigma^2$)
    - Latent vector $z = \mu + \epsilon \odot \sigma$ (Reparameterization trick)
- **Decoder:**
    - Input: $z \in \mathbb{R}^{\text{latent\_dim}=2}$
    - $\texttt{Linear}(\text{latent\_dim} = 2, 512) \rightarrow \texttt{ReLU}$
    - $\texttt{Linear}(512, 3136) \rightarrow \texttt{ReLU}$
    - Reshape to $64 \times 7 \times 7$
    - $\texttt{ConvTranspose2d}(64, 32, \text{kernel\_size} = 4, \text{stride} = 2, \text{padding} = 1) \rightarrow \texttt{ReLU}$ $(32 \times 14 \times 14)$
    - $\texttt{ConvTranspose2d}(32, 1, \text{kernel\_size} = 4, \text{stride} = 2, \text{padding} = 1) \rightarrow \texttt{Sigmoid}$ $(1 \times 28 \times 28)$

**Dataset Augmentation for Auxiliary Models.** To ensure robust training of the AE and the digit classifier, we prepare a balanced and augmented training set from MNIST digits 0 and 1. The original MNIST training set contains an unequal number of samples for these digits. We balance these classes by applying data augmentation to the minority class until its sample count matches the majority class. Augmentations include random affine transformations (degrees: $\pm 15°$, translation: $\pm 0.15$ of image dimension, scale: $0.85 - 1.15\times$) and random rotations ($\pm 15°$). This balanced and augmented dataset is used exclusively for training the AE and the binary (0 vs. 1) digit classifier. We found that having a balanced dataset for training AE would lead to a balanced latent space for MNIST Digit 0 and 1.

**Pre-training an MNIST Digit Classifier.** A convolutional neural network classifier is pre-trained to distinguish between MNIST digits 0 and 1. It is trained for 20 epochs on the balanced and augmented dataset of these two digits, using the Adam optimizer with a learning rate of $1 \times 10^{-3}$ and a Cross-Entropy loss function. This classifier achieves approximately 99.99% accuracy on a test set of unseen MNIST 0s and 1s and is subsequently used (with frozen weights) to evaluate the class labels of images generated by the main generative model.

The Classifier, $C : [0, 1]^{1 \times 28 \times 28} \rightarrow \mathbb{R}^2$, architecture is as follows:

- Input: $1 \times 28 \times 28$ (decoded image)
- $\texttt{Conv2d}(1, 32, \text{kernel\_size} = 3, \text{stride} = 1, \text{padding} = 1) \rightarrow \texttt{ReLU}$  $(32 \times 28 \times 28)$
- $\texttt{MaxPool2d}(\text{kernel\_size} = 2, \text{stride} = 2)$  $(32 \times 14 \times 14)$
- $\texttt{Conv2d}(32, 64, \text{kernel\_size} = 3, \text{stride} = 1, \text{padding} = 1) \rightarrow \texttt{ReLU}$  $(64 \times 14 \times 14)$

- `MaxPool2d`(kernel_size $= 2$, stride $= 2$)    ($64 \times 7 \times 7$)
- Flatten: $64 \times 7 \times 7 = 3136$ features
- `Linear`$(3136, 128) \rightarrow$ `ReLU`
- `Linear`$(128, $ num_classes $= 2)$ (Logits for classes 0 and 1)

**Constructing the Observed (Contaminated) Dataset $X_{\text{obs}}$** The observed dataset $X_{\text{obs}}$ for training the generator $G$ consists of latent representations. These are obtained by encoding MNIST images of digits 0 (target class) and 1 (outlier class) using the pre-trained AE's encoder. Specifically, $X_{\text{obs}}$ is a mixture comprising 90% samples from the true latent distribution of digit 0 ($\mathcal{X}_0$) and 10% samples (outliers) from the true latent distribution of digit 1 ($\mathcal{X}_1$). To construct this, we sample latent vectors $z'$ from the prior $\mathcal{U}[-1, 1]^2$, decode them to images $x' = AE_{\text{dec}}(z')$, and classify $x'$ using the pre-trained 0/1 classifier. If $x'$ is classified as 0 (or 1), $z'$ is added to a pool for $\mathcal{X}_0$ (or $\mathcal{X}_1$). We collect samples until we can form a dataset of $N_{\text{obs}} = 50,000$ latent points, with the 90/10 proportion. These latent points constitute $X_{\text{obs}}$ and are scaled to approximately reside within $[-1, 1]^2$.

**Training the Generator $G$.** The generator $G : \mathcal{N}(0, I_2) \rightarrow [-1, 1]^2$ is a multi-layer perceptron (MLP) designed to map 2D Gaussian noise $Z \sim \mathcal{N}(0, I_2)$ to the target latent space. The generator is trained by minimizing a (Partial) Optimal Transport distance $D(G(Z), X_{\text{obs}})$, where $Z$ is a batch of noise samples. Training is performed for 30 epochs using the Adam optimizer with a learning rate of $2 \times 10^{-4}$ and a batch size of 256. Specific (P)OT-based distances $D$ used for PartialTSW and baseline methods are detailed in the main paper.

The generator architecture is:

- Input: $Z \in \mathbb{R}^2 \sim \mathcal{N}(0, I_2)$
- `Linear`$(2, 4) \rightarrow$ `BatchNorm1d`$(4) \rightarrow$ `LeakyReLU`$(0.2)$
- `Linear`$(4, 8) \rightarrow$ `BatchNorm1d`$(8) \rightarrow$ `LeakyReLU`$(0.2)$
- `Linear`$(8, 2) \rightarrow$ `Tanh` (Output $z_{\text{gen}} \in [-1, 1]^2$)

**Evaluation.** To evaluate the generator's ability to learn the target distribution $\mathcal{X}_0$ while ignoring outliers from $\mathcal{X}_1$, we employ two main criteria:

1. **Outlier Rate:** We generate $N_{\text{eval}} = 5,000$ latent samples $z_{\text{gen}} = G(Z)$. These latent samples are decoded into images $\hat{x} = AE_{\text{dec}}(z_{\text{gen}})$ using the pre-trained AE's decoder. The resulting images are then classified by the pre-trained 0/1 digit classifier. The outlier rate is the percentage of generated images classified as digit 1. A lower rate indicates better robustness.

2. **Sample Quality and Diversity:** We qualitatively assess the generated samples by visualizing the decoded images $\hat{x}$ and their corresponding latent representations $z_{\text{gen}}$. We look for high-fidelity generation of digit 0 and good coverage of its variations, as indicated by a well-distributed latent space for the generated samples classified as 0.

Performance summaries, including outlier rates and visual comparisons, are provided in Figure 4 and Table 1 in the main text.

**Hardware Settings.** The experiments for all methods were conducted on a system equipped with an Intel Xeon Platinum 8580 CPU and one NVIDIA H100 GPU.

### E.8.2  Ablation result for baselines

We evaluate the impact of hyperparameter settings on each method's ability to isolate the target MNIST 0 distribution from the $10\%$ MNIST 1 outliers present in the training data. The following summarizes these ablation results (Figures 9–15), focusing on the percentage of generated MNIST 1 outliers and, where applicable, qualitative aspects of the learned distributions and generated samples.

**SPOT [9].** Figure 9 demonstrates SPOT's varying success in isolating the target MNIST 0 data from the $10\%$ 1 outliers, contingent on its hyperparameter $k$. While very small $k$ values (e.g., $k = 10$, yielding $45.62\%$ MNIST 1 outliers) or very large $k$ values (e.g., $k = 256$, yielding $16.20\%$ outliers) result in poor outlier rejection, an optimal range for $k$ around $200 - 210$ reduces the MNIST 1 outlier rate to $6 - 7\%$. This indicates substantial but incomplete removal of the $10\%$ outliers.

**SOPT [2].** SOPT's effectiveness in discarding the $10\%$ MNIST 1 outliers is modulated by its regularization parameter $\lambda$, as shown in Figure 10. The lowest outlier percentage achieved by SOPT is $13.28\%$ (at $\lambda = 0.01$), which still exceeds the initial $10\%$ contamination level. Larger values of $\lambda$ lead to even higher and relatively stable outlier rates (around $15 - 16.42\%$), indicating a persistent difficulty for SOPT in cleanly separating the target distribution in this setup.

**Sinkhorn [74].** Sinkhorn shows the potential for complete removal of the $10\%$ MNIST 1 outliers when its entropic regularization $reg$ and marginal regularization $reg_m$ are appropriately co-tuned (Figure 11). Specifically, setting $reg = reg_m$ at values of $0.5, 0.7$, or $0.9$ results in $0\%$ MNIST 1 outlier generation, successfully achieving the task's objective. However, imbalanced or overly small regularization values lead to substantial outlier contamination (e.g., $52.48\%$ for $reg = reg_m = 0.3$, or $70.56\%$ for $reg = 0.9, reg_m = 0.1$). Moreover, qualitative inspection of the results (Figure 11, particularly for $reg = reg_m \in \{0.5, 0.7, 0.9\}$) reveals that while Sinkhorn effectively removes outliers, the generated latent distribution for MNIST 0 digits appears clustered, and the corresponding decoded images may lack diversity compared to the true distribution. This suggests a potential trade-off between perfect outlier rejection and capturing the full diversity of the target class for this method under these settings.

**SUOT [7].** The performance of SUOT in the task of removing $10\%$ MNIST 1 outliers is consistently poor across the explored range of its marginal regularization parameters $\rho_1$ and $\rho_2$, as detailed in Figure 12. The method yields a high MNIST 1 outlier rate of approximately $41\%$ regardless of the hyperparameter settings tested, indicating a failure to distinguish the target MNIST 0 distribution from the contaminants.

**USOT [7].** USOT, while performing better than SUOT, still struggles to fully reject the $10\%$ MNIST 1 outliers (Figure 13). Across the tested range of its $\rho_1$ and $\rho_2$ hyperparameters, USOT yields a consistent MNIST 1 outlier rate of approximately $17.08\%$. This suggests that while it mitigates some contamination, it does not fully isolate the target MNIST 0 distribution in this scenario.

**PAWL [13].** PAWL demonstrates exceptional success in the goal of removing $10\%$ MNIST 1 outliers, as shown in its ablation study (Figure 14). It consistently achieves a $0\%$ MNIST 1 outlier rate across all tested values of its hyperparameter $k$ (from 10 to 256). This indicates PAWL's strong capability to identify and learn the target MNIST 0 distribution while completely ignoring outliers, exhibiting robust performance across a wide range of $k$. However, as noted in the main text and suggested by qualitative inspection of Figure 14, this strong outlier rejection by PAWL may be accompanied by less sample diversity, with its learned latent space showing heavily concentrated clusters.

**PartialTSW (Ours).** Our PartialTSW method shows strong capabilities in removing the $10\%$ MNIST 1 outliers, with performance critically depending on its mass parameter $\nu(\mathcal{T})$ (Figure 15). Complete outlier rejection ($0\%$ MNIST 1 outliers) is achieved for $\nu(\mathcal{T})$ values between $0.3$ and $0.6$. Setting $\nu(\mathcal{T})$ closer to the true inlier fraction of $0.9$ (which yields $9.72\%$ MNIST 1 outliers) leads to the model fitting the outliers. This highlights that optimal robustness for PartialTSW is achieved when $\nu(\mathcal{T})$ is chosen to be somewhat less than the actual inlier data proportion in the contaminated dataset. Qualitatively, as seen in Figure 15 and highlighted in our main findings, the settings achieving complete outlier rejection (e.g., $\nu(\mathcal{T}) \in [0.3, 0.6]$) also yield a well-distributed latent space and diverse image samples for the MNIST 0 class, effectively capturing the target distribution.

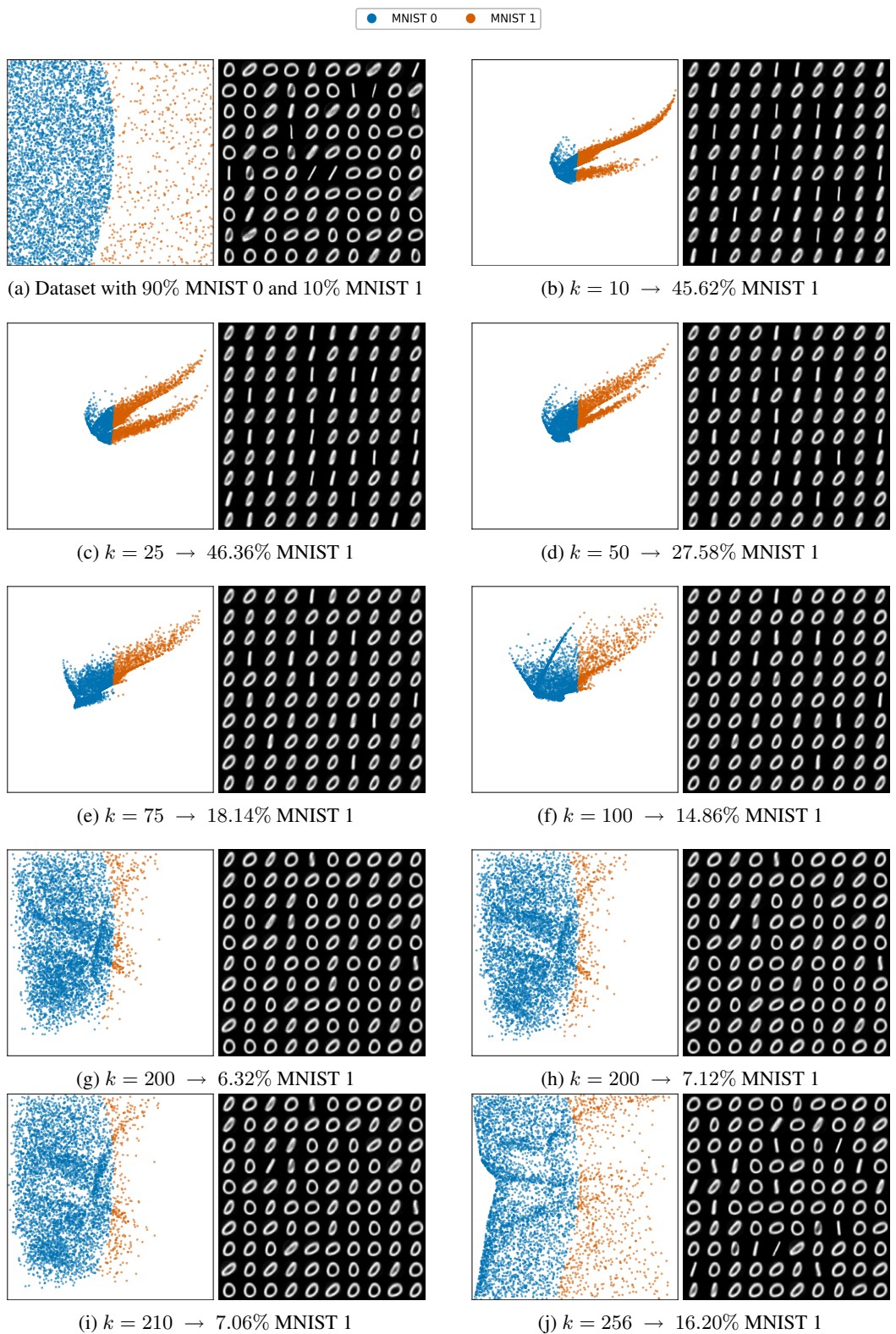

Figure 9: Ablation study of SPOT [9] for robust image generation. The figure illustrates the percentage of generated MNIST 1 digits (outliers), along with the corresponding learned latent distributions (left) and decoded image samples (right).

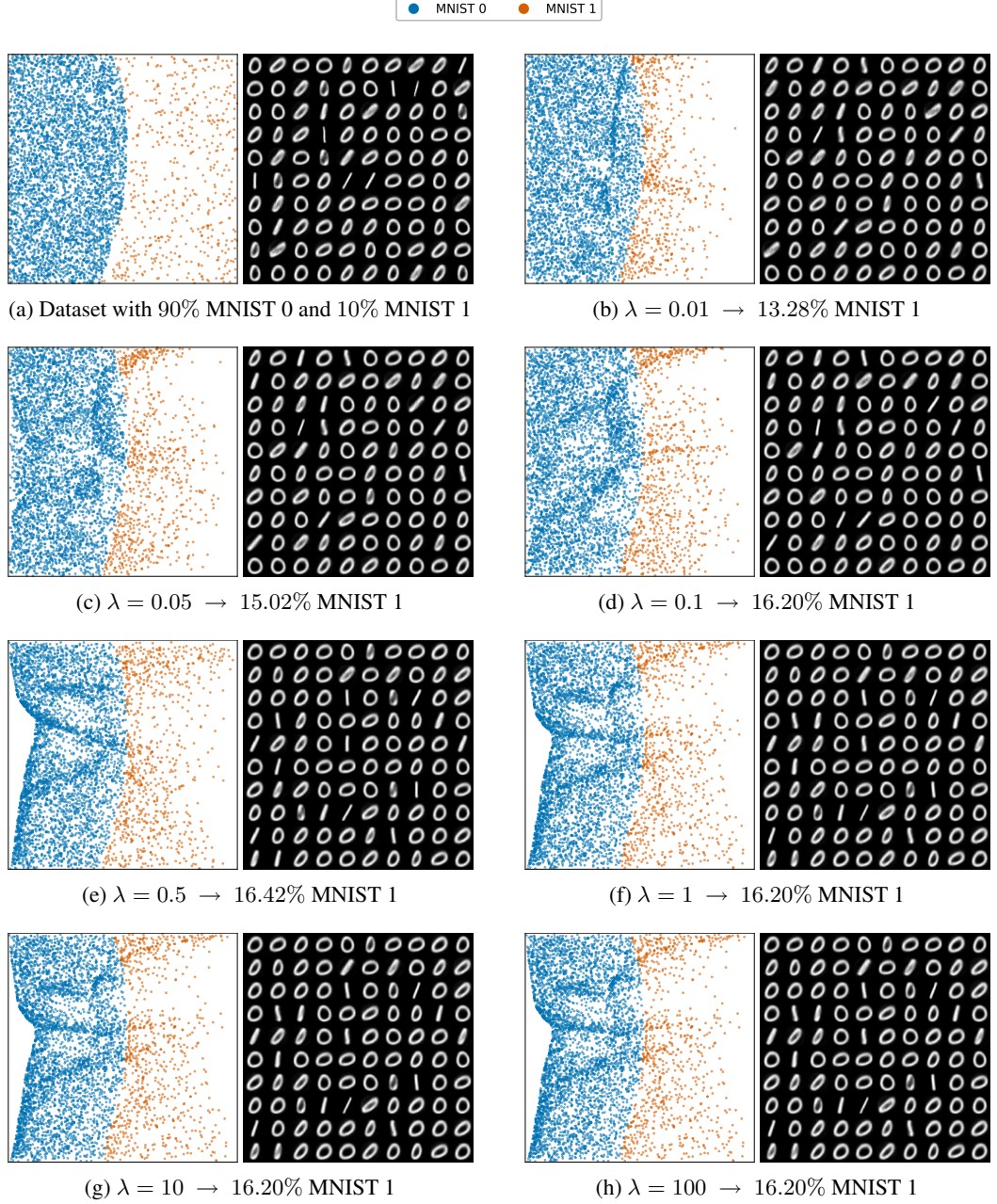

Figure 10: Ablation study of SOPT [2] for robust image generation. The figure illustrates the percentage of generated MNIST 1 digits (outliers), along with the corresponding learned latent distributions (left) and decoded image samples (right).

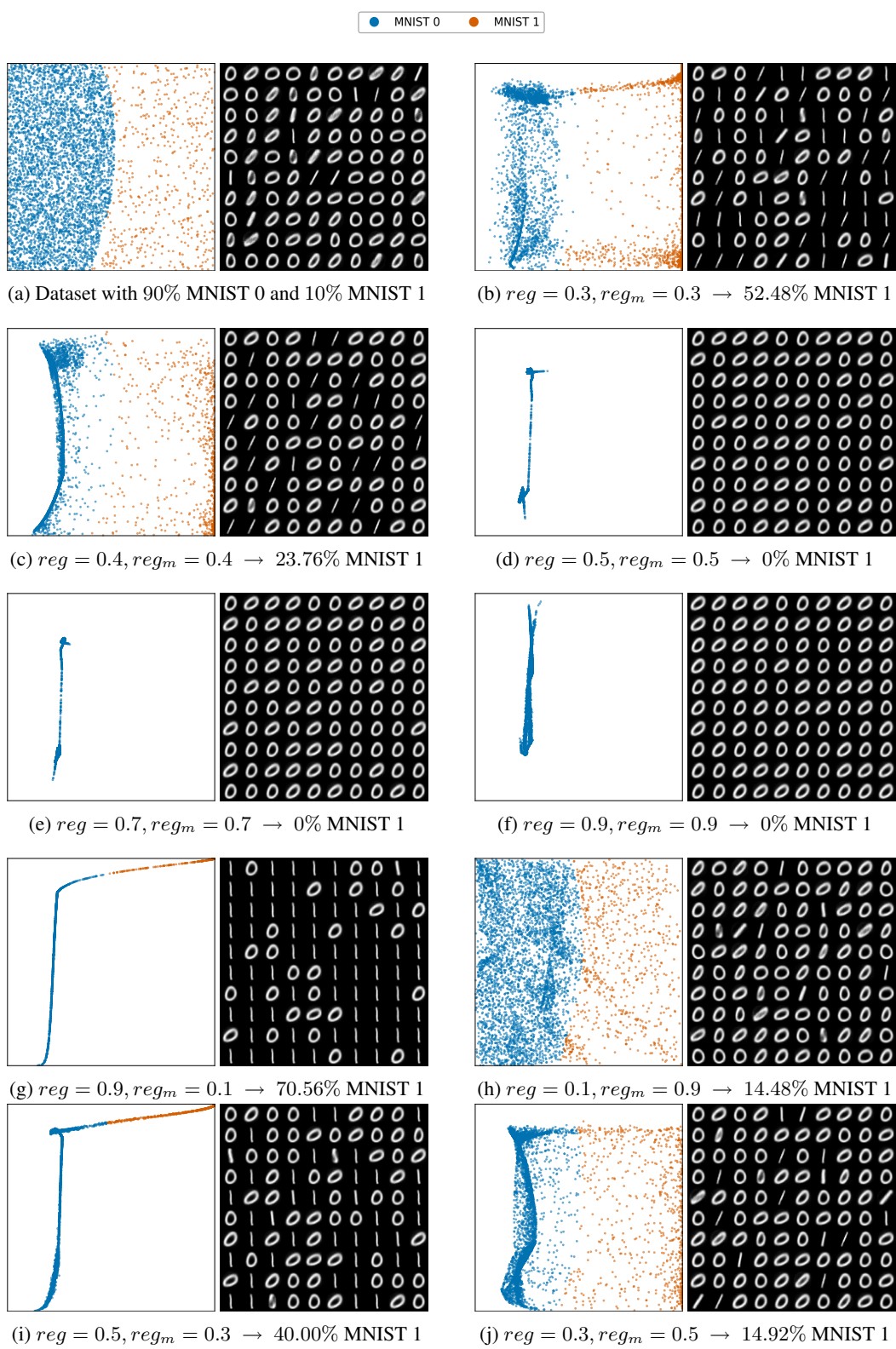

Figure 11: Ablation study of Sinkhorn [74] for robust image generation. The figure illustrates the percentage of generated MNIST 1 digits (outliers), along with the corresponding learned latent distributions (left) and decoded image samples (right).

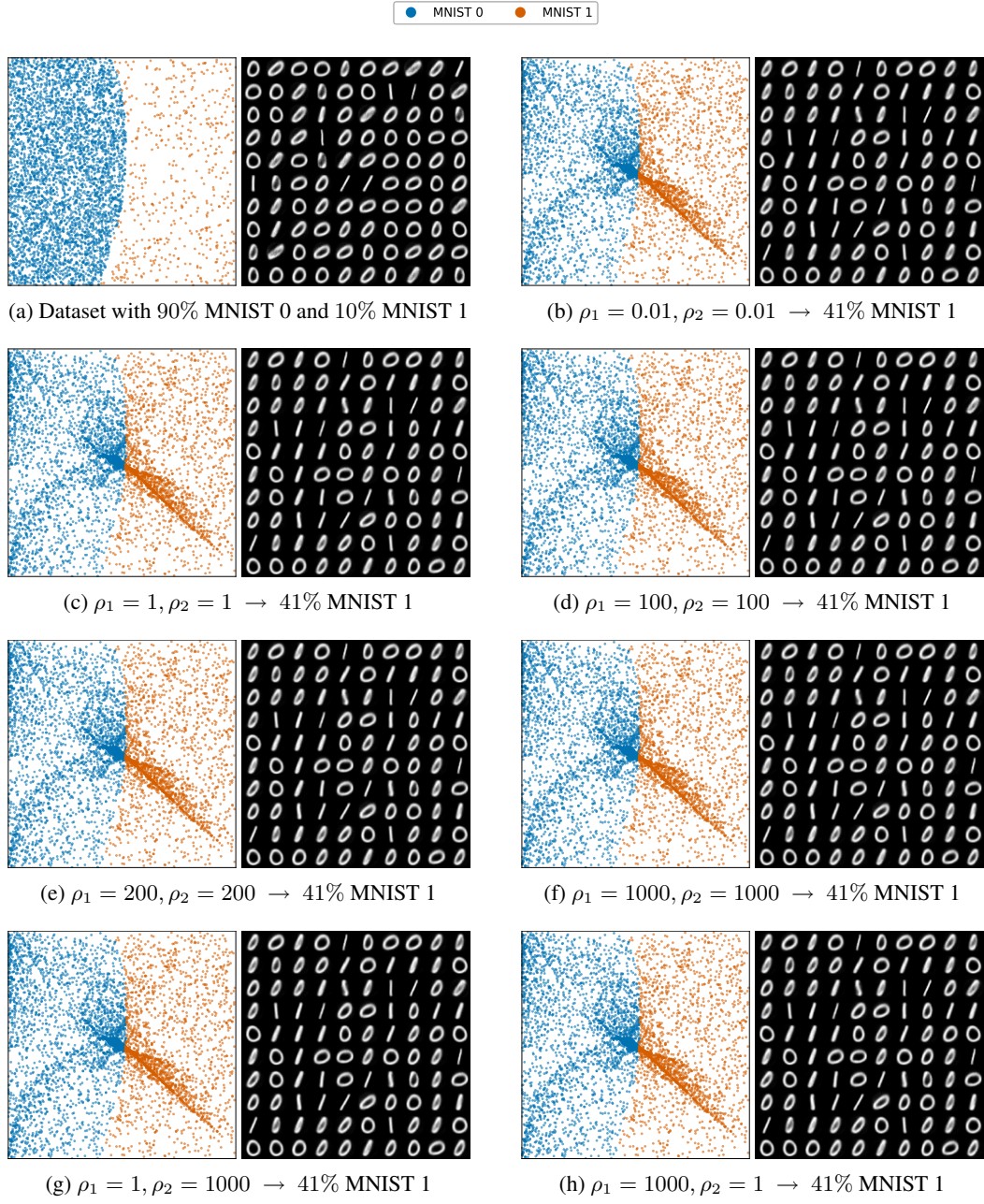

Figure 12: Ablation study of SUOT [7] for robust image generation. The figure illustrates the percentage of generated MNIST 1 digits (outliers), along with the corresponding learned latent distributions (left) and decoded image samples (right).

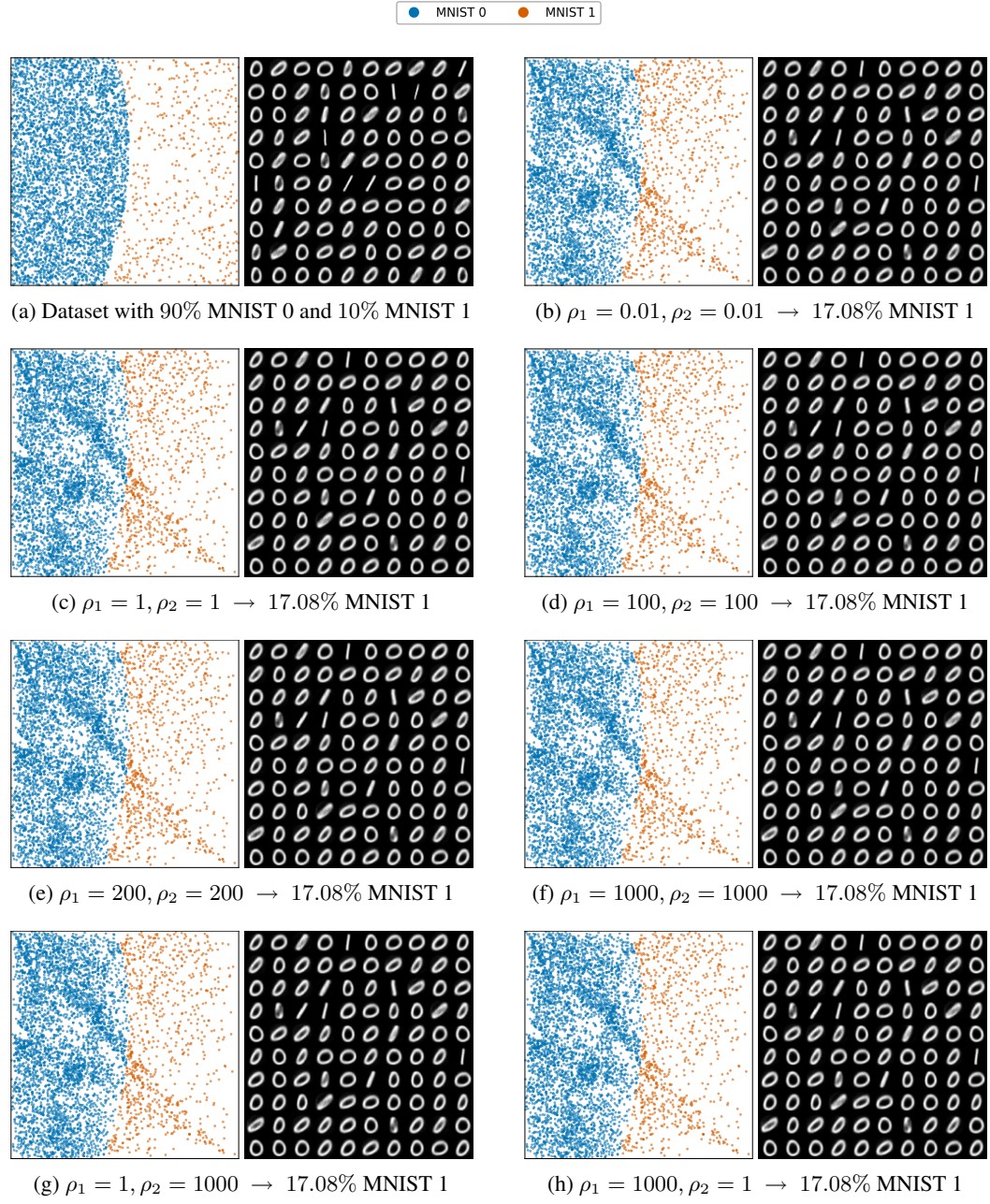

Figure 13: Ablation study of USOT [7] for robust image generation. The figure illustrates the percentage of generated MNIST 1 digits (outliers), along with the corresponding learned latent distributions (left) and decoded image samples (right).

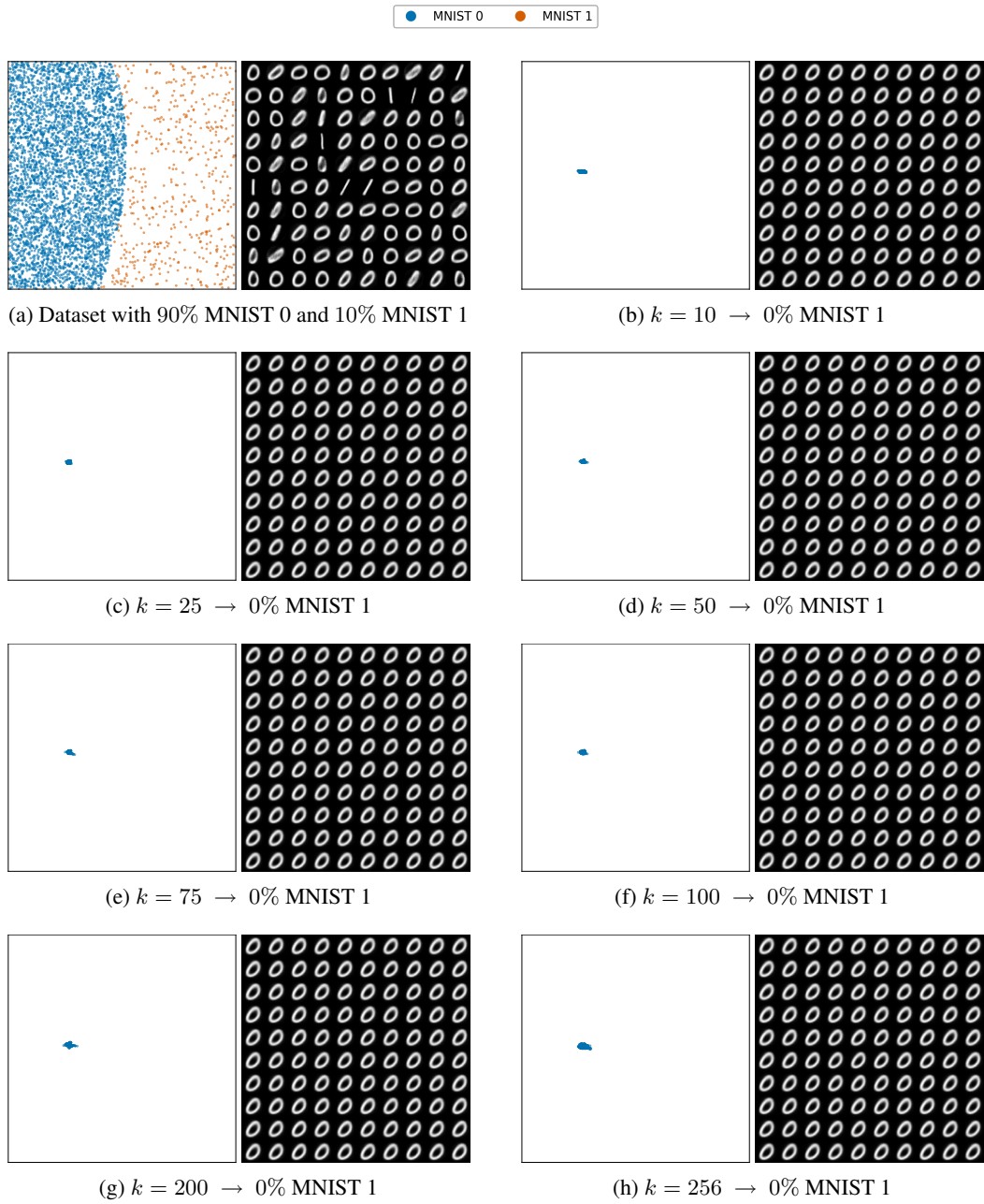

Figure 14: Ablation study of PAWL [13] for robust image generation. The figure illustrates the percentage of generated MNIST 1 digits (outliers), along with the corresponding learned latent distributions (left) and decoded image samples (right).

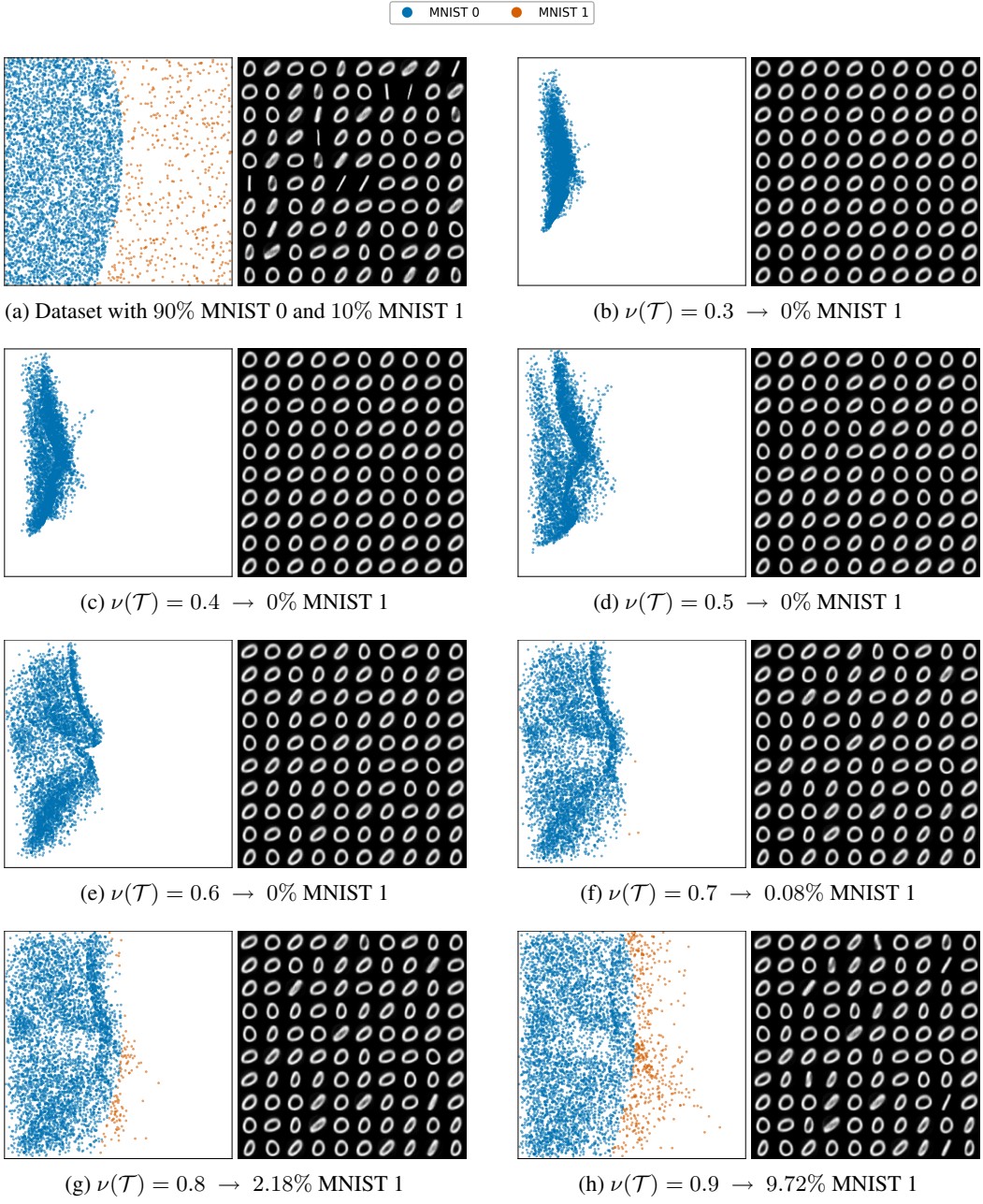

Figure 15: Ablation study of Partial-TSW (Ours) for robust image generation. The figure illustrates the percentage of generated MNIST 1 digits (outliers), along with the corresponding learned latent distributions (left) and decoded image samples (right).

### E.9 Imbalance Image to Image Translation

#### E.9.1 Implementation detail

This section outlines the experimental setup for the imbalanced image-to-image translation task, specifically converting "Young" faces to "Adult" faces.

**Dataset.** Our experimental dataset and preprocessing follow [41]. We utilize the FFHQ dataset [37] of $1024 \times 1024$ images. These images are encoded into a 512-dimensional latent space using a pre-trained ALAE autoencoder [64]. The resulting latent representations are categorized into two classes: "Young" and "Adult", based on a cutoff age of 45 years. This process yields an imbalanced dataset comprising approximately 38,000 "Young" latent vectors and 10,500 "Adult" latent vectors. The translation is performed within this 512-dimensional latent space.

**Translation Accuracy.** To evaluate the accuracy of the translation from the "Young" to the "Adult" domain, we adapt the procedure from [28]. A classifier is pre-trained on the 512-dimensional latent vectors to distinguish between "Young" and "Adult" images. This pre-trained classifier, which achieves 99% accuracy on a held-out test set of latent vectors, is then used to assess whether the translated latent vectors $M(X)$ (where $X$ are latents from the "Young" domain) are correctly classified as "Adult".

**Perceptual Similarity.** We measure the perceptual similarity between the original images (reconstructed from $X$) and the translated images (reconstructed from $M(X)$) using the Learned Perceptual Image Patch Similarity (LPIPS) metric [90]. For LPIPS calculations, we use the AlexNet backbone with pre-trained weights. While some prior work, such as [28], employed attribute-specific metrics like "Keep Accuracy" (e.g., for preserving gender), we selected LPIPS to offer a more comprehensive assessment of overall visual fidelity post-translation, rather than focusing on a single attribute.

**UOT-FM Baseline.** We compare against Unbalanced Optimal Transport Flow Matching (UOT-FM) [22]. Following [28], we parameterize vector field $v_\theta$ using a 2-layer feed-forward network with 512 hidden neurons and ReLU activation. We apply their default configuration for Flow Matching. Consistent with the approach in [28], we perform an ablation study over the regularization parameter $\lambda$, which controls the penalization of deviations from marginal constraints.

**ULightOT Baseline.** We also include ULightOT [28] as a baseline. We adapted the publicly available code and default models for our experiments. Following the methodology in [28], we ablate the parameter $\tau$, which governs the degree of mass conservation in the transport plan. Our empirical observations indicate that the performance of ULightOT saturates for $\tau > 1000$. For instance, increasing $\tau$ to 10000 yielded negligible changes in the Accuracy-LPIPS trade-off compared to $\tau = 1000$, as shown by the results (e.g., in Figure 5 of the main text).

**Mapping Network Architecture.**

For methods such as SW, Db-TSW, and our PartialTSW, the mapping network $M$ is implemented using a `ResidualMLP`. The input to this network is a latent vector $z \in \mathbb{R}^{512}$. The specific `ResidualMLP` configuration used has an input/output dimension of $512$, with `num_hidden_blocks=0` and `hidden_dim_multiplier=1`.

The core of this network, denoted as $\mathrm{MLP}_{\mathrm{core}}$, processes the input $z$ through the following sequence of operations:

- Apply an initial linear transformation: `Linear`$(512, 512)$
- Followed by layer normalization: `LayerNorm`$(512)$
- Then, apply the GELU activation function: `GELU`$()$
- Apply dropout with a rate of $0.1$: `Dropout`$(0.1)$
- Finally, apply an output linear projection: `Linear`$(512, 512)$

Let the output of this sequential $\mathrm{MLP}_{\mathrm{core}}$ block be $\mathrm{MLP}_{\mathrm{core}}(z)$.

The final output of the mapping network $M(z)$ is obtained by adding a scaled residual connection to the original input:

$$M(z) = z + \alpha \cdot \mathrm{MLP}_{\mathrm{core}}(z)$$

where $\alpha$ is a learnable scalar parameter (analogous to LayerScale) that is initialized to $0.1$.

Table 5: Comparison of model size and training time per epoch for the Young-to-Adult translation.

| Method | Number of Parameters | Time (s/epoch) |
|---|---|---|
| SW | 526,337 | 6 |
| Db-TSW | 526,337 | 25 |
| **PartialTSW (Ours)** | **526,337** | **25** |
| UOT-FM | 788,224 | 35 |
| ULightOT | 5,263,380 | 60 |

Table 6: Full image-to-image translation results, averaged over 5 runs. Our method, PartialTSW, consistently demonstrates a superior trade-off between Accuracy ($\uparrow$) and LPIPS ($\downarrow$) across all translation directions.

| Method | Parameter | W→M | | M→W | | A→Y | |
|---|---|---|---|---|---|---|---|
| | | Acc (%) $\uparrow$ | LPIPS $\downarrow$ | Acc (%) $\uparrow$ | LPIPS $\downarrow$ | Acc (%) $\uparrow$ | LPIPS $\downarrow$ |
| **SW** | – | 93.95 | 0.4418 | 91.68 | 0.4546 | 89.82 | 0.4041 |
| **Db-TSW** | – | 94.13 | 0.4436 | 92.07 | 0.4546 | 89.58 | 0.4022 |
| **UOT-FM** | $\epsilon = 0.0005$ | 49.21 | 0.3914 | 85.52 | 0.4269 | 79.77 | 0.3836 |
| | $\epsilon = 0.005$ | 70.69 | 0.4531 | 94.19 | 0.4749 | 92.91 | 0.4129 |
| | $\epsilon = 0.05$ | 80.28 | 0.4899 | 95.40 | 0.5106 | 97.40 | 0.4693 |
| | $\epsilon = 0.1$ | 81.50 | 0.5198 | 97.91 | 0.5369 | 98.43 | 0.4828 |
| **ULightOT** | $\tau = 50.0$ | 76.36 | 0.4102 | 85.91 | 0.4086 | 84.31 | 0.3452 |
| | $\tau = 250.0$ | 86.36 | 0.4466 | 92.81 | 0.4516 | 90.49 | 0.3906 |
| | $\tau = 1000.0$ | 88.07 | 0.4557 | 93.91 | 0.4626 | 92.00 | 0.4060 |
| | $\tau = 10000.0$ | 88.75 | 0.4589 | 94.37 | 0.4663 | 92.49 | 0.4112 |
| **PartialTSW (Ours)** | $\nu(\mathcal{T}) = 0.3$ | 99.66 | 0.6058 | 99.13 | 0.6088 | 97.64 | 0.5595 |
| | $\nu(\mathcal{T}) = 0.5$ | 98.04 | 0.5377 | 95.36 | 0.5493 | 93.76 | 0.4928 |
| | $\nu(\mathcal{T}) = 0.9$ | 95.67 | 0.4515 | 94.39 | 0.4682 | 91.16 | 0.4024 |
| | $\nu(\mathcal{T}) = 1.1$ | 92.03 | 0.4408 | 90.62 | 0.4533 | 89.34 | 0.4011 |

### E.9.2 Additional Experimental Results

This section provides further details on model parameterization and presents a complete set of results for all image-to-image translation directions.

**Parameterization and Efficiency.** As detailed in the implementation section, we adhered to the official configurations for the baseline methods. We utilized a Gaussian Mixture Model (GMM) for ULightOT [28] and a flow-matching network for UOT-FM [22], ensuring a faithful and robust comparison.

For our method (PartialTSW) and the other standard OT baselines (SW [10], Db-TSW [80]), we employed the `ResidualMLP` architecture described in Appendix E.9. While a GMM parameterization is theoretically feasible for our method, we chose the neural network as it represents a more common, flexible, and standard approach for generative modeling tasks in recent literature.

This choice of parameterization is not only standard but also highly efficient. As shown in Table 5, our `ResidualMLP` approach is significantly more lightweight and faster per epoch than the complex models required by UOT-FM and ULightOT. This demonstrates that PartialTSW is not only effective but also computationally efficient.

**Results for All Translation Directions.** In the main paper, our analysis centered on the Young-to-Adult (Y→A) translation task. This direction was chosen as it represents the most significant class imbalance within the dataset. The class distribution of the pre-processed FFHQ latent dataset is as follows: Young (approximately 15K Man, 23K Woman) and Adult (approximately 7K Man, 3.5K Woman).

To provide a comprehensive analysis, we conducted additional experiments for all other possible translation directions: Woman-to-Man (W→M), Man-to-Woman (M→W), and Adult-to-Young (A→Y). The results, averaged over 5 independent runs, are presented in Table 6.

These findings confirm that Partial-TSW consistently achieves a superior trade-off between translation accuracy and perceptual similarity (LPIPS) across all translation settings. For instance, in the W→M setting, Partial-TSW (with $\nu(\mathcal{T}) = 0.9$) achieves a high accuracy of $95.67\%$ while maintaining a strong LPIPS of $0.4515$. In contrast, for UOT-FM to reach a competitive accuracy (e.g., $81.50\%$ with $\epsilon = 0.1$), it incurs a significantly worse LPIPS of $0.5198$. This pattern, visible across the tasks, highlights our method's ability to find a more effective and stable balance between the two competing objectives.

## F   Boarder Impacts

The introduction of the PartialTSW in this paper has a substantial societal impact by enhancing the precision and adaptability of optimal transport methods in various practical applications. This method can drive progress in numerous fields, such as healthcare, where better image processing techniques can aid in more accurate medical imaging diagnostics, or in the arts and entertainment industry, where enhanced generative models can lead to more sophisticated and creative outputs. Furthermore, the ability to handle dynamic settings efficiently opens new possibilities for real-time data analysis and decision-making in various sectors, including finance, logistics, and environmental monitoring. Ultimately, the method contributes to making advanced computational techniques more versatile and applicable to a broader range of real-world problems, thereby fostering innovation and improving societal well-being.

