# OpenReview forum: "Tree-Sliced Entropy Partial Transport"
_NeurIPS.cc/2025/Conference — NeurIPS 2025 poster_

### Official Review · Reviewer_d8LY · 2025-07-02

**Clarity:** 3
**Significance:** 2
**Originality:** 2
**Rating:** 4
**Confidence:** 4

**Summary:**

The paper introduces Tree-Sliced Entropy Partial Transport (PartialTSW), obtained by averaging an entropy–regularised partial transport cost over sets of random tree metrics. It propose to compute the partial transport problem on a tree as a balanced OT problem between modified measures. The distance is shown to satisfy the metric proprieties and inherits rotation-invariance from the random‐tree construction. Experiments includes synthetic point-cloud interpolation with noise, image latent-space auto-encoding and class-imbalanced image translation, reporting speed-ups over Sinkhorn-based unbalanced OT and robustness to added outliers.

**Questions:**

Without a quantitative cost–accuracy curve the runtime tables (Fig. 2, Table 1) risk comparing methods at different error tolerances. Could
 you please either provide (i) a variance/concentration bound for the Monte-Carlo estimator and a bias analysis for finite 𝑘, or (ii) empirical convergence plots so readers can see how many projections are needed for stable distances.

Could you include the partial-transport baselines (SUOT, SPOT, USOT, PAWL) in the point-cloud and generative experiments, with the same transported-mass budget ν(T)?

Is there any possible to add an outlier-robustness theorem or, at minimum, systematic contamination sweeps that hold ν(T) fixed across methods? Without such evidence, the robustness claim is week.

Could you supply guidance for choosing L and k? An empirical plot of the estimation error against Lk would enable users to trade accuracy for speed.

**Ethical Concerns:**

["NO or VERY MINOR ethics concerns only"]

**Final Justification:**

I have reviewed the response and believed they addressed my concerns. Raised my score to 4.

**Quality:**

3

**Strengths And Weaknesses:**

Quality: The algebraic reduction from partial to balanced OT on a tree and proofs of metric properties looks good to me. Implement use cases are enough. But the numerical accuracy is not controlled across baselines; runtime comparisons therefore lack a common error tolerance. No variance or convergence analysis is provided for the Monte-Carlo estimator.

Clarity: The manuscript is generally well written. But the role and tuning of the transported-mass parameter ν(T) are only discussed heuristically.

Significance: The O(nlogn) time complexity is appealing by matching the complexity of balanced TSW, and runs faster and appears more robust than the rest of baselines in experiments. However, without cost–accuracy curves or formal robustness lemma the empirical impact is uncertain.

Originality: Extending TSW to the partial setting is incremental. I think the novel step is the balanced-reduction identity on trees; all other components (random-tree averaging, robustness via partial mass) follow from prior work on TSW and partial OT.

Overall, the work lacks both a theoretical error analysis and decisive empirical evidence that PartialTSW is more robust or faster at equal accuracy than existing partial or unbalanced OT solvers. I therefore lean to reject.

---

> ### Author Rebuttal · Authors · 2025-07-31
>
> We address the concerns raised by the Reviewer in the **Weaknesses** and **Questions** sections as follows.
>
> ---
>
> **W1. The role and tuning ... discussed heuristically.**
>
> **Answer W1**. We argue that tuning the transported-mass parameter $\nu(\mathcal{T})$ is a key practical advantage of our method due to its interpretability and principled range.
>
> Unlike baseline methods where partiality is controlled by an abstract regularization term, $\nu(\mathcal{T})$ directly corresponds to the **fraction of mass to be transported**. This allows for an intuitive, data-driven tuning process. For example, in a setting with an estimated 20% outlier or noise ratio, a principled starting point for tuning is $\nu(\mathcal{T}) \approx 0.8$.
>
> Furthermore, this provides a constrained and interpretable search space for the hyperparameter, typically within $(0, 1]$. This contrasts sharply with methods like ULightOT [1] or UOT-FM [2], where the equivalent parameters ($\tau$ or $\epsilon$) are not directly interpretable and must be tuned across several orders of magnitude to achieve meaningful differences in partiality, as shown in our experiments (Table 2).
>
> **Q1. Without a quantitative cost–accuracy ... how many projections are needed for stable distances.**
>
> **W2. The $O(n \log n)$ time complexity ... empirical impact is uncertain.**
>
> **Answer Q1+W2**. We thank the reviewer for raising this crucial point about the accuracy-cost trade-off. We agree that runtime comparisons are only meaningful at a similar error tolerance.
>
> To address this, we first clarify the roles of our method's parameters. For a number of lines ($k$), a ground-truth distance $PartialTSW(\mu, \nu)$ is defined. Our method then uses a **Monte Carlo estimation over the number of random trees ($L$)** to approximate this distance. The reviewer's concern about the stability of this estimator is well-founded.
>
> As requested, we provide an empirical analysis of this estimator's convergence. The table below shows the **Coefficient of Variation (CoV = $\sigma/\mu$)**, a normalized measure of stability, as a function of both $L$ (the Monte Carlo parameter) and $k$ (a structural parameter).
>
> |$L$|$k=5$|$k=10$|$k=100$|
> |:-|:-:|:-:|:-:|
> |10|0.1098|0.1016|0.0456|
> |50 | 0.0526 | 0.0252|0.0183|
> |100|0.0263|0.0239|0.0091|
> |500|0.0105 |0.0133|0.0070|
> |1000|0.0076|0.0070|0.0041|
> |10000|0.0029| 0.0018|0.0023|
> |20000|0.0016|0.0024|0.0010|
>
> The results empirically verify that for any fixed $k$, the estimator's stability improves as $L$ increases, aligning with the theoretical $O(L^{-1/2})$ Monte Carlo convergence rate. For instance, at $k=5$, increasing $L$ from 10 to 1000 reduces the CoV by over 14x. In our large-scale experiments, we use a high number of projections ($L \ge 1000$) to ensure a stable estimate. We will add this analysis to the appendix to guide practitioners.
>
> For the runtime comparisons in Figure 2 and Table 1, we ensured a fair comparison across all sliced-based methods. We used an identical number of projections ($L$) for all methods. Additionally, for the tree-based methods (TSW and our PartialTSW), the tree complexity parameter ($k$) was held constant.
>
> More importantly, your question highlights a key advantage of our method. The computational complexity of PartialTSW is **independent** of its robustness parameter, $\nu(\mathcal{T})$. In contrast, the complexity of baselines like PAWL [3] is directly tied to their robustness parameter (the number of samples $k$ to transport). This means that for such baselines, increasing robustness inherently increases computational cost. PartialTSW does not face this trade-off. The fact that it is both highly robust and empirically among the fastest methods underscores its practical significance.
>
> **W3. Extending TSW to the partial setting is incremental. I think the novel step is the balanced-reduction identity on trees; all other components (random-tree averaging, robustness via partial mass) follow from prior work on TSW and partial OT.**
>
> **Answer W3**. We would like to clarify that we view our extension of TSW to the partial setting as a substantive contribution, for the following reasons:
>
> - The Entropy Partial Transport (EPT) problem on tree metric spaces has been carefully studied, primarily for its computational advantages in large-scale settings. However, prior applications have been limited to distributions supported on graph- or tree-structured spaces, which significantly restricts their applicability to broader settings involving Euclidean measures or generative tasks. Extending EPT to Euclidean measures through the TSW framework requires a new conceptual bridge, insight that, to the best of our knowledge, has not been previously established.
>
> - Our work provides both theoretical and experimental contributions that detail the adoption of the EPT formulation within the TSW framework for Euclidean data. This includes the derivation of a valid metric between measures, $\text{E}(d)$-invariance, supported by closed-form approximations and efficient implementations. All theoretical claims are accompanied by rigorous proofs.
>
> - While prior works have applied unbalanced or partial transport to sliced variants of Wasserstein distances (see [4], [5]), the EPT problem on tree metrics introduces unique structural requirements, specifically, the need to augment the support with an additional point (as noted in the Reviewer’s question). This structural modification cannot be accommodated within the standard sliced setting and instead necessitates the full TSW framework. We address this by proposing a closed-form approximation of our distance (see Equation (9)), which leads to significant computational advantages. Empirically, our method outperforms prior approaches in terms of runtime.
>
> For these reasons, we believe our contribution is substantive and non-incremental, offering both theoretical novelty and practical utility within the broader context of optimal transport on Euclidean domains.
>
> **Q2. Could you include ... same transported-mass budget $\nu({\mathcal{T}})$?**
>
> **Q3. Is there any possibility ... robustness claim is weak.**
>
> **Answer Q2+Q3.** Regarding **Q2**, we agree that comparing methods with a matched transported-mass budget is ideal. However, this is often not feasible due to fundamental differences in how the baselines are formulated. While our PartialTSW allows for a flexible, continuous mass budget via the $\nu(\mathcal{T})$ parameter, competing methods control partiality indirectly:
> * Methods like **SPOT** [4] and **PAWL** [3] rely on selecting an integer number of samples, $k$.
> * For **USOT** [5] and **SUOT** [5], while they can theoretically be re-weighted, we found their available implementations did not support this direct control in our experiments, necessitating that partiality be managed via their regularization parameters ($\rho_1, \rho_2$).
>
> Because a direct matching of the mass budget is not possible, we focused on finding the optimal robustness setting for each method.
>
> This directly addresses **Q3**. While we have not included a formal outlier-robustness theorem, we did perform a **systematic hyperparameter sweep for all baselines** (detailed in Appendix E.4 and E.7.2) to find the settings that best reject outliers.
>
> Our findings show that even at their optimal settings, competing methods struggle. Baselines like USOT and SUOT, despite extensive tuning of their regularization parameters, could not completely ignore outliers, demonstrating the limitations of their indirect control mechanism. While a recent method like PAWL can achieve strong outlier rejection, it is slower than our method. In contrast, our experiments demonstrate that PartialTSW provides a direct, efficient, and effective mechanism for handling outliers.
>
> **Q4. Could you supply guidance for choosing $L$ and $k$? ... trade accuracy for speed.**
>
> **Answer Q4.** As established in our answer to **Q1+W2**, the Monte Carlo estimation error is a function of the number of trees, $L$, while the computational complexity scales linearly with both $L$ and the number of lines per tree, $k$.
>
> Our key recommendation for a fixed computational budget $O(Lk)$ is to prioritize a **large number of trees (high $L$, low $k$)**. The reasoning is that the primary source of error is the Monte Carlo variance, which is most effectively reduced by increasing $L$. Our empirical analysis in the Coefficient of Variation table (in our response to **Q1**) supports this, showing that increasing $L$ consistently provides the largest and most stable reduction in estimation error.
>
> Based on our experiments, we offer the following practical guidance: for large-scale tasks like the ones presented in our paper, we found a low number of lines **$k \in [2, 10]$** combined with a large number of trees, **$L \in [1000, 10000]$**, provides the best trade-off between speed and stable performance. We will add this explicit guidance to the appendix.
>
> ---
>
> **References.**
>
> [1] Milena Gazdieva et al., Light Unbalanced Optimal Transport. NeurIPS 2024.
>
> [2] Luca Eyring et al., Unbalancedness in Neural Monge Maps Improves Unpaired Domain Translation. ICLR 2024.
>
> [3] Laetitia Chapel and Romain Tavenard, One for all and all for one: Efficient computation of partial Wasserstein distances on the line. ICLR 2025.
>
> [4] Nicolas Bonneel and David Coeurjolly, SPOT: Sliced Partial Optimal Transport. ACM Transactions on Graphics (TOG) 2019.
>
> [5] Clément Bonet, et al., Slicing Unbalanced Optimal Transport. TMLR 2024.
>
> ---
>
> We sincerely thank the reviewer for the constructive feedback and for identifying the typographical errors in our submission. We will make the necessary revisions to address these issues. If the provided clarifications are deemed satisfactory, we would be grateful if the reviewer could consider adjusting the evaluation accordingly. We remain at the reviewer’s disposal and would be pleased to address any additional concerns in the subsequent stage of the discussion.

---

> > ### Author Response · Authors · 2025-08-08
> >
> > Dear Reviewer,
> >
> > As we approach the final days of the discussion phase, we would like to kindly follow up regarding the concerns you raised during the review process. We sincerely hope that our responses have addressed your questions and clarified the key aspects of our work.
> >
> > If you find our clarifications satisfactory, we would be grateful if you could consider updating your evaluation to reflect this. Of course, if there remain any unresolved points or further questions, we would be more than happy to continue the discussion.
> >
> > We truly value the thoughtful feedback we have received throughout the review process. Engaging with experts across different areas has greatly contributed to strengthening our work, and we are thankful for the opportunity to benefit from your insights.
> >
> > Warm regards,
> >
> > The Authors

---

### Official Review · Reviewer_uZKD · 2025-07-02

**Clarity:** 4
**Significance:** 2
**Originality:** 2
**Rating:** 4
**Confidence:** 4

**Summary:**

Tree-based approach to compute Sliced Partial Optimal Transport.

**Questions:**

1) How is the parametrization of the proposed approach tool different from the comparison methods, such as ULIghtOT, which uses a mixture of Gaussians, not neural networks? This makes the comparison unfair. Can you parameterize your method using a mixture of Gaussians?

2) How do the results depend on the parameter k? Can we control this parameter while *building a tree*?

3) Does PartialTSW have lower transport plan costs or Wasserstein distances between the target and generated distributions than other unbalanced methods?

4) Why did the authors only consider young-to-adult translation for the image generation task? The same dataset could be used for adult-to-young, man-to-woman, and woman-to-man experiments without changing the setup.

5) More diverse datasets should be included. In which real-world scenarios could your approach be used?

**Ethical Concerns:**

["NO or VERY MINOR ethics concerns only"]

**Final Justification:**

My concerns are satisfied, rising my score.

**Limitations:**

Yes

**Paper Formatting Concerns:**

No paper formatting concerns.

**Quality:**

3

**Strengths And Weaknesses:**

**Strengths:** I would highlight the clarity of the paper as one of its strengths. It is well-written and logically structured, making it easy to follow. The approach of efficiently solving partial sliced optimal transport using tree-based priors is also novel and promising for easily computing unbalanced OT.  I appreciate the authors' experiments on the dependence of unbalanced/tau parameters, which are significant for unbalanced/partial settings. The proposed method clearly demonstrates stability and dependence on the parameters, and, in comparison to other OT approaches, it provides an interpretable value between 0 and 1.

**Weaknesses:** The only change from [73, 77] is the switch to an Unbalanced OT objective, which minimizes the methodological contribution. Aside from the theorem that PartialTSW is an E(d)-invariant metric on M(Rd), the theoretical contributions are limited. Besides the theoretical contribution, the practical justification is also limited. The authors provided several experiments. I think the approach needs a more expressive, high-dimensional justification of the proposed method. The robust generative modeling experiments are interesting, but they are very artificial (e.g., MNIST). and do not cover non-real-world scenarios This makes the justification for the proposed method unclear.

---

> ### Author Rebuttal · Authors · 2025-07-31
>
> We address the concerns raised by the Reviewer in the **Weaknesses** and **Questions** sections as follows.
>
> ---
>
> **W1. The only change ... minimizes the methodological contribution.**
>
> **W2. Aside from the theorem ... theoretical contributions are limited.**
>
> **Answer W1+W2.** We would like to clarify that we view our extension of TSW to the partial setting as a substantive contribution, for the following reasons:
>
> - The Entropy Partial Transport (EPT) problem on tree metric spaces has been carefully studied, primarily for its computational advantages in large-scale settings. However, prior applications have been limited to distributions supported on graph- or tree-structured spaces, which significantly restricts their applicability to broader settings involving Euclidean measures or generative tasks. Extending EPT to Euclidean measures through the Tree-Sliced Wasserstein (TSW) framework requires a new conceptual bridge—insight that, to the best of our knowledge, has not been previously established.
>
> - Our work provides both theoretical and experimental contributions that detail the adoption of the EPT formulation within the TSW framework for Euclidean data. This includes the derivation of a valid metric between measures, $\text{E}(d)$-invariance, supported by closed-form approximations and efficient implementations. All theoretical claims are accompanied by rigorous proofs.
>
> - While prior works have applied unbalanced or partial transport to sliced variants of Wasserstein distances (see [1], [2]), the EPT problem on tree metrics introduces unique structural requirements—specifically, the need to augment the support with an additional point (as noted in the Reviewer’s question). This structural modification cannot be accommodated within the standard sliced setting and instead necessitates the full TSW framework. We address this by proposing a closed-form approximation of our distance (see Equation (9)), which leads to significant computational advantages. Empirically, our method outperforms prior approaches in terms of runtime.
>
> For these reasons, we believe our contribution is substantive and non-incremental, offering both theoretical novelty and practical utility within the broader context of optimal transport on Euclidean domains.
>
> **W3. Besides the theoretical contribution, the practical justification ...justification for the proposed method unclear.**
>
> **Q5. More diverse datasets ...real-world scenarios could your approach be used?**
>
> **Answer W3+Q5.** We appreciate the reviewer's positive feedback on our generative modeling experiment. We designed this experiment on MNIST as a clear, illustrative study to visually demonstrate the qualitative advantages of our method's robustness. While this setup is simplified for clarity, the principle of using PartialTSW to build resilience against outlier data is a general one, applicable to more complex datasets and other modalities.
>
> For a high-dimensional justification, we presented an image-to-image translation task in Section 4.3 ($d=512$), adapted from ULightOT [5]. The strong performance of PartialTSW in this experiment, particularly its ability to handle significant class imbalance, directly demonstrates its effectiveness and scalability in a challenging, non-trivial setting.
>
> The applicability of PartialTSW extends beyond our presented experiments. As a robust and differentiable loss function, it is well-suited for tasks like domain adaptation [9] and self-supervised learning [10]. We note, however, that it is not designed for assignment-based problems (e.g., object detection [11]), as it does not compute an explicit transport plan.
>
> **Q1. How is the parametrization ... using a mixture of Gaussians?**
>
> We thank the reviewer for this important question regarding the parameterization in the image-to-image translation experiments.
>
> For the baselines, we faithfully followed the official settings from their respective papers, using a Gaussian Mixture Model for ULightOT [5] and a flow-matching network for UOT-FM [4], to ensure a strong and fair representation of their methods. These details are presented in Appendix E.8.
>
> For our method (PartialTSW) and the other OT-based baselines (SW, Db-TSW [6]), we used a standard neural network. While parameterizing our method with a GMM is theoretically possible, we opted for a neural network as it is a more common and flexible choice in the current literature for generative tasks.
>
> We respectfully argue that this comparison is not unfair to the baselines; in fact, our chosen parameterization is significantly more efficient. The following table summarizes the parameter counts and per-epoch training times:
>
> |Method|Number of Parameters|Time (s/epoch)|
> |:-|:-|:-|
> |SW|526,337|6|
> |Db-TSW|526,337|25|
> |**PartialTSW (Ours)**|**526,337**|**25**|
> |UOT-FM|788,224|35|
> |ULightOT|5,263,380|60|
>
> As shown, our approach achieves its results using fewer parameters and faster training time per epoch than ULightOT and UOT-FM. This demonstrates that our method is not only effective but also computationally efficient.
>
> **Q2.  How do the results ... control this parameter while building a tree?**
>
> The number of lines per tree, $k$, is a hyperparameter that is set during the tree construction process, which is detailed in Appendix C.1.
>
> Regarding its effect on performance, our empirical analysis shows that the best results are achieved by prioritizing a large number of trees ($L$) over a high tree complexity ($k$). Specifically, a low value for $k$ (e.g., $k \in [2, 10]$) provides the best trade-off between the stability of the distance estimate and overall computational speed. We provide a more detailed treatment of the interplay between $L$ and $k$ in our response to Q4 from Reviewer d8LY.
>
> **Q3. Does PartialTSW have lower transport plan costs or Wasserstein distances between the target and generated distributions than other unbalanced methods?**
>
> **Answer Q3.** We are not entirely certain about the motivation behind the question raised by the Reviewer. In our setting, PartialTSW—like other baselines—is used as a loss function to be minimized, not for direct magnitude comparison. Final evaluations are typically based on a universal metric, ideally from unbalanced OT.
>
> Therefore, comparing the absolute values of distances computed by PartialTSW and other baselines is not necessarily meaningful. In designing a distance (or discrepancy) between probability measures, the primary goal is to use it as a loss function in applications such as generative modeling or distribution alignment, where the objective is to minimize divergence between the model and empirical distributions.
>
> This objective is well-defined when the discrepancy is a true metric—i.e., it satisfies the identity of indiscernibles: $\text{discrepancy}(\mu,\nu) = 0$ if and only if $\mu =\nu$. Without this property, minimizing the loss may not guarantee distributional equivalence, making the optimization ambiguous. Hence, this property is essential for both theoretical validity and interpretability.
>
> Each proposed distance—including the baselines and our PartialTSW—induces a different loss landscape, varying in smoothness, optimization difficulty, and geometric alignment.  Empirically, PartialTSW yields a favorable loss landscape that leads to improved convergence and performance across several tasks.
>
> **Q4. Why did the authors ... without changing the setup.**
>
> **Answer Q4.** For the main paper, we focused on the Young-to-Adult (Y→A) translation task as it presents the most significant class imbalance in the dataset. The class distribution is as follows:
>
> ||Man|Woman|
> |:-|:-:|:-:|
> |**Young**|15K|23K|
> |**Adult**|7K|3.5K|
>
> As suggested by the Reviewer, we conducted additional experiments for all translation directions. The results, averaged over 5 runs, are summarized in the table below. They confirm that **Partial-TSW consistently achieves a better trade-off between task accuracy and perceptual similarity (LPIPS)** across all settings.
>
> For instance, in the Woman-to-Man (W→M) setting, Partial-TSW with $\nu(\mathcal{T})=0.9$ achieves a high accuracy of 95.67% while maintaining an excellent LPIPS score of 0.4515. In contrast, for UOT-FM to achieve a competitive accuracy, it requires a large regularization ($\epsilon=0.1$), which results in a significantly worse LPIPS of 0.5198. This highlights our method's ability to find a more effective balance.
>
> |Method||W$\rightarrow$M Acc (%) $\uparrow$|W$\rightarrow$M LPIPS $\downarrow$|M$\rightarrow$W Acc (%) $\uparrow$|M$\rightarrow$W LPIPS $\downarrow$|A$\rightarrow$Y Acc (%) $\uparrow$|A$\rightarrow$Y LPIPS $\downarrow$|
> |:-|:-|:-:|:-:|:-:|:-:|:-:|:-:|
> |**SW**||93.95|0.4418|91.68|0.4546|89.82|0.4041|
> |**Db-TSW**||94.13|0.4436|92.07|0.4546|89.58|0.4022|
> |**UOT-FM**||||||||
> ||$\epsilon=0.0005$|49.21|0.3914|85.52|0.4269|79.77|0.3836|
> ||$\epsilon=0.005$|70.69|0.4531|94.19|0.4749|92.91|0.4129|
> ||$\epsilon=0.05$|80.28|0.4899|95.40|0.5106|97.40|0.4693|
> ||$\epsilon=0.1$|81.50|0.5198|97.91|0.5369|98.43|0.4828|
> |**ULightOT**||||||||
> ||$\tau=50.0$|76.36|0.4102|85.91|0.4086|84.31|0.3452|
> ||$\tau=250.0$|86.36|0.4466|92.81|0.4516|90.49|0.3906|
> ||$\tau=1000.0$|88.07|0.4557|93.91|0.4626|92.00|0.4060|
> ||$\tau=10000.0$|88.75|0.4589|94.37|0.4663|92.49|0.4112|
> |**PartialTSW (Ours)**||||||||
> ||$\nu(\mathcal{T})=0.3$|99.66|0.6058|99.13|0.6088|97.64|0.5595|
> ||$\nu(\mathcal{T})=0.5$|98.04|0.5377|95.36|0.5493|93.76|0.4928|
> ||$\nu(\mathcal{T})=0.9$|95.67|0.4515|94.39|0.4682|91.16|0.4024|
> ||$\nu(\mathcal{T})=1.1$|92.03|0.4408|90.62|0.4533|89.34|0.4011|
>
> ---
>
> **References.** Due to space constraints, we refer the Reviewer to the references provided at the end of our response to **Reviewer fEkm**.
>
> ---
>
> We thank the Reviewer for the constructive feedback and will revise the manuscript accordingly. If our clarifications are satisfactory, we kindly hope the evaluation may be reconsidered. We remain available to address any further concerns in the next stage of the discussion.

---

> > ### Author Response · Authors · 2025-08-08
> >
> > Dear Reviewer,
> >
> > As we approach the final days of the discussion phase, we would like to kindly follow up regarding the concerns you raised during the review process. We sincerely hope that our responses have addressed your questions and clarified the key aspects of our work.
> >
> > If you find our clarifications satisfactory, we would be grateful if you could consider updating your evaluation to reflect this. Of course, if there remain any unresolved points or further questions, we would be more than happy to continue the discussion.
> >
> > We truly value the thoughtful feedback we have received throughout the review process. Engaging with experts across different areas has greatly contributed to strengthening our work, and we are thankful for the opportunity to benefit from your insights.
> >
> > Warm regards,
> >
> > The Authors

---

### Official Review · Reviewer_nitj · 2025-07-02

**Clarity:** 2
**Significance:** 2
**Originality:** 3
**Rating:** 3
**Confidence:** 1

**Summary:**

The paper considers approximations to optimal transport for measures on trees. More precisely, they consider a sliced approximation of the (possibly unbalanced) measures. Slicing is a popular way to reduce computational costs as computing optimal transport in 1D is often very easy. Partial transport relaxes the assumption that measures need to be probability measures and allows for unequal mass. This paper combines the two ideas to propose a tree-sliced partial optimal transport method.

**Questions:**

Questions:
(1) What is *the* tree metric $d_{\mathcal{T}}$? How does it relate to $\mathrm{d}$?
(2) I wasn't clear on what $\alpha$ is. Unless I missed something, it first gets used just above eq (10) but isn't defined until after eq (13). And even then I wasn't sure what it means. Could you add more detail?
(3) In Section 2.2, the paragraph on Tree System finishes with specifying the distribution $\sigma$. It's just stated that this is a "random sampling procedure", is it uniform? Or any probability distribution over lines?
(4) I didn't find the sentence "we then compute the regularized transport cost $d_a(\mu_{\mathcal{T}},\nu_{\mathcal{T}})$ as in Equation (10)" very clear. Eq (10) defines what I understand to be a 1-D projection. I feel there is a step missing in the logic. Could you add more detail.
(5) Where does "entropy" in the name come from?

Minor typo: Eq. (1) is the p-Wasserstein distance (not the 1-Wasserstein distance) as is written above.

**Ethical Concerns:**

["NO or VERY MINOR ethics concerns only"]

**Final Justification:**

This was the hardest of all the papers I had to review to make a judgement on. I suspect that from the slow responses of the other referees that their experience may be similar. And I understand how frustrating this is to the authors (and the AC). But it's a very long paper which I couldn't go through in detail in the time span available. I did go back through Appendix A and C which answered some of my questions. And it's perhaps reasonable that my other misunderstandings would be satisfied by going through everything else. I do accept the authors last response to my comments. However, without a better understanding I'm not comfortable increasing my score (and I'm not sure I think conferences are the best venue for long papers anyway), however I did decrease my confidence rating.

**Limitations:**

There are a couple of sentences at the end of the paper.

**Paper Formatting Concerns:**

None.

**Quality:**

2

**Strengths And Weaknesses:**

Combining two ideas to produce new methods is popular in optimal transport where the geometry is often desirable but the computational costs and assumptions on the inputs (i.e. probability measures) can be prohibitive. So this is a nice idea that brings together two things, partial and slicing, to tree-based optimal transport distances.

The numerical results appear good. Although I wasn't sure why more comparisons weren't included in Section 4.1 (for example the methods in Section 4.2). It also wasn't clear how the tree graphs were constructed in the experimental section.

---

> ### Author Rebuttal · Authors · 2025-07-28
>
> **General Response.** Based on the reviewer’s question, we believe there may be a lack of familiarity with the background of the Tree-Sliced Wasserstein (TSW) distance, as introduced in [1, 2]. In our submission, we provide a summary of the essential background in Section 2.2, with further technical details available in Appendices A and C. We offer here a brief introduction to the framework and encourage the reviewer to consult the aforementioned appendices for a deeper understanding.
>
> The Tree-Sliced Wasserstein distance [2] was proposed as an extension of the traditional Sliced Wasserstein (SW) framework, where the projection domain, typically a line, is replaced by a more expressive structure known as a tree system. This enhancement has been empirically shown to capture richer structural information than a single line, while still admitting a closed-form approximation that allows for efficient computation.
>
> A tree system in $\mathbb{R}^d$, roughly speaking, is a collection of multiple lines in $\mathbb{R}^d$ that are connected through a tree structure. This configuration induces a tree-metric space, on which the optimal transport problem admits closed-form solutions, as detailed in Section 2.1 and Appendix A.
>
> Within this tree-metric space, the TSW framework differs fundamentally from the SW approach. While SW projects the entire mass at each support point of a distribution onto a single line, TSW distributes the mass at each support point into $k$ parts, each assigned to one of the $k$ lines in the tree system. This process takes into account the geometric relationship between the support points and the structure of the tree system, leading to a significantly different projection mechanism.
>
> Once projected, the TSW framework leverages the closed-form solution for optimal transport on tree-metric spaces to compute the TSW distance efficiently. This makes the computational complexity of TSW comparable to that of SW. However, TSW has been shown to yield superior empirical performance, making it a promising alternative to the traditional SW distance and motivating further study of the tree-sliced approach.
>
> ---
>
> We address the concerns raised by the Reviewer in the **Weaknesses** and **Questions** sections as follows.
>
> ---
>
> **Q1. What is the tree metric $d_\mathcal{T}$? How does it relate to $d$?**
>
> **Answer Q1.** Here, the $d_\mathcal{T}$ is the tree metric of the metric space $\mathcal{T}$. We are not sure what $d$ that the Reviewer mentioned, and assume it is the $d_a$ in the Equation (9). Heree, $d_a$ is a metric on the space of measures, that is studied in literature.
>
> **Q2. I wasn't clear on what $\alpha$ is. Unless I missed something, it first gets used just above eq (10) but isn't defined until after eq (13). And even then I wasn't sure what it means. Could you add more detail?**
>
> **Answer Q2.** We acknowledge that, during the compression of the paper to meet the required length constraints, we inadvertently omitted the definition of the notion $\alpha$. The map $\alpha$, referred to as the splitting map, is an element of the space $\mathcal{C}(\mathbb{R}^d \times \mathbb{T}^d_k, \Delta_{k-1})$. Its definition is provided in Appendix C, specifically in Appendix C.2 (line 685).
>
> This map encodes the mass-splitting mechanism: for each support point $x \in \mathbb{R}^d$, the density is distributed across the $k$ lines of a tree system $\mathcal{T} \in \mathbb{T}^d_k$ according to the probability vector $\alpha(x, \mathcal{T}) \in \Delta_{k-1}$. While the framework allows for any choice of $\alpha$ in $\mathcal{C}(\mathbb{R}^d \times \mathbb{T}^d_k, \Delta_{k-1})$, the choice proposed in [1], based on the relative distances between the point $x$ and the lines of the tree system, has been shown to perform well in practice. This motivates the specific form of $\alpha$ used in our submission, as defined in Equation (13).
>
> We thank the reviewer for identifying this omission and will revise the submission accordingly to clarify the definition and role of $\alpha$.
>
> **W2. It also wasn't clear how the tree graphs were constructed in the experimental section.**
>
> **Q3. In Section 2.2, the paragraph on Tree System finishes with specifying the distribution $\sigma$. It's just stated that this is a "random sampling procedure", is it uniform? Or any probability distribution over lines?**
>
> **Answer W2+Q3.** The detailed construction of the distribution $\sigma$ is provided in Appendix C.1. This distribution is defined based on the procedure used to construct the tree system, specifically through the sampling of the root point and the lines that form the tree structure.
>
> **Q4. I didn't find the sentence "we then compute the regularized transport cost $\mathcal{T}_{\epsilon}(\mu, \nu)$ as in Equation (10)" very clear. Eq (10) defines what I understand to be a 1-D projection. I feel there is a step missing in the logic. Could you add more detail.**
>
> **Answer Q4.** We acknowledge the typo in the original sentence and thank the reviewer for pointing it out. The corrected sentence should read: “We then compute the regularized transport cost $\mathcal{T}\_{a}(\mu\_\mathcal{T}, \nu\_\mathcal{T})$ as in Equation (9).” Here, Equation (9) explicitly defines the computation of $\mathcal{T}\_{a}(\mu\_\mathcal{T}, \nu\_\mathcal{T})$. We kindly invite the reviewer to refer to this equation for clarification.
>
> **Q5. Where does "entropy" in the name come from?**
>
> **Answer Q5.** The term “Entropy” originates from Equations (5) and (6), where the Partial Transport problem is formulated with regularization terms expressed as weighted relative entropies.
>
> **W1. Although I wasn't sure why more comparisons weren't included in Section 4.1 (for example the methods in Section 4.2)**
>
> **Answer W1.** The baselines were carefully selected for each experiment to ensure fair and meaningful comparisons based on both relevance and feasibility.
>
> The gradient flow experiment (Sec 4.1) was designed as a focused comparison to isolate the benefit of our partial transport approach. We included only the direct balanced counterparts (SW and Db-TSW) to clearly demonstrate that the partial mechanism is a necessary advancement for handling outliers.
>
> The robust generative modeling (Sec 4.2) and image-to-image translation (Sec 4.3) experiments provide a comprehensive evaluation against a wide range of state-of-the-art Unbalanced/Partial OT methods. We separated the baselines for two key technical reasons:
> * The methods from Sec 4.2 (e.g., USOT, SUOT, SPOT) proved to be numerically unstable in the high-dimensional setting of Sec 4.3, frequently causing model divergence in our preliminary experiments.
> * Conversely, the methods from Sec 4.3 (UOT-FM and ULightOT) use highly specialized parameterizations (flow-matching networks and GMMs). Adapting them to the generative modeling task in Sec 4.2 would have required architectural changes, deviating from their original implementations and risking an unfair comparison.
>
> We believe this approach provides the most rigorous evaluation, as PartialTSW is benchmarked against the most recent baselines within technical constraints of each task (e.g., PAWL [3] for Sec 4.2; UOT-FM [4] and ULightOT [5] for Sec 4.3).
>
> ---
>
> **References.**
>
> [1] Viet-Hoang Tran et al., Distance-based Tree-Sliced Wasserstein Distance. ICLR 2025.
>
> [2] Viet-Hoang Tran et al., Tree-Sliced Wasserstein Distance: A Geometric Perspective. ICML 2025.
>
> [3] Laetitia Chapel and Romain Tavenard, One for all and all for one: Efficient computation of partial Wasserstein distances on the line. ICLR 2025.
>
> [4] Luca Eyring et al., Unbalancedness in Neural Monge Maps Improves Unpaired Domain Translation. ICLR 2024.
>
> [5] Milena Gazdieva et al., Light Unbalanced Optimal Transport. NeurIPS 2024.
>
> ---
>
> We sincerely thank the reviewer for the constructive feedback and for identifying the typographical errors in our submission. We will make the necessary revisions to address these issues. If the provided clarifications are deemed satisfactory, we would be grateful if the reviewer could consider adjusting the evaluation accordingly. We remain at the reviewer’s disposal and would be pleased to address any additional concerns in the subsequent stage of the discussion.

---

> > ### Comment · Reviewer_nitj · 2025-08-05
> >
> > I thank the reviewers for their response. And I accept that I am not familiar with tree-sliced distances (hence my low confidence rating which should be taken into account). I did look at the Appendix A and C, and it is helpful to explain tree-sliced distances. But the paper is 54 pages long and I haven't had time to look at every part of it (with the other 5 Neurips reviews plus my usual commitments). I still have a lot of confusion with the paper: for example, the motivation behind regularising with entropy isn't clear to me with sliced distances. Isn't there already a closed form solution for 1D problems (at least there is for the classical sliced-Wasserstein distance, but I don't know if this holds for tree-sliced distances)? I'm not sure what regularising with entropy gives you.
> >
> > I'm also not convinced that generalized sliced Wasserstein distances (reference [34]) would not be competitive.
> >
> > I've decided to maintain my score.

---

> ### Author Response · Authors · 2025-08-05
>
> We thank the Reviewer for their valuable feedback.
>
> We believe the source of the Reviewer’s confusion stems from a fundamental distinction: **our work addresses the Unbalanced Optimal Transport (UOT) problem, not the classical Optimal Transport (OT) problem**.
>
> > “The motivation behind regularising with entropy isn't clear to me with sliced distances. Isn't there already a closed-form solution for 1D problems (at least there is for the classical sliced-Wasserstein distance, but I don't know if this holds for tree-sliced distances)? I'm not sure what regularising with entropy gives you.”
>
> **In the case of UOT, there is no closed-form solution, even in 1D**. However, as discussed in Section 2.1, we observe that on tree-metric spaces, **introducing entropic regularization not only results in a valid metric, but also yields a closed-form solution**. This is a significant theoretical and practical advantage that does not hold in the unregularized UOT setting. Furthermore, since this approach is not directly applicable in 1D, the tree-sliced framework becomes essential to enable both theoretical and computational tractability in our setting.
>
> > “I'm also not convinced that generalized sliced Wasserstein distances (reference [34]) would not be competitive.”
>
> We would like to clarify that the Generalized Sliced Wasserstein (GSW) distance and other sliced OT variants are **mostly designed for balanced OT problems**. As our focus is on the unbalanced case, **these methods are not applicable to our setting**. We have therefore chosen appropriate baselines that are tailored to the UOT problem, ensuring a fair and meaningful comparison.
>
> ---
>
> We sincerely acknowledge the significant workload that Reviewers undertake, and we truly appreciate your time and effort. As researchers, we also serve as reviewers and understand the demands involved in this important role.
>
> We respectfully hope that you may consider revisiting our submission. Feedback and evaluation from experts in the field are extremely valuable to us and play a crucial role in ensuring a fair and constructive assessment of our work.
>
> Once again, we thank you for your thoughtful comments and engagement.

---

> > ### Author Response · Authors · 2025-08-08
> >
> > Dear Reviewer,
> >
> > As we approach the final days of the discussion phase, we would like to kindly follow up regarding the concerns you raised during the review process. We sincerely hope that our responses have addressed your questions and clarified the key aspects of our work.
> >
> > If you find our clarifications satisfactory, we would be grateful if you could consider updating your evaluation to reflect this. Of course, if there remain any unresolved points or further questions, we would be more than happy to continue the discussion.
> >
> > We truly value the thoughtful feedback we have received throughout the review process. Engaging with experts across different areas has greatly contributed to strengthening our work, and we are thankful for the opportunity to benefit from your insights.
> >
> > Warm regards,
> >
> > The Authors

---

### Official Review · Reviewer_fEkm · 2025-07-03

**Clarity:** 4
**Significance:** 3
**Originality:** 3
**Rating:** 5
**Confidence:** 4

**Summary:**

This paper extends the entropy partial transport framework on the tree sliced transport. Previously, the partial transport was introduced to address the issue of unbalanced mass, and the tree sliced transport was introduced to alleviate the computation burden of optimal transport and remedy the curse of dimensionality.  The proposed PartialTSW inherits the benefit of both frameworks. Theoretically the authors show that PartialTSW is a metric invariant to Euclidean transformations. In the experiments, the noise robustness of the proposed method is shown in the tasks of point cloud alignment, outlier rejection in generative modeling and image-to-image translation.

**Questions:**

1. Could the authors provide insights into the asymptotic behavior of PartialTSW with respect to the parameter $b$?
3. Besides tuning $\nu(\mathcal{T})$ and fixing the slitting map to be softmax, how are the remaining regularization parameters chosen?
5. Why are different baselines considered in different experiments?
6. In the construction of tree systems, all lines pass through a common point $x$ in Algorithm 1. Is there a specific reason for this design choice? Additionally, how would the performance of the proposed method be affected if different $x_i$'s were randomly sampled instead?

**Ethical Concerns:**

["NO or VERY MINOR ethics concerns only"]

**Limitations:**

yes

**Quality:**

4

**Strengths And Weaknesses:**

Strengths:
1. The motivation is solid and clear. In practical applications, most distributions for matching are imbalanced, and efficiency is crucial. PartialTSW improves the robustness and efficiency of the classic optimal transport.
2. The proposed PartialTSW is novel, and the theoretical justifications are well-written.
3. The outlier robustness of PartialTSW is excellent and significant in experiments. I appreciate the experimental details given in Appendix E that help the reader understand the setting.

Weaknesses:
Including a discussion on sample complexity and projection complexity would enhance the completeness of the paper.

---

> ### Author Rebuttal · Authors · 2025-07-31
>
> We address the concerns raised by the Reviewer in the **Weaknesses** and **Questions** sections as follows.
>
> ---
>
> **W1. Including a discussion on sample complexity and projection complexity would enhance the completeness of the paper.**
>
> **Answer W1.** Similar to other applications of Monte Carlo (MC) estimation, the approximation error of the Tree-Sliced Wasserstein (TSW) distance decreases at a rate of $\mathcal{O}(L^{-1/2})$, where $L$ denotes the number of independently sampled tree slices. This rate reflects the standard convergence behavior of MC-based approximations.
>
> Concretely, as discussed in the computational complexity analysis in Section 3 (lines 208), each tree slice in TSW contains $k$ projection lines. Therefore, with $L$ tree samples, the total number of projections is $Lk$. To ensure a fair comparison, we maintain the same total number of projections across all methods by adjusting $L$ accordingly, i.e., $L = \text{(total number of projections)} / k$.
>
> In the literature on TSW, it has been observed that using more than one line per tree slice ($k > 1$) enhances the capacity of the method to capture topological and structural features of the data. Thus, there is a natural trade-off: increasing $k$ improves topological expressiveness but requires reducing $L$ to preserve a fixed computational budget.
>
> In our experiments, we tune the number of lines per tree slice $k$ based on empirical performance and adjust the number of tree slices $L$ accordingly to ensure that the total number of projections $Lk$ remains fixed. This approach enables us to explore the advantages of using richer tree structures through larger values of $k$ while maintaining a fair and consistent basis for comparison with other baseline methods under the same computational budget. Empirically, this strategy consistently leads to improved performance, highlighting the effectiveness of balancing structural expressiveness with projection efficiency.
>
> **Q1. Could the authors provide insights into the asymptotic behavior of PartialTSW with respect to the parameter $b$?**
>
> **Answer Q1.** In the Entropy Partial Transport (EPT) problem on tree metric spaces, the parameter $b$ appears as a weighting term in front of the transport cost:
>
> $W\_m(\mu, \nu) = \inf\_{\substack{\gamma \in \Pi\_{\leq}(\mu, \nu)\\ \gamma(\mathcal{T} \times \mathcal{T}) = m}} \left[ \mathcal{F}\_1(\gamma\_1 \mid \mu) + \mathcal{F}\_2(\gamma\_2 \mid \nu) + b \int\_{\mathcal{T} \times \mathcal{T}} d\_\mathcal{T}(x, y) \, \gamma(dx, dy) \right]$
>
> As $b \to \infty$, the term $b \int\_{\mathcal{T} \times \mathcal{T}} d\_\mathcal{T}(x, y) \, \gamma(dx, dy)$ dominates the objective. This corresponds to **penalizing the transport cost much more heavily** than the entropy terms $\mathcal{F}\_1$ and $\mathcal{F}\_2$. In this regime, the optimization will strongly prefer couplings that minimize the transport distance, possibly at the cost of higher entropy (i.e., less smooth or less spread-out plans).
>
> In the limit $b \to \infty$, the optimizer would attempt to select a coupling $\gamma$ that is highly concentrated around minimizing the distance $d_\mathcal{T}(x, y)$, potentially leading to deterministic or near-deterministic couplings (subject to marginal constraints and the entropy penalty).
>
> An analogous behavior is observed as $b \to 0$. In conclusion,
>
> * **As $b \to 0$**: the entropy terms dominate; the optimizer prefers smooth, spread-out couplings with less concern for transport distance.
>
> * **As $b \to \infty$**: the transport distance dominates; the optimizer prioritizes short transport paths, sacrificing entropy regularity.
>
> Thus, $b$ controls the *bias-variance trade-off* between smoothness (entropy) and accuracy (transport cost). In practice, tuning $b$ balances regularization and fidelity.
>
>
> **Q2. Besides tuning $\nu({\mathcal{T}})$ and fixing the slitting map to be softmax, how are the remaining regularization parameters chosen?**
>
> **Answer Q2.** As discussed in Section 3.3, other regularization parameters (such as $a, b,$ and $\lambda$) are not necessary for our method's primary application. We focus on computing gradients with respect to sample **locations**, while the sample **masses** are fixed and uniform (e.g., $1/n$) within a mini-batch.
>
> Since the model cannot learn to alter these masses, the parameters that would control mass-related penalties become irrelevant to the training process. This simplifies the formulation, leaving the transported mass ratio, $\nu(\mathcal{T})$, as the sole and intuitive hyperparameter for controlling the degree of partiality.
>
> **Q3. Why are different baselines considered in different experiments?**
>
> **Answer Q3.** The baselines were carefully selected for each experiment to ensure fair and meaningful comparisons based on both relevance and feasibility.
>
> The gradient flow experiment (Sec 4.1) was designed as a focused comparison to isolate the benefit of our partial transport approach. We included only the direct balanced counterparts (SW and Db-TSW) to clearly demonstrate that the partial mechanism is a necessary advancement for handling outliers.
>
> The robust generative modeling (Sec 4.2) and image-to-image translation (Sec 4.3) experiments provide a comprehensive evaluation against a wide range of state-of-the-art Unbalanced/Partial OT methods. We separated the baselines for two key technical reasons:
> * The methods from Sec 4.2 (e.g., USOT [1], SUOT [1], SPOT [2]) proved to be numerically unstable in the high-dimensional setting of Sec 4.3, frequently causing model divergence in our preliminary experiments.
> * Conversely, the methods from Sec 4.3 (UOT-FM and ULightOT) use highly specialized parameterizations (flow-matching networks and GMMs). Adapting them to the generative modeling task in Sec 4.2 would have required architectural changes, deviating from their original implementations and risking an unfair comparison.
>
> We believe this approach provides the most rigorous evaluation, as PartialTSW is benchmarked against the most recent baselines within technical constraints of each task (e.g., PAWL [3] for Sec 4.2; UOT-FM [4] and ULightOT [5] for Sec 4.3).
>
> **Q4. In the construction of tree systems, all lines pass through a common point $x_0$ in Algorithm 1. Is there a specific reason for this design choice? Additionally, how would the performance of the proposed method be affected if different $x_0$'s were randomly sampled instead?**
>
> **Answer Q4.** The design of the tree system, in which each tree consists of $k$ lines emanating from a common root point, facilitates efficient algorithmic implementation. This design follows the approach in [6], where the authors provide a significantly more optimized implementation for such a tree structure, in contrast to the original formulation in [7] on Tree-Sliced Wasserstein (TSW).
>
> Our choice to adopt the tree-sliced framework, as introduced in [6, 7], is primarily driven by a foundational result of Caffarelli and McCann [8], which is summarized in lines 102–118 of our submission and elaborated in detail in Appendix B. Their result pertains to Entropic Partial Transport on tree-metric spaces. As outlined in Appendix B (lines 594–600), the construction involves the introduction of an auxiliary point $\hat{s}$ into the space, which in turn necessitates working within a tree-metric structure rather than a conventional one-dimensional domain. This structural requirement naturally motivates the use of the tree-sliced framework over the traditional sliced approach.
>
> It is important to note that in our implementation, the root point $x_0$ is not fixed. Rather, different $x_0$'s are sampled when computing the distance. As shown in Equation (17), the distance is estimated via Monte Carlo approximation by sampling $L$ tree systems, each consisting of $k$ lines. Each tree system is rooted independently, with its own randomly sampled $x_0$.
>
> ---
> **References.**
>
> [1] Clément Bonet, et al., Slicing Unbalanced Optimal Transport. TMLR 2024.
>
> [2] Nicolas Bonneel and David Coeurjolly, SPOT: Sliced Partial Optimal Transport. ACM Transactions on Graphics (TOG) 2019.
>
> [3] Laetitia Chapel and Romain Tavenard, One for all and all for one: Efficient computation of partial Wasserstein distances on the line. ICLR 2025.
>
> [4] Luca Eyring et al., Unbalancedness in Neural Monge Maps Improves Unpaired Domain Translation. ICLR 2024.
>
> [5] Milena Gazdieva et al., Light Unbalanced Optimal Transport. NeuRIPS 2024.
>
> [6] Viet-Hoang Tran et al., Distance-based Tree-Sliced Wasserstein Distance. ICLR 2025.
>
> [7] Viet-Hoang Tran et al., Tree-Sliced Wasserstein Distance: A Geometric Perspective. ICML 2025.
>
> [8] Luis A Caffarelli and Robert J McCann, Free boundaries in optimal transport and Monge Ampere obstacle problems. Annals of Mathematics 2010.
>
> [9] Kilian Fatras et al., Unbalanced minibatch Optimal Transport; applications to Domain Adaptation. ICML 2021.
>
> [10] Chuyu Zhang et al., P$^2$OT: Progressive Partial Optimal Transport for Deep Imbalanced Clustering. ICLR 2024.
>
> [11] Henri De Plaen et al., Unbalanced Optimal Transport: A Unified Framework for Object Detection. CPVR 2023.
>
> ---
>
> We sincerely thank the Reviewer for the constructive feedback and for identifying the typographical errors in our submission. We will make the necessary revisions to address these issues. If the provided clarifications are deemed satisfactory, we would be grateful if the Reviewer could consider adjusting the evaluation accordingly. We remain at the Reviewer’s disposal and would be pleased to address any additional concerns in the subsequent stage of the discussion.

---

> > ### Comment · Reviewer_fEkm · 2025-08-05
> >
> > I appreciate the authors' clarifications. My concerns about the different baselines and the design of the tree system are well addressed. I agree with the authors' comment about the sample complexity. However, I would like to ask these follow-up questions:
> > * Can you provide trade-off comparisons between computation time and performance, one for varying $k$ (with $L$ fixed), and one for varying $L$ (with $k$ fixed)? I agree with the rationale of fixing the computation budget $Lk$. But the effect of $L$ and $k$ should be disentangled on the distance values or performances on tasks. Given the limited response time, I would appreciate any further discussion on this point. In future revisions, I believe it is necessary to include a detailed discussion on sample complexity, computation considerations and trade-off analysis.
> > * It is clear from the definition how $b$ affects the PartialTSW. Specifically, I was wondering about the limits, similar to Sinkhorn divergence interpolates between OT and MMD [1]. For example, when $b\rightarrow \infty$, it will recover a unregularized version of Tree-Sliced Partial Transport. Will it recover any well-known distance with $b\rightarrow 0$?
> >
> > Other comments:
> > * Since the Partial TSW has multiple parameters, it would be great if the authors could add a table summarizing the effects of each of them. For example, the conclusion in the authors' response on $b$ is a wonderful explanation. Also in Section E.4. lines 962-964, the authors did a great job clarifying $\nu(\mathcal{T})$. This would clear up some confusion about the experimental strategies.
> >
> > I will stick to my recommendation to accept this paper. That said, it could be further strengthened if the authors incorporate the comments above in the revisions.
> >
> > [1] Feydy, J., Séjourné, T., Vialard, F.X., Amari, S.I., Trouvé, A. and Peyré, G., 2019, April. Interpolating between optimal transport and mmd using sinkhorn divergences. In The 22nd international conference on artificial intelligence and statistics (pp. 2681-2690). PMLR.

---

> > > ### Author Response · Authors · 2025-08-07
> > >
> > > We thank the Reviewer for their insightful questions and comments. Below, we provide our final responses to the points raised.
> > >
> > > > Can you provide trade-off comparisons between computation time and performance, one for varying $k$ (with $L$ fixed), and one for varying $L$ (with $k$ fixed)? I agree with the rationale of fixing the computation budget $Lk$. But the effect of $L$ and $k$ should be disentangled on the distance values or performances on tasks. Given the limited response time, I would appreciate any further discussion on this point. In future revisions, I believe it is necessary to include a detailed discussion on sample complexity, computation considerations and trade-off analysis.
> > >
> > > **Answer.** We greatly appreciate this thoughtful suggestion. While our current experiments have primarily focused on the setting with a fixed projection budget $Lk$, we fully agree that a more granular investigation - independently varying $L$ and $k$ - would provide deeper insights into the trade-offs between computational cost and performance. Due to current constraints on computational resources and time, we were unable to include this ablation in this discussion phrase. We fully acknowledge the importance of such an analysis and will include it in the revision.
> > >
> > > > It is clear from the definition how $b$ affects the PartialTSW. Specifically, I was wondering about the limits, similar to Sinkhorn divergence interpolates between OT and MMD [1]. For example, when $b\rightarrow \infty$, it will recover a unregularized version of Tree-Sliced Partial Transport. Will it recover any well-known distance with $b\rightarrow 0$?
> > >
> > > **Answer.** When $b=0$, one has $W_m(\mu, \nu) = \inf_{\substack{\gamma \in \Pi_{\le (\mu, \nu)}, \gamma(\mathcal{T} \times \mathcal{T}) = m}} \left[ \mathcal{F}_1(\gamma_1 \mid \mu) + \mathcal{F}_2(\gamma_2 \mid \nu)  \right]$
> > >
> > > One has $\mathcal{F}\_1(\gamma_1 \mid \mu)  = \int_\mathcal{T} w(x) \mathcal{F}(f_1(x)) \mu(dx) = \int_\mathcal{T} w(x)[\mu(x)- \gamma_1(x)] dx$
> > >
> > > WLOG, assume that $\mu(\mathcal{T}) \ge \nu(\mathcal{T})$. Then, if $m = m' = \nu(\mathcal{T})$, one has $\mathcal{F}_2(\gamma_2 \mid \nu) = 0$.
> > >
> > > Thus,
> > >
> > > $W_{m'}(\mu, \nu) =  \inf_{\gamma_1(\mathcal{T}) = m'} \int_\mathcal{T} w(x)[\mu(x)- \gamma_1(x)] dx$
> > >
> > > This is the derivation for the $W_{mm'}(\mu, \nu)$. In the uniform case, where $w(x) = c$ (const $c$), then $W_m(\mu, \nu) = k(\mu(\mathcal{T}) - \nu(\mathcal{T}))$.
> > >
> > > > Since the Partial TSW has multiple parameters, it would be great if the authors could add a table summarizing the effects of each of them. For example, the conclusion in the authors' response on $b$ is a wonderful explanation. Also in Section E.4. lines 962-964, the authors did a great job clarifying $\nu(\mathcal{T})$. This would clear up some confusion about the experimental strategies.
> > >
> > > We are glad to hear that our earlier clarifications regarding the parameter $b$ and the role of $\nu(\mathcal{T})$ were helpful in resolving the confusion. Following your advice, we will include a table in the revised version of the manuscript summarizing the effects and interpretations of the key parameters in Partial-TSW.
> > >
> > > ---
> > >
> > > We once again thank the Reviewer for their time and effort throughout this discussion. We sincerely appreciate the opportunity to engage in thoughtful dialogue with fellow scholars, which we believe contributes meaningfully to improving the quality and clarity of our work.

---

### Decision · Program_Chairs · 2025-09-17

**Decision:**

Accept (poster)

**Comment:**

This paper proposes a new method for partial optimal transport (with entropic divergence) using tree-sliced
OT. The authors prove metric properties of the method (with  E(d)-invariance)
and provide a closed form approximation for computing it. They evaluate the
method on point cloud gradient flow, robust generative modeling and unbalanced image to
image translation and their proposed distance is competitive with the selected
baselines.

The reviewers found the paper interesting and rigorous but had some
concerns about the originality of the method (combination of Tree Sliced and
Unbalanced OT), the clarity of the presentation
(lacking some details), and the limited numerical experiments (missing
sensitivity to L and k independently, missing more image translation exp.).

The authors did a nice and detailed response to all those points with clarifications
and additional experiments. The reviewers appreciated those clarifications and
new results (L/K, image translation) and now agree that the paper can be accepted
if the authors include all those in the final version.

I agree with the reviewers that this paper is interesting and has a good
theoretical contribution and provides a new method for partial tree-sliced
optimal transport. I recommend an accept and expect the final version to include
the discussed improvements.